# ANOVA-NODE: An identifiable neural network for the functional ANOVA model for better interpretability

## Abstract

Interpretability for machine learning models is becoming more and more important as machine learning models become more complex. The functional ANOVA model, which decomposes a high-dimensional function into a sum of lower dimensional functions so called *components*, is one of the most popular tools for interpretable AI, and recently, various neural network models have been developed for estimating each component in the functional ANOVA model. However, such neural networks are highly unstable when estimating components since the components themselves are not uniquely defined. That is, there are multiple functional ANOVA decompositions for a given function. In this paper, we propose a novel interpretable model which guarantees a unique functional ANOVA decomposition and thus is able to estimate each component stably. We call our proposed model ANOVA-NODE since it is a modification of Neural Oblivious Decision Ensembles (NODE) for the functional ANOVA model. Theoretically, we prove that ANOVA-NODE can approximate a smooth function well. Additionally, we experimentally show that ANOVA-NODE provides much more stable estimation of each component and thus much more stable interpretation when training data and initial values of the model parameters vary than existing neural network models do.

## 1 Introduction

Interpretability has become more important as artificial intelligence (AI) models have become more sophisticated and complicated in recent years. Various methods for interpretable AI can be categorized into two groups. One is transparent box design, where interpretable machine learning models such as linear models and decision trees are used to learn a prediction model. Typically, interpretable models are inferior to black box models in terms of prediction powers. The other group consists of post-hoc interpretation methods that try to interpret a given black box models (Lundberg, 2017; Ribeiro et al., 2016). While post-hoc interpretation methods do not hamper the prediction power of a given black box model at all, they often exhibit instability and lack faithfulness (Slack et al., 2020).

In this paper, we focus on the transparent box design based on the functional ANOVA model (Hoeffding, 1992). The functional ANOVA model approximates a given complex high-dimensional function by the sum of low dimensional (e.g., one or two dimensional) functions referred to as *components*. One of the most representative examples of the functional ANOVA model is the generalized additive model (GAM, Hastie & Tibshirani (1987)), which consists of the summation of one-dimensional functions, each corresponding to an input feature. Low dimensional functions are easier to understand, and thus the functional ANOVA model is popularly used for interpretable AI (Lengerich et al., 2020; Märtens & Yau, 2020).

Recently, various learning algorithms for the functional ANOVA model based on neural networks have been proposed (Agarwal et al., 2021; Radenovic et al., 2022; Chang et al., 2021). While they provide accurate prediction models, existing neural networks struggle to estimate each component in the functional ANOVA model due to unidentifiability (Lengerich et al., 2020). That is, completely different components could result in the same prediction model. Figure 1 presents the plots of the main effects estimated by $NA^2M$, $NB^2M$, and our proposed ANOVA-$N^2ODE$ on California

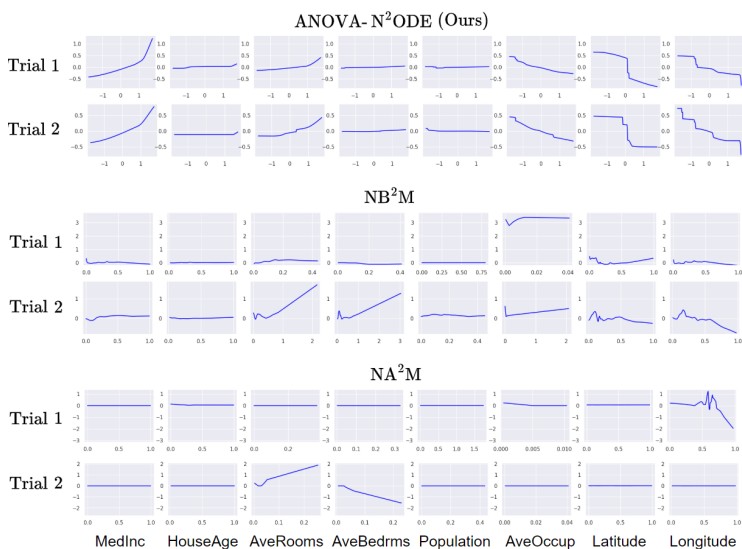

Figure 1: **Plots of the estimated main components on CALHOUSING dataset.**

HOUSING dataset for two trials. Figure 1 shows that our proposed model reliably estimates the components, whereas the other models do not. The experimental results for additional trials can be found in Appendix F.1. Reliable estimation of components is crucial for interpretation, as the functional ANOVA model is interpreted through the interpretation of each component.

A simple remedy for resolving the issue of unidentifiability is to impose a constraint on the components to make them identifiable. There are several such constraints, some of which can be found, for example, in Gu & Wahba (1993); Chastaing et al. (2012); Hooker (2007). One of the most popular identifiability constraints is the so called 'sum-to-zero' condition, which ensures that each component is orthogonal in a certain Hilbert space.

Learning the functional ANOVA model by using neural networks under such an identifiability constraint, however, is not easy since a standard gradient descent algorithm could not be applicable. The aim of this paper is to develop a specially designed neural network that automatically satisfies the sum-to-zero condition and thus can be learned by a standard gradient descent algorithm. The proposed neural network is a modification of Neural Oblivious Decision Ensembles (NODE, Popov et al. (2019)) for the functional ANOVA model. Chang et al. (2021) propose a modification of NODE for the GAM called NODE-GAM, but NODE-GAM is not effective in estimating each component due to the identifiability issue. We modify NODE such that each component in the functional ANOVA model is estimated uniquely. We call our algorithm ANOVA-NODE.

There exist various learning algorithms for the functional ANOVA model under the sum-to-zero condition such as Gu & Wahba (1993); Lin & Zhang (2006); Kim et al. (2009) that do not use neural networks. An advantage of ANOVA-NODE compared to these non-neural algorithms is that a gradient descent based optimization algorithm can be used in learning and hence end-to-end learning is possible when it is combined with other neural networks. See Section 4.5 for an example.

Overall, our contributions are as follows.

- We propose a novel XAI model (ANOVA-NODE) which trains the functional ANOVA model under the sum-to-zero condition using a gradient descent-based optimization algorithm.

- We prove the universal approximation property in the sense that ANOVA-NODE can approximate any smooth function up to an arbitrary precision.

- By analyzing multiple benchmark datasets, we illustrate that ANOVA-NODE provides more stable estimation and interpretation of each component compared to the baseline models, including NAM (Agarwal et al., 2021), NBM (Radenovic et al., 2022), NODE-GAM (Chang et al., 2021) and XGB (Chen & Guestrin, 2016) without losing prediction accuracy.

## 2 BACKGROUND

### 2.1 NOTATION

Let $\mathbf{x} = (x_1, ..., x_p)^\top \in \mathcal{X} = \mathcal{X}_1 \times ... \times \mathcal{X}_p$ be a vector of input features, where we assume $\mathcal{X} \subseteq [-a, a]^p$ for some $a > 0$. We denote $[p] = \{1, \ldots, p\}$, and denote its power set as $\mathcal{P}([p])$. For $\mathbf{x} \in \mathcal{X}$ and $S \subseteq [p]$, let $\mathbf{x}_S = (x_j, j \in S)^\top$. We denote $f_S$ as a function of $\mathbf{x}_S$. For a real-valued function $f : \mathcal{X} \to \mathbb{R}$, we denote $||f||_\infty = \sup_{\mathbf{x} \in \mathcal{X}} |f(\mathbf{x})|$.

### 2.2 FUNCTIONAL ANOVA MODEL

The functional ANOVA model (Hoeffding, 1992) decomposes a high-dimensional function $f$ into the sum of low-dimensional functions

$$f(\mathbf{x}) = \beta_0 + \sum_{j=1}^{p} f_j(x_j) + \sum_{j<k} f_{jk}(x_j, x_k) + \cdots,$$

which is considered as one of the most important XAI tools. Here, $f(\mathbf{x}) = g(\mathbb{E}(Y|X = x))$ where $g$ is a link function and $Y$ is target variable, $f_j, j \in [p]$ are called the main effects, and $f_{j,k}, (j, k) \in [p]^2$ are called the second interaction terms and so on. In practice, only interactions of lower orders (e.g., the main and second order only) are included in the decomposition for a more transparent interpretation.

Generalized Additive Model (GAM, Hastie & Tibshirani (1987)) is a special form of the functional ANOVA model where only the main effects are included in the model, that is

$$f(\mathbf{x}) = \beta_0 + \sum_{j=1}^{p} f_j(x_j).$$

Similarly, GA$^2$M is defined as the functional ANOVA model including all of the main effects and second order interactions,

$$f(\mathbf{x}) = \beta_0 + \sum_{S \subseteq [p], |S| \leq 2} f_S(\mathbf{x}_S),$$

or more generally we can consider GA$^d$M defined as

$$f(\mathbf{x}) = \beta_0 + \sum_{S \subseteq [p], |S| \leq d} f_S(\mathbf{x}_S).$$

Several learning algorithms for the functional ANOVA model have been proposed. Gu & Wahba (1993) applied the smoothing spline to learn the functional ANOVA model, Lin & Zhang (2006) developed a component-wise sparse penalty, and Kim et al. (2009) proposed a boosting algorithm for the functional ANOVA model. In addition, the functional ANOVA model has been applied to various problems such as sensitivity analysis (Chastaing et al., 2012), survival analysis (Huang et al., 2000), diagnostics of high-dimensional functions (Hooker, 2007) and machine learning models (Lengerich et al., 2020; Märtens & Yau, 2020).

Recently, learning the functional ANOVA model using neural networks has received much attention since gradient descent based learning algorithms can be easily scaled up. Examples are Neural Additive Model (NAM, Agarwal et al. (2021)), Neural Basis Model (NBM, Radenovic et al. (2022)) and NODE-GAM (Chang et al., 2021). NAM uses deep neural network (DNN) to train each component of GAM. NBM achieves a significant reduction in training time compared to NAM by using basis DNNs to train all components. NODE-GAM extends Neural Obilvious Decision Ensembles (NODE, Popov et al. (2019)) for GAM.

### 2.3 NEURAL OBLIVIOUS DECISION ENSEMBLES (NODE)

Oblivious Decision Tree (ODT) (Kohavi & Li, 1995) is a decision tree which has the following two properties:

1. All terminal nodes have the same depth.
2. All rules at each depth are identical.

Let $d$ be the depth of ODT, for $t \leq d$, $F^t(\mathbf{x})$ be the feature function used for the rule at the depth $t$, $b^t$ is the split value in the depth $t$ and $\mathcal{B} \in \mathbb{R}^{2^d}$ be the height parameters at the terminal nodes of ODT. Then, the ODT model is given as

$$h(\mathbf{x}) = \mathcal{B}' \left( \begin{bmatrix} \mathbb{I}(F^1(\mathbf{x}) \leq b^1) \\ \mathbb{I}(F^1(\mathbf{x}) > b^1) \end{bmatrix} \otimes \begin{bmatrix} \mathbb{I}(F^2(\mathbf{x}) \leq b^2) \\ \mathbb{I}(F^2(\mathbf{x}) > b^2) \end{bmatrix} \otimes \cdots \otimes \begin{bmatrix} \mathbb{I}(F^d(\mathbf{x}) \leq b^d) \\ \mathbb{I}(F^d(\mathbf{x}) > b^d) \end{bmatrix} \right),$$

where $\mathbb{I}$ is the indicator function and $\otimes$ is the outer product.

To train ODT using gradient descent methods, Popov et al. (2019) replaced the indicator function $\mathbb{I}(F(\mathbf{x}) > b)$ with the $entmax_\nu((\frac{F(\mathbf{x})-b}{\gamma}, 0)')_1$ (Peters et al., 2019) which is differentiable. Moreover, the feature function $F^t$ is a weighed sum of input features. Note that $entmax_\nu((\frac{F(\mathbf{x})-b}{\gamma}, 0)')_1$ works similarly to a sparse sigmoid. They referred to this ODT as Neural ODT (NODT).

In addition, Popov et al. (2019) proposed a layer architecture which involves the output of the current layer being concatenated with the current input and then fed into the next layer. The final output of NODT with a layer architecture can be obtained by averaging the outputs of each layer. Finally, they proposed Neural Oblivious Decision Ensembles (NODE) as an ensemble of NODTs.

## 3 PROPOSED MODEL

### 3.1 IDENTIFIABILITY ISSUE

An unsolved but important problem in neural functional ANOVA algorithms is the *identifiability* of each component. The functional ANOVA model itself is not identifiable. That is, there are multiple functional ANOVA decompositions of a given function. For example,

$$f(x_1, x_2) = f_1(x_1) + f_2(x_2) + f_{12}(x_1, x_2)$$

where $f_1(x_1) = x_1, f_2(x_2) = x_2, f_{1,2}(x_1, x_2) = x_1 x_2$ can be expressed as

$$f(x_1, x_2) = f_1^*(x_1) + f_2^*(x_2) + f_{1,2}^*(x_1, x_2)$$

where $f_1^*(x_1) = -x_1, f_2^*(x_2) = x_2, f_{12}^*(x_1, x_2) = x_1(x_2 + 2)$ (Lengerich et al., 2020). Without the identifiability, each component cannot be estimated uniquely and thus interpretation of the model becomes unstable and inaccurate.

A simple remedy to ensure the identifiability of each component is to put constraints. One of the most popular constraints for the identifiability of the components in the functional ANOVA model is so called the *sum-to-zero* condition (Gu & Wahba, 1993; Kim et al., 2009). When we consider the functional ANOVA model using interactions in $\mathbf{S} \subseteq \mathcal{P}([p])$, the *sum-to-zero* condition is

$$\forall S \in \mathbf{S}, \ \forall j \in S, \ \forall \mathbf{z} \in \mathcal{X}_{S \setminus \{j\}}, \ \int_{\mathcal{X}_j} f_S(\mathbf{x}_{S \setminus \{j\}} = \mathbf{z}, x_j) \mu_j(dx_j) = 0 \tag{1}$$

for some probability measure $\mu_j$ on $\mathcal{X}_j$. In practice, we can use the empirical distribution of the input feature $x_j$ for $\mu_j$ or the uniform distribution. With the sum-to-zero in (1), the functional ANOVA model becomes identifiable, as can be seen in proposition 3.1. Let $\mu = \prod_j \mu_j$.

**Proposition 3.1.** *(Hooker, 2007) Consider two component sets $\{f_S^1, S \in \mathbf{S}\}$ and $\{f_S^2, S \in \mathbf{S}\}$ which satisfy (1). Then, $\sum_{S \in \mathbf{S}} f_S^1(\cdot) \equiv \sum_{S \in \mathbf{S}} f_S^2(\cdot)$ almost everywhere (with respect to $\mu$) if and only if $f_S^1(\cdot) \equiv f_S^2(\cdot)$ almost everywhere (with respect to $\mu$) for every $S \in \mathbf{S}$.*

The sum-to-zero condition is not a unique identifiability condition. However, Herren & Hahn (2022) demonstrated that there is an interesting relation between the sum-to-zero condition and SHAP (Lundberg, 2017) which is a well known interpretable AI method. That is, SHAP value of a given input can be calculated easily from the prediction values of each component under regularity conditions. For a given model $f$ and input vector $\mathbf{x}$, SHAP value of the $j$th input variable is defined as

$$\phi_j(f, \mathbf{x}) = \sum_{S \subseteq [p] \setminus \{j\}} \frac{|S|!(p - |S| - 1)!}{p!} (v_f(S \cup \{j\}) - v_f(S)),$$

where $v_f(S) = \mathbb{E}[f(\mathbf{X})|\mathbf{X}_S = \mathbf{x}_S]$, where $\mathbf{X} \sim \mu$.

**Proposition 3.2.** *(Herren & Hahn, 2022) For a given $f$ which is the $GA^dM$ satisfying the sum-to-zero condition. Then, we have*

$$\phi_j(f, \mathbf{x}) = \sum_{S \subseteq [p], |S| \leq d, j \in S} f_S(\mathbf{x}_S)/|S|. \tag{2}$$

The result in equation (2) provides an interesting implication - the functional ANOVA model satisfying the sum-to-zero condition also decomposes SHAP value. That is, the contribution of the interaction between $x_j$ and $\mathbf{x}_S$ to SHAP value $\phi_j(f, \mathbf{x})$ is $f(\mathbf{x}_{S'})/|S'|$, where $S' = S \cup \{j\}$. Note that this interesting relation is not generally valid for identifiability conditions other than the sum-to-zero condition.

Therefore, for general $GA^dM$, we can calculate SHAP value using equation (2), which we refer to as *ANOVA-SHAP*. One advantage of ANOVA-SHAP is that it is significantly faster to compute compared to Deep-SHAP and Kernel-SHAP proposed by Lundberg (2017), as well as Tree-SHAP proposed by Lundberg et al. (2018). The ANOVA-SHAP experiment results are in Appendix K.

## 3.2 ANOVA-NODE

Incorporating the sum-to-zero condition into existing neural functional ANOVA models would be difficult because standard gradient descent algorithms cannot be applied due to identifiability constraints. The aim of this subsection is to propose a special neural network function for the functional ANOVA model that automatically satisfies the sum-to-zero condition and thus each component can be estimated uniquely by using a standard gradient descent based optimization algorithm.

The main idea of the proposed neural network is to model each component as the sum of special but simple neural networks that satisfy the sum-to-zero condition. That is, we set

$$f_S(\mathbf{x}_S) = \sum_{k=1}^{K_S} h^S(\mathbf{x}_S | \phi_{S,k}),$$

where $\{h^S( \cdot | \phi_{S,k})\}_{k=1}^{K_S}$ are neural networks with learnable parameters $\phi_{S,k}$ satisfying the sum-to-zero condition.

**ANOVA-NODT.** For $h^S( \cdot | \phi_{S,k})$, we propose ANOVA-NODT, which is a modification of ODT, as follows. First, we set the depth of $h^S$ to be equal to $|S|$. For $t \in \{1, ..., |S|\}$, we use the feature function as

$$F^t(\mathbf{x}_S) = (\mathbf{x}_S)_t,$$

where $(\mathbf{x}_S)_t$ is the $t$th feature of $\mathbf{x}_S$. To approximate the indicator function $\mathbb{I}$ in ODT as a differentiable function, we use $entmax_\nu$ as Peters et al. (2019) does. For $b \in \mathbb{R}$ and $\gamma \in \mathbb{R}^+$, we define

$$c_1(x|b, \gamma) := 1 - entmax_\nu\left( \left( \frac{x - b}{\gamma}, 0 \right)' \right)_1, \quad c_2(x|b, \gamma) := 1 - c_1(x|b, \gamma),$$

where $\nu$ is a hyper-parameter. Finally, we define the ANOVA-NODT model $h^S$ as

$$h^S(\mathbf{x}_S) = (\mathcal{B}^S)' \left( \begin{bmatrix} c_1((\mathbf{x}_S)_1|b_1, \gamma_1) \\ c_2((\mathbf{x}_S)_1|b_1, \gamma_1) \end{bmatrix} \otimes \begin{bmatrix} c_1((\mathbf{x}_S)_2|b_2, \gamma_2) \\ c_2((\mathbf{x}_S)_2|b_2, \gamma_2) \end{bmatrix} \otimes \cdots \otimes \begin{bmatrix} c_1((\mathbf{x}_S)_{|S|}|b_{|S|}, \gamma_{|S|}) \\ c_2((\mathbf{x}_S)_{|S|}|b_{|S|}, \gamma_{|S|}) \end{bmatrix} \right),$$

where $\mathcal{B}^S \in \mathbb{R}^{2^{|S|}}$, $b_1, \ldots, b_{|S|}$ and $\gamma_1, \ldots, \gamma_{|S|}$ are learnable parameters.

A novel part of ANOVA-NODT is to parameterize $\mathcal{B}^S$ to make $h^S$ always satisfy the sum-to-zero condition. Here, we address the case where $|S| = 1$, and the case where $|S| = 2$ is described in Appendix B.2. For the case of $S = \{j\}$, we consider

$$h^j(x_j) = \beta_{1,j} c_1(x_j|b_{1,j}, \gamma_{1,j}) + \beta_{2,j} c_2(x_j|b_{1,j}, \gamma_{1,j}).$$

For a given $\theta_j \in \mathbb{R}$, we let $\beta_{1,j} = \theta_j$ and $\beta_{2,j} = -\frac{\mathbb{E}[c_1(X_j|b_{1,j}, \gamma_{1,j})]}{\mathbb{E}[c_2(X_j|b_{1,j}, \gamma_{1,j})]} \theta_j$. Then, it is easy to see that $\mathbb{E}[h^j(X_j)] = 0$, meaning that it satisfies the sum-to-zero condition. Hence, we can parameterize $h^j(x_j)$ by $\phi_j = (\theta_j, b_{1,j}, \gamma_j)$. Note that there is no constraint on $\phi_j$ and thus standard gradient descent algorithms can be used to learn $\phi_j$.

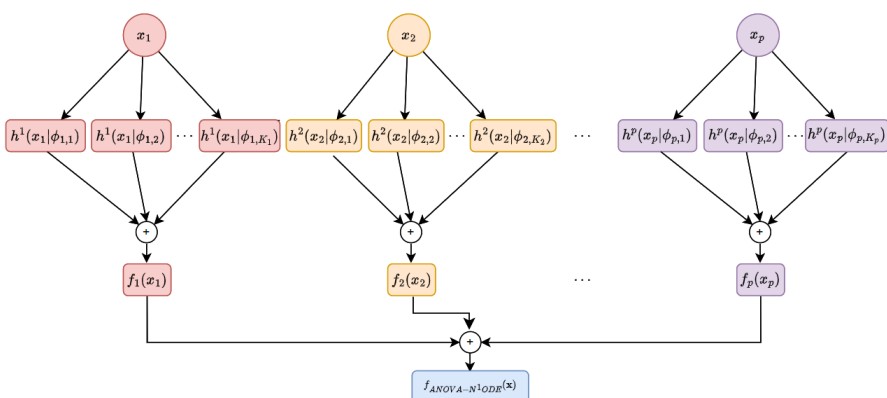

Figure 2: **ANOVA-N$^1$ODE architecture**

Finally, for general $S \subseteq [p]$, ANOVA-NODT can be parameterized by

$$\phi_S = \{\theta_S, (b_{1,S}, \gamma_{1,S}), \ldots, (b_{|S|,S}, \gamma_{|S|,S})\}$$

as

$$h^S(\mathbf{x}_S|\phi_S) = \sum_{\mathbf{s} \in \{1,2\}^{|S|}} \beta_{\mathbf{s},S}(\phi_S) \prod_{i=1}^{|S|} c_{s_i}((\mathbf{x}_S)_i|b_{i,S}, \gamma_{i,S}),$$

where $\mathbf{s} = (s_1, \ldots, s_{|S|})'$ and $\beta_{\mathbf{s},S}(\phi_S)$ are the functions of $\phi_S$ with $\beta_{\mathbf{1},S}(\phi_S) = \theta_S$.

**ANOVA-N$^d$ODE.** ANOVA-N$^d$ODE is an ensemble of ANOVA-NODT for the order of interactions up to $d$. That is, we set

$$f_{\text{ANOVA-N}^d\text{ODE}}(\mathbf{x}) = \sum_{S \subseteq [p], |S| \leq d} \sum_{k=1}^{K_S} h^S(\mathbf{x}_S|\phi_{S,k}),$$

where $K_S$ is the number of ANOVA-NODTs corresponding to the component $S$. In practice, $d = 1$, which corresponds to the GAM, and $d = 2$ are commonly used. The architecture of ANOVA-N$^1$ODE is shown in Figure 2. Note that ANOVA-N$^d$ODE always satisfies the sum-to-zero condition since each ANOVA-NODT does so. Therefore, standard gradient descent algorithms can be applied without modification. Unless there is any confusion, we write ANOVA-NODE instead of ANOVA-N$^d$ODE for general $d$.

An interesting theoretical property of ANOVA-NODE is the universal approximation property as the standard neural network has (Hornik et al., 1989). That is, ANOVA-NODE can approximate any arbitrary GA$^d$M function to a desired level of accuracy, as stated in the following theorem.

**Theorem 3.3.** *Let $g_0(\mathbf{x}) := \sum_{S \subset [p], |S| \leq d} g_{0,S}(\mathbf{x}_S)$ be a given GA$^d$M function satisfying the sum-to-zero condition. Consider the $entmax_\nu$ for $\nu = 1$, and let $\mu$ be a measure for any distribution which has a bounded density. If each $g_{0,S}$ is $L$-Lipschitz continuous[1] for some $L > 0$, then for any $\epsilon > 0$, there exists $f_{ANOVA\text{-}N^d ODE}$ with sufficiently large $K_S$s such that*

$$\left\| g_0(\cdot) - f_{ANOVA\text{-}N^d ODE}(\cdot) \right\|_\infty < \epsilon.$$

**Training** For a given train data and loss function, ANOVA-NODE is trained using any gradient descent algorithm to minimize the empirical risk.

**Data preprocessing.** To satisfy the sum-to-zero condition in ANOVA-NODT, one element in $\mathcal{B}^S$ is a free parameter, while the remaining elements are parameterized by the product of constants (e.g., $-\mathbb{E}[c_1(X_j|b_{1,j}, \gamma_{1,j})]/\mathbb{E}[c_2(X_j|b_{1,j}, \gamma_{1,j})])$ and the free parameter. In this case, since the constants can become close to or equal to zero depending on the trained $\{(b_{i,S}, \gamma_{1,S}), i = 1, ..., |S|\}$, we scale the data by using a transformation based on the quantiles of a uniform distribution to ensure stable learning.

---

[1]A given function $v$ defined on $\mathcal{Z}$ is $L$-Lipschitz continuous if $|v(z_1) - v(z_2)| \leq L\|z_1 - z_2\|$ for all $z_1, z_2 \in \mathcal{Z}$, where $\|\cdot\|$ is a certain norm defined on $\mathcal{Z}$.

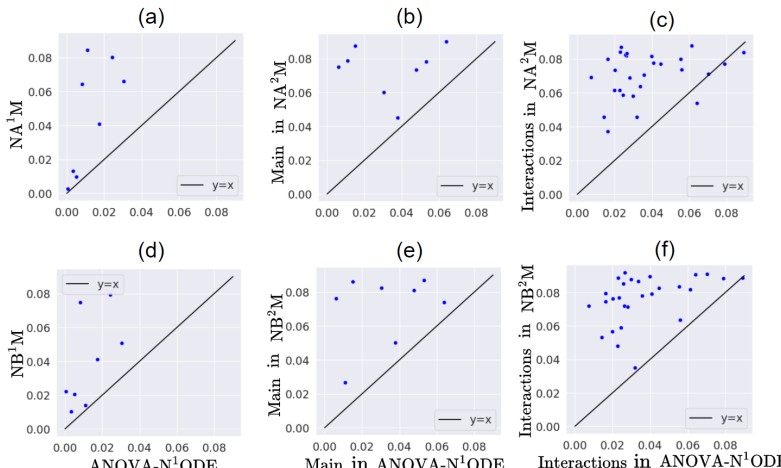

Figure 3: **Scatter plots of the stability scores on CALHOUSING dataset.** Figures (a) and (d) are the scatter plots for the stability scores of the main effects, where the x-axis is the stability score of ANOVA-N$^1$ODE and the y-axis is the stability scores of NA$^1$M in (a) and NB$^1$M in (d). Figures (b) and (e) are the scatter plots for the stability scores of the main effects, where the x-axis is the stability score of ANOVA-N$^2$ODE, and the y-axis is the stability score of NA$^2$M in (b) and NB$^2$M in (e). Figures (c) and (f) compare the stability scores of the second order interactions of ANOVA-N$^2$ODE to those of NA$^2$M and NB$^2$M. Each dot in the scatter plots corresponds to each component.

## 4 EXPERIMENTS

This section presents the results of numerical experiments. More results along with details about data, algorithms and selection of hyper-parameters are provided in Appendices C to M.

### 4.1 STABILITY IN COMPONENT ESTIMATION

Similarly to NAM (Agarwal et al., 2021) and NBM (Radenovic et al., 2022), ANOVA-NODE provides interpretation through the estimated components. Thus, if the components are not estimated stably, the interpretations based on the estimated components would not be reliable. In this subsection, we investigate the stability of the component estimation of ANOVA-NODE compared with the other baseline models including NAM and NBM. For this purpose, we generate randomly sampled training data and estimate the components of the functional ANOVA model. We repeat this procedure 10 times to obtain 10 estimates of each component, and measure how similar these 10 estimates are. For the similarity measure, we use

$$\mathcal{SC}(f_S) = \frac{1}{n} \sum_{i=1}^{n} \frac{\sum_{j=1}^{10} (f_S^j(\mathbf{x}_i) - \bar{f}_S(\mathbf{x}_i))^2}{\sum_{j=1}^{10} (f_S^j(\mathbf{x}_i))^2}$$

for given pre-selected $n$ many input vectors $\mathbf{x}_i$, $i = 1, \ldots, n$, where $f_S^j$, $j = 1, \ldots, 10$ are the 10 estimates of $f_S$ and $\bar{f}_S$ is their average. A smaller value of $\mathcal{SC}(f_S)$ means a more stable estimation (and thus more stable interpretation).

In Figure 3, we present the scatter plots of the stability scores for each estimated component between (NA$^1$M, NB$^1$M) vs ANOVA-N$^1$ODE and (NA$^2$M, NB$^2$M) vs ANOVA-N$^2$ODE on CALHOUSING dataset. It is obvious that our models are more stable than the baselines in component estimation. Consistent results are also observed with other datasets, which are presented in Appendix F.3. Also, the plots of the functional relation of the main effects are provided in Appendix F.1, from which we can feel how much NAM and NBM are unstable in estimating the components.

In addition, we compare the overall stability score $\mathcal{SC}(f) = \sum_S \mathcal{SC}(f_S)$. For each of nine benchmark datasets, we calculate the ratios of the overall stability scores of ANOVA-NODE, NAM, and NBM *normalized* by the overall stability score of ANOVA-NODE, whose results are given in Table 1. The unnormalized results of stability score are in Appendix G.1. The results again confirm that ANOVA-NODE is superior in terms of the stability of component estimation.

The stability of ANOVA-NODE with respect to the choice of initial values are illustrated in Appendix D.2.

Table 2: **Performance of component selection.** We report the averages (standard deviations) of AUROCs of the estimated importance scores of each component on $f^{(1)}$, $f^{(2)}$, $f^{(3)}$ synthetic datasets. The bold faces highlight the best results.

| True model | $f^{(1)}$ | | | $f^{(2)}$ | | | $f^{(3)}$ | | |
|---|---|---|---|---|---|---|---|---|---|
| Models | ANOVA N$^2$ODE | NA$^2$M | NB$^2$M | ANOVA N$^2$ODE | NA$^2$M | NB$^2$M | ANOVA N$^2$ODE | NA$^2$M | NB$^2$M |
| AUROC ↑ | **1.000** (0.00) | 0.330 (0.08) | 0.522 (0.16) | **0.943** (0.01) | 0.311 (0.08) | 0.481 (0.09) | **0.956** (0.02) | 0.381 (0.13) | 0.477 (0.07) |

Table 1: **Stability scores on real datasets.** For each dataset, stability scores of GAM (GA$^2$M) models are normalized by the that of ANOVA-N$^1$ODE (ANOVA-N$^2$ODE). Lower stability score means more stable interpretation. The bold faces highlight the best results.

| | | GAM | | | GA$^2$M | | |
|---|---|---|---|---|---|---|---|
| Dataset | ANOVA N$^1$ODE | NA$^1$M | NB$^1$M | ANOVA N$^2$ODE | NA$^2$M | NB$^2$M |
| CALHOUSING (Pedregosa et al., 2011a) | **1.000** | 3.750 | 3.250 | **1.000** | 2.209 | 2.143 |
| WINE(Cortez et al., 2009) | **1.000** | 5.273 | 3.909 | **1.000** | 1.776 | 1.327 |
| ONLINE (Fernandes et al., 2015) | **1.000** | 2.452 | 1.742 | **1.000** | 1.385 | 1.385 |
| ABALONE (Warwick et al., 1995) | **1.000** | 1.625 | 3.250 | **1.000** | 1.679 | 1.357 |
| FICO (fic, 2018) | **1.000** | 1.314 | 1.314 | **1.000** | 1.854 | 1.563 |
| CHURN(chu, 2017) | **1.000** | 1.588 | 2.765 | **1.000** | 1.894 | 1.702 |
| CREDIT (cre, 2015) | **1.000** | 3.286 | 1.190 | **1.000** | 2.472 | 1.472 |
| LETTER (Slate, 1991) | 1.000 | 1.294 | **0.824** | **1.000** | 2.885 | 1.962 |
| DRYBEAN(dry, 2020) | **1.000** | 2.643 | 2.500 | **1.000** | 1.660 | 1.528 |

## 4.2 PERFORMANCE IN COMPONENT SELECTION

An important implication of stable estimation of the components is the ability of selecting signal components. That is, ANOVA-NODE can effectively identify signal components in the true function by measuring the variations of the estimated components. For example, we can consider the $l_1$ norm of each estimated component (i.e, $\|f_S(\mathbf{x}_S)\|_1$) as the important score, and select the components whose important scores are large. This simple component selection procedure would not perform well if component estimation is unstable.

To investigate how well ANOVA-NODE selects the true signal components, we conduct an experiment similar to the one in Tsang et al. (2017). We generate synthetic datasets from $Y = f(\mathbf{x}) + \epsilon$, where $f$ is the true prediction model and $\epsilon$ is a noise generated from a Gaussian distribution with mean 0 and variance $\sigma_\epsilon^2$. Then, we apply ANOVA-N$^2$ODE, NA$^2$M and NB$^2$M to calculate the importance scores of the main effects and second order interactions and examine how well they predict whether a given component is signal. For the performance measure of component selection, we use AUROC obtained from the pairs of $\|\hat{f}_S\|_1$ and $r_S$ for all $S \subset [p]$ with $|S| \leq 2$, where $\hat{f}_S$ are the estimates of $f_S$ in $f$ and $r_S = \mathbb{I}(\|f_S^{(k)}\|_1 > 0)$ are the indicators whether $f_S$ are signal or not.

For the true prediction model, we consider the three functions $f^{(k)}$, $k = 1, 2, 3$ whose details are given in Appendix C.1. We set the data size to 15,000 and set $\sigma_\epsilon^2$ to make the signal-to-noise ratio become 5. Table 2 compares the AUROCs of ANOVA-N$^2$ODE, NA$^2$M and NB$^2$M, which clearly indicates that ANOVA-N$^2$ODE outperforms the baseline models in component selection. More details of component selection with ANOVA-N$^2$ODE are given in Appendix H.

## 4.3 PREDICTION PERFORMANCE

We compare prediction performance of ANOVA-NODE with baseline models. We randomly split the train, validation and test data into the ratio 70/10/20, where the validation data is used to select the optimal epoch and the test data is used to measure the prediction performance of the estimated models. We repeat this random split 10 times to obtain 10 performance measures for prediction. For the performance measure, we use the Root Mean Square Error (RMSE) for regression datasets and the Area Under the ROC curve (AUROC) for classification datasets.

Table 3 presents the results of prediction performance of ANOVA-NODE, NODE-GAM, NAM, and NBM as well as two black box models. It is obvious that ANOVA-NODE favorably competes with its competitors in view of prediction performance. In addition, at the final line, the average ranks of each model over the nine datasets are given, which shows that ANOVA-NO$^2$DE exhibits comparable prediction performance compared to the baseline models. Details about the experiments are given in Appendix C.2.

Table 3: **Prediction performance.** We report the averages (standard deviations) of the prediction performance measure. In addition, we report the averages of ranks of prediction performance of each model on nine datasets. The optimal (or suboptimal) results are highlighted in **bold** (or underlined).

| Dataset | Measure | GAM | | | | GA$^2$M | | | | Black box | |
|---|---|---|---|---|---|---|---|---|---|---|---|
| | | ANOVA N$^1$ODE | NODE GA$^1$M | NA$^1$M | NB$^1$M | ANOVA N$^2$ODE | NODE GA$^2$M | NA$^2$M | NB$^2$M | XGB | NODE |
| CALHOUSING | RMSE ↓ | 0.614 (0.01) | 0.581 (0.01) | 0.659 (0.01) | 0.594 (0.08) | 0.512 (0.01) | 0.515 (0.01) | 0.525 (0.02) | 0.502 (0.03) | **0.452** (0.01) | 0.482 (0.01) |
| WINE | RMSE ↓ | 0.725 (0.02) | 0.723 (0.02) | 0.733 (0.02) | 0.724 (0.02) | 0.704 (0.02) | 0.730 (0.02) | 0.720 (0.02) | 0.702 (0.03) | **0.635** (0.03) | 0.646 (0.03) |
| ONLINE | RMSE ↓ | **1.111** (0.25) | 1.121 (0.27) | 1.350 (0.57) | 1.187 (0.25) | **1.111** (0.25) | 1.137 (0.26) | 1.313 (0.46) | 1.179 (0.21) | 1.122 (0.26) | 1.112 (0.27) |
| ABALONE | RMSE ↓ | 2.135 (0.09) | 2.141 (0.09) | 2.171 (0.08) | 2.167 (0.09) | 2.087 (0.08) | 2.100 (0.10) | 2.088 (0.08) | 2.088 (0.08) | 2.164 (0.09) | **2.086** (0.09) |
| FICO | AUROC ↑ | 0.799 (0.007) | 0.795 (0.009) | 0.788 (0.006) | 0.797 (0.006) | **0.800** (0.007) | 0.793 (0.007) | 0.799 (0.007) | 0.799 (0.008) | 0.796 (0.008) | 0.795 (0.008) |
| CHURN | AUROC ↑ | 0.839 (0.012) | 0.824 (0.012) | **0.846** (0.011) | 0.845 (0.012) | 0.842 (0.012) | 0.830 (0.011) | 0.844 (0.011) | 0.844 (0.011) | **0.846** (0.012) | 0.844 (0.013) |
| CREDIT | AUROC ↑ | 0.983 (0.005) | 0.983 (0.005) | 0.976 (0.012) | 0.972 (0.011) | 0.984 (0.006) | **0.985** (0.006) | 0.980 (0.007) | **0.985** (0.004) | 0.983 (0.004) | 0.984 (0.009) |
| LETTER | AUROC ↑ | 0.900 (0.003) | 0.910 (0.002) | 0.904 (0.001) | 0.910 (0.001) | 0.984 (0.001) | 0.988 (0.001) | 0.986 (0.001) | 0.990 (0.001) | 0.997 (0.001) | **0.998** (0.001) |
| DRYBEAN | AUROC ↑ | 0.995 (0.001) | 0.996 (0.001) | 0.996 (0.001) | 0.994 (0.001) | **0.998** (0.001) | 0.996 (0.001) | 0.995 (0.001) | 0.995 (0.001) | 0.997 (0.001) | 0.996 (0.001) |
| | Rank avg ↓ | 6.33 | 5.56 | 8.00 | 7.44 | 3.33 | 5.67 | 5.44 | 3.67 | 3.44 | **2.89** |

## 4.4 APPLICATION TO HIGH-DIMENSIONAL DATA

To see whether ANOVA-NODE is applicable to high-dimensional data, we analyze three additional datasets with input dimensions ranging from 136 to 699. See Table 8 of Appendix for details of these three datasets. For ANOVA-N$^1$ODE, we include all main effects into the model. For ANOVA-N$^2$ODE, however, the number of second order interactions is too large so that considering all the main effects and second order interactions would be difficult unless very large computing resources are available. A simple alternative is to screen out unnecessary second order interactions a priori and include only selected second order interactions (and all the main effects) into the model. In the experiment, we use Neural Interaction Detection (NID, Tsang et al. (2017)) for the interaction screening. The number of selected interactions is given in Appendix C.2.

From Table 4 and Table 5, we observe that ANOVA-NODE shows favorable prediction performance compared with NAM and NBM and estimates the components more stably on high-dimensional datasets. In addition, note that the RMSE of NB$^2$M with all second order interactions on MICROSOFT dataset is reported as 0.750 by Radenovic et al. (2022). That is, screening interactions using NID does not hamper prediction performance much.

Table 4: **Prediction performance on high-dimensional datasets.** We report the averages (standard deviations) of the prediction performance for 10 randomly sampled training data from the high-dimensional datasets. The bold faces highlight the best results.

| Dataset | Measure | GAM | | | GA$^2$M | | |
|---|---|---|---|---|---|---|---|
| | | ANOVA N$^1$ODE | NA$^1$M | NB$^1$M | NID + ANOVA N$^2$ODE | NID + NA$^2$M | NID + NB$^2$M |
| MICROSOFT | RMSE ↓ | 0.756 (0.001) | 0.774 (0.001) | 0.770 (0.001) | **0.754** (0.001) | 0.761 (0.001) | 0.755 (0.001) |
| YAHOO | RMSE ↓ | 0.787 (0.002) | 0.797 (0.002) | 0.783 (0.002) | **0.779** (0.001) | 0.793 (0.002) | **0.779** (0.002) |
| MADELON | AUROC ↑ | 0.587 (0.02) | 0.587 (0.02) | 0.582 (0.03) | **0.605** (0.01) | 0.568 (0.03) | 0.594 (0.02) |

## 4.5 ANOVA-NODE WITH MONOTONE CONSTRAINTS

**Monotone constraint.** In practice, prior knowledge that some main effects are monotone functions are available and it is needed to reflect this prior knowledge in the training phase. A notable example is the credit scoring model where certain input features should have monotone main effects (Chen & Li, 2014; Chen & Ye, 2022). An additional advantage of ANOVA-NODE is to accommodate the monotone constraints in the model easily. Suppose that $f_j$ is monotonically increasing.

Then, ANOVA-NODE can estimate $f_j$ monotonically increasingly by letting the $\theta_j$ in ANOVA-NODT $h_j(x_j|\phi_j)$ be less than or equal to 0. See Appendix B.1 for details.

**Application to Image data.** Monotone constraint helps avoiding unreasonable interpretation. To illustrate this advantage, we conduct an experiment with an image dataset. We use CELEBA (Liu

Table 5: **Stability scores on the high-dimensional datasets.** For each dataset, stability scores of of GAM (GA$^2$M) models normalized by the that of ANOVA-N$^1$ODE (ANOVA-N$^2$ODE) are presented. Lower stability scores imply more stable interpretation. The bold faces highlight the best results.

| Dataset | GAM | | | GA$^2$M | | |
|---|---|---|---|---|---|---|
| | ANOVA N$^1$ODE | NA$^1$M | NB$^1$M | NID + ANOVA N$^2$ODE | NID + NA$^2$M | NID + NB$^2$M |
| MICROSOFT (Qin & Liu, 2013) | **1.000** | 2.967 | 3.933 | **1.000** | 2.075 | 2.225 |
| YAHOO (Yahoo, 2010) | **1.000** | 1.760 | 2.520 | **1.000** | 1.834 | 1.514 |
| MADELON (Guyon, 2004) | **1.000** | 1.957 | 2.014 | **1.000** | 1.184 | 1.132 |

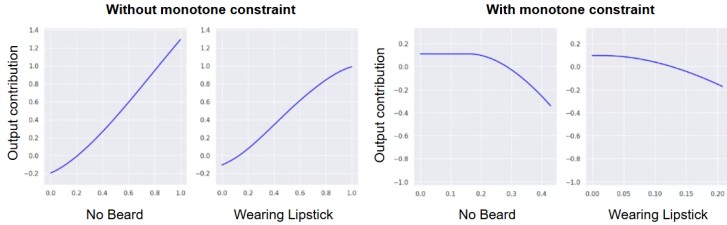

Figure 4: **Plots of the functional relations of 'No Beard' and 'Wearing Lipstick' on CELEBA dataset estimated by ANOVA-N$^1$ODE with and without the monotone constraint.**

et al., 2015) dataset which has 40 binary attributes for each image. To apply ANOVA-NODE to CELEBA dataset, we consider Concept Bottleneck Model (CBM, Koh et al. (2020)) similar to the one used in Radenovic et al. (2022). In CBM, rather than directly inputting the embedding vector derived from an image data through a CNN into a classifier, the CNN initially predicts each concept accompanied with each image. Then, these predicted values of each concept are subsequently used as the input of a DNN classifier. For our experiment, we use a pretrained ResNet18, where the last layer consists of a linear transformation with a softmax activation function, and we replace the final DNN classifier with ANOVA-NODE.

Among the attributes, we set 'gender' as the target label and the remaining attributes are set as concepts for images. Since 'male' is labeled as 1 and 'female' as 0, a higher value of each component results in a higher chance of being classified as 'male'.

Figure 4 presents two functional relations of the main effects of the concepts 'No Beard' and 'Wearing Lipstick', estimated on a randomly sampled training dataset with and without the monotone constraint. Note that the functional relations are quite different even though their prediction performances, which is given in Table 15 of Appendix E.1.2, are similar. It is a common sense that an image having the concept of 'No Beard' and 'Wearing Lipstick' has a higher chance of being a female and thus the functional relations are expected to be decrease. Figure 4 illustrates that a completely opposite result to our common sense could be obtained in practice. Implications of the opposite functional relations to interpretation of each image are discusses in Appendix E.1.2.

### 4.6 ADDITIONAL EXPERIMENTS

In Appendix L, we confirm that the component estimation of ANOVA-N$^2$ODE becomes highly unstable when the sum-to-zero condition is not imposed, and in Appendix M, we discuss a method for enforcing the sum-to-zero condition after training NAM or NBM.

### 5 CONCLUSION

In this paper, we propose a novel XAI model called ANOVA-NODE for estimating the functional ANOVA model stably based on Neural Oblivious Decision Tree (NODT). We theoretically demonstrate that ANOVA-NODE can approximate a smooth function well. We also empirically show that prediction performance of ANOVA-NODE is comparable to its competitors.

One way to make ANOVA-NODE computationally more efficient is to combine ANOVA-NODE and NBM (Radenovic et al., 2022), which has significantly fewer weight parameters. We explored this approach, and the algorithm and results of the experiment are given in Appendix G.2. Even though this algorithm is computationally more efficient than ANOVA-NODE itself, analyzing high-dimensional data is still computationally demanding due to too many components. In general, scaling-up interpretable AI algorithms is a promising future research topic.

**Reproducibility Statement.** The complete proofs of the theoretical results are presented thoroughly and rigorously in Appendix A. Details regarding the experimental implementation, datasets, libraries, and hyper-parameters are outlined in Appendix C. Furthermore, the proposed ANOVA-NODE in this paper is an ensemble of ANOVA-NODT, and Appendix B provides a detailed explanation of ANOVA-NODT. Therefore, by referring to Appendix B, ANOVA-NODT can be implemented for each component $S$, and the ensemble ANOVA-NODE can also be easily implemented. Finally, We will provide the source code and make it open access by uploading it on the web after acceptance.

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

# Supplementary material

# Appendix

## A    PROOFS FOR THEORETICAL RESULTS

### A.1    PROOF OF PROPOSITION 3.1

For a given function $f$, consider two component functions sets $\{f_S^1, S \in \mathbf{S}\}$ and $\{f_S^2, S \in \mathbf{S}\}$ such that

$$f(\mathbf{x}) = \sum_{S \in \mathbf{S}} f_S^1(\mathbf{x}_S) = \sum_{S \in \mathbf{S}} f_S^2(\mathbf{x}_S)$$

for every $\mathbf{x} \in \mathcal{X}$, where each component function satisfies the sum-to-zero condition (1).

For any $S, V \in \mathbf{S}$ such that $S \neq V$, we have $S \backslash V \neq \emptyset$ or $V \backslash S \neq \emptyset$. Assume $S \backslash V \neq \emptyset$ without loss of generality. For $i_1, i_2 \in \{1, 2\}$ and $s \in S \backslash V$, we get

$$\int_{\mathcal{X}} f_S^{i_1}(\mathbf{X}_S) f_V^{i_2}(\mathbf{X}_V) d\Pi_{j=1}^p \mu_j$$

$$= \int_{\mathcal{X}_{[p] \backslash \{s\}}} \left[ \int_{\mathcal{X}_s} f_S^{i_1}\left(\mathbf{x}_{S \backslash \{j\}} = \mathbf{X}_{S \backslash \{j\}}, \mathbf{x}_S = \mathbf{X}_S\right) d\mu_s \right] f_V^{i_2}(\mathbf{X}_V) d\Pi_{j \neq s} \mu_j$$

$$= 0$$

by the sum-to-zero condition, and hence

$$\int_{\mathcal{X}} (f_S^1(\mathbf{X}_S) - f_S^2(\mathbf{X}_S))(f_V^1(\mathbf{X}_V) - f_V^2(\mathbf{X}_V)) d\Pi_{j=1}^p \mu_j = 0.$$

Then, we obtain

$$\sum_{S \in \mathbf{S}} \int_{\mathcal{X}} (f_S^1(\mathbf{X}_S) - f_S^2(\mathbf{X}_S))^2 d\Pi_{j=1}^p \mu_j$$

$$= \sum_{S \in \mathbf{S}} \int_{\mathcal{X}} (f_S^1(\mathbf{X}_S) - f_S^2(\mathbf{X}_S))^2 d\Pi_{j=1}^p \mu_j$$

$$+ \sum_{S, V \in \mathbf{S}, S \neq V} \int_{\mathcal{X}} 2(f_S^1(\mathbf{X}_S) - f_S^2(\mathbf{X}_S))(f_V^1(\mathbf{X}_V) - f_V^2(\mathbf{X}_V)) d\Pi_{j=1}^p \mu_j$$

$$= \int_{\mathcal{X}} \left[ \sum_{S \in \mathbf{S}} \left(f_S^1(\mathbf{X}_S) - f_S^2(\mathbf{X}_S)\right) \right]^2 d\Pi_{j=1}^p \mu_j$$

$$= \int_{\mathcal{X}} [f(\mathbf{X}) - f(\mathbf{X})]^2 d\Pi_{j=1}^p \mu_j$$

$$= 0.$$

To sum up, we have $f_S^1(\cdot) \equiv f_S^2(\cdot)$ almost everywhere for $S \in \mathbf{S}$, which completes the proof.

$\square$

## A.2 PROOF OF THEOREM 3.3

**Challenging part of Theorem 3.3.** The most important and challenging part of Theorem 3.3 is decomposing the ensemble function $f_{\mathcal{E}}$, which approximates the true function well, into ANOVA-NODTs.

**Case of $d = 1$.** It is enough to show that for every $j \in [p]$, there exists a set of ANOVA-NODTs $\{h^S(\cdot | \phi_{j,k})\}_{k=1}^{K_S}$ such that

$$\left\| g_{0,j}(\cdot) - \sum_{k=1}^{K_S} h^S(\cdot | \phi_{j,k}) \right\|_{\infty} < \frac{\epsilon}{p}.$$

We denote $\sigma(x) := 1/(1 + \exp(-x))$ as the sigmoid function. We consider $\nu = 1$ for simplicity, which results in $entmax_{\nu}((x, 0)') = \sigma(x)$. Also, we assume that $\mu_j$ for $j \in [p]$ admits a density with respect to Lebesgue measure which is bounded above and below. In cases where the density does not exist, such as the empirical distribution, we can construct a $K$-equal-sized partition of $\mathcal{X}_j$ and then handle regions with zero measure and non-zero measure separately, similar to the proof described below.

Let $0 < p_L < p_R < \infty$ be the lower and upper bound of the density of $\mu_j$, respectively. Let $\{\Omega_k\}_{k=1}^K = \{[\chi_{k-1}, \chi_k)\}_{k=1}^K$ be a interval partition of $\mathcal{X}_j$ such that $\mu_j(\Omega_k) = \frac{1}{K}$. We have $|\chi_k - \chi_{k-1}| \leq \frac{1}{p_L K}$ for $k \in [K]$. For $\gamma = 1/K^3$, we define $\ell_k(\cdot)$ as

$$\ell_1(x) = 1 - \sigma\left(\frac{x - \chi_1}{\gamma}\right),$$

$$\ell_k(x) = \sigma\left(\frac{x - \chi_{k-1}}{\gamma}\right) - \sigma\left(\frac{x - \chi_k}{\gamma}\right), \qquad k \in \{2, \ldots, K-1\}$$

$$\ell_K(x) = \sigma\left(\frac{x - \chi_{K-1}}{\gamma}\right).$$

Note that for every $x \in \mathcal{X}_j$, $\sum_{k=1}^K \ell_k(x) = 1$ and $0 \leq \ell_k(\cdot) \leq 1$ holds for every $k \in [K]$. Also, $\{\ell_k\}_{k=1}^K$ have the following properties.

**Lemma A.1.** *For any $k \in [K]$, we have*

$$\mathbb{E}_{\mathcal{X}_j}[\ell_k(X_j)\mathbb{I}(X_j \notin \Omega_k)] < \frac{3p_U}{K^2}$$

*and*

$$\mathbb{E}_{\mathcal{X}_j}[\ell_k(X_j)\mathbb{I}(X_j \in \Omega_k)] > \frac{1}{3K}.$$

The proof of Lemma A.1 is provided in Section A.3. Now, we consider the ensemble function

$$f_{\mathcal{E}}(x) = \sum_{k=1}^K \delta_k \ell_k(x),$$

where $\delta_k$ is defined as

$$\delta_k = \frac{\mathbb{E}_{\mathcal{X}_j}[\ell_k(X_j)g_0(X_j)]}{\mathbb{E}_{\mathcal{X}_j}[\ell_k(X_j)]}.$$

For any $x \in \mathcal{X}_j$, we have

$$
\begin{aligned}
\left| g_0(x) - f_{\mathcal{E}}(x) \right| &= \left| g_0(x) - \sum_{k=1}^{K} \delta_k \ell_k(x) \right| \\
&= \left| \sum_{k=1}^{K} (g_0(x) - \delta_k) \ell_k(x) \right| \\
&\leq \sum_{k=1}^{K} |g_0(x) - \delta_k| \ell_k(x) \\
&= \sum_{k=1}^{K} \left| g_0(x) - \frac{\mathbb{E}_{\mathcal{X}_j}[\ell_k(X_j) g_0(X_j)]}{\mathbb{E}_{\mathcal{X}_j}[\ell_k(X_j)]} \right| \ell_k(x) \\
&= \sum_{k=1}^{K} \left| \frac{\mathbb{E}_{\mathcal{X}_j}[\ell_k(X_j) g_0(x)] - \mathbb{E}_{\mathcal{X}_j}[g_0(X_j) \ell_k(X_j)]}{\mathbb{E}_{\mathcal{X}_j}[\ell_k(X_j)]} \right| \ell_k(x) \\
&\leq L \sum_{k=1}^{K} \left| \frac{\mathbb{E}_{\mathcal{X}_j}[\ell_k(X_j)|x - X_j|]}{\mathbb{E}_{\mathcal{X}_j}[\ell_k(X_j)]} \right| \ell_k(x). \quad (3)
\end{aligned}
$$

Let $r \in [K]$ is the index of partition such that $x \in [\chi_{r-1}, \chi_r)$. For $k \in \{r-1, r, r+1\}$, we have

$$
\begin{aligned}
\frac{\mathbb{E}_{\mathcal{X}_j}[\ell_k(X_j)|x - X_j|]}{\mathbb{E}_{\mathcal{X}_j}[\ell_k(X_j)]} &\leq \frac{\mathbb{E}_{\mathcal{X}_j}[\ell_k(X_j)|x - X_j|\mathbb{I}(X_j \in \Omega_k)]}{\mathbb{E}_{\mathcal{X}_j}[\ell_k(X_j)\mathbb{I}(X_j \in \Omega_k)]} + \frac{\mathbb{E}_{\mathcal{X}_j}[\ell_k(X_j)|x - X_j|\mathbb{I}(X_j \notin \Omega_k)]}{\mathbb{E}_{\mathcal{X}_j}[\ell_k(X_j)\mathbb{I}(X_j \in \Omega_k)]} \\
&\leq \frac{\mathbb{E}_{\mathcal{X}_j}[\ell_k(X_j)(\frac{2}{p_L K})\mathbb{I}(X_j \in \Omega_k)]}{\mathbb{E}_{\mathcal{X}_j}[\ell_k(X_j)\mathbb{I}(X_j \in \Omega_k)]} + \frac{2a \cdot \mathbb{E}_{\mathcal{X}_j}[\ell_k(X_j)\mathbb{I}(X_j \notin \Omega_k)]}{\mathbb{E}_{\mathcal{X}_j}[\ell_k(X_j)\mathbb{I}(X_j \in \Omega_k)]} \\
&\leq \frac{2}{p_L K} + \frac{12 a C_{max}}{K} \\
&\leq \frac{C'}{K},
\end{aligned}
$$

for some constant $C' > 0$, where the third inequality holds by Lemma A.1. For $k \leq r - 2$, we have $x \geq \chi_{r-1}$ and $\chi_k \leq \chi_{r-2}$ and hence

$$
\begin{aligned}
|\ell_k(x)| &\leq 1 - \sigma\left( \frac{x - \chi_k}{\gamma} \right) \\
&\leq 1 - \sigma\left( \frac{\chi_{r-1} - \chi_{r-2}}{\gamma} \right) \\
&< \frac{1}{1 + \exp(K)}.
\end{aligned}
$$

Also, for $k \geq r + 2$, we have $x \leq \chi_r$ and $\chi_{k-1} \geq \chi_{r+1}$ and hence

$$
\begin{aligned}
|\ell_k(x)| &\leq \sigma\left( \frac{x - \chi_{k-1}}{\gamma} \right) \\
&\leq \sigma\left( \frac{\chi_r - \chi_{r+1}}{\gamma} \right) \\
&< \frac{1}{1 + \exp(K)}.
\end{aligned}
$$

To sum up, we get

$$
\begin{aligned}
|g_0(x) - f_{\mathcal{E}}(x)| &\leq (3) \\
&\leq L\left( \frac{C'}{K} + \frac{K-1}{1 + \exp(K)} \right).
\end{aligned}
$$

Now, for $f_{\mathcal{E}}(x) = \sum_{k=1}^{K} \delta_k \ell_k(x)$, our goal is to show that there exists a set of ANOVA-NODTs $\{h^S(\,\cdot\,|\phi_{j,k})\}_{k=1}^{K_S}$ such that

$$\sum_{j=1}^{K_S} h^S(x|\phi_{S,j}) = f_{\mathcal{E}}(x)$$

for every $x \in \mathcal{X}_j$. To derive the result, we use the following lemma.

**Lemma A.2.** *For every $T \in \{2, \ldots, K\}$, we define*

$$\ell_{1,T}(x) = 1 - \sigma\left(\frac{x - \chi_1}{\gamma}\right),$$

$$\ell_{t,T}(x) = \sigma\left(\frac{x - \chi_{t-1}}{\gamma}\right) - \sigma\left(\frac{x - \chi_t}{\gamma}\right), \qquad t \in \{2, \ldots, T-1\}$$

$$\ell_{T,T}(x) = \sigma\left(\frac{x - \chi_{T-1}}{\gamma}\right).$$

*Then, for any given $T \in \{3, \ldots, K\}$ and for any $\rho_1, \ldots, \rho_T$ satisfying*

$$\mathbb{E}_{\mathcal{X}_j}\left[\sum_{t=1}^{T} \rho_t \ell_{t,T}(X_j)\right] = 0, \tag{4}$$

*there exist $\kappa_1, \ldots, \kappa_{T-1}$, $\eta$ and $\tau$ such that*

$$\sum_{t=1}^{T} \rho_t \ell_{t,T}(x) = \sum_{t=1}^{T-1} \kappa_t \ell_{t,T-1}(x) + \left[\eta \cdot c_1(x|\chi_{T-1},\gamma) + \tau \cdot c_2(x|\chi_{T-1},\gamma)\right], \tag{5}$$

$$\mathbb{E}_{\mathcal{X}_j}\left[\sum_{t=1}^{T-1} \kappa_t \ell_{t,T-1}(X_j)\right] = 0, \quad \mathbb{E}_{\mathcal{X}_j}\left[\eta \cdot c_1(X_j|\chi_{T-1},\gamma) + \tau \cdot c_2(X_j|\chi_{T-1},\gamma)\right] = 0. \tag{6}$$

The proof of Lemma A.2 is provided in Section A.3. Since

$$\mathbb{E}_{\mathcal{X}_j}[f_{\mathcal{E}}(X_j)] = \mathbb{E}_{\mathcal{X}_j}\left[\sum_{k=1}^{K} \delta_k \ell_k(X_j)\right]$$

$$= \sum_{k=1}^{K} \frac{\mathbb{E}_{\mathcal{X}_j}[\ell_k(X_j) g_0(X_j)]}{\mathbb{E}_{\mathcal{X}_j}[\ell_k(X_j)]} \mathbb{E}_{\mathcal{X}_j}[\ell_k(X_j)]$$

$$= \sum_{k=1}^{K} \mathbb{E}_{\mathcal{X}_j}[\ell_k(X_j) g_0(X_j)]$$

$$= \mathbb{E}_{\mathcal{X}_j}\left[\sum_{k=1}^{K} \ell_k(X_j) g_0(X_j)\right]$$

$$= \mathbb{E}_{\mathcal{X}_j}[g_0(X_j)]$$

$$= 0,$$

we can decompose $f_{\mathcal{E}}(x)$ using Lemma A.2 by numerical induction. Note that $\rho_1 \ell_{1,2}(\cdot) + \rho_2 \ell_{1,2}(\cdot)$ with $\mathbb{E}_{\mathcal{X}_j}[\rho_1 \ell_{2,2}(X_j) + \rho_2 \ell_{1,2}(X_j)] = 0$ is an ANOVA-NODT and for $k \in \{2, \ldots, K\}$, $\eta \cdot c_1(\cdot|\chi_{T-1},\gamma) + \tau \cdot c_2(\cdot|\chi_{T-1},\gamma)$ with $\mathbb{E}_{\mathcal{X}_j}[\eta \cdot c_1(X_j|\chi_{T-1},\gamma) + \tau \cdot c_2(X_j|\chi_{T-1},\gamma)]$ is also an ANOVA-NODT. Hence, we can find a set of ANOVA-NODTs $\{h^S(\,\cdot\,|\phi_{j,k})\}_{k=1}^{K_S}$ such that

$$\left\| g_{0,j} - \sum_{k=1}^{K_S} h^S(x|\phi_{j,k}) = f_{\mathcal{E}}(x) \right\|_{\infty} < L\left(\frac{C'}{K} + \frac{K-1}{1 + \exp(K)}\right).$$

By choosing sufficiently large $K$, we obtain the assertion.

$\square$

### A.3 PROOFS FOR AUXILIARY LEMMAS

*Lemma A.1.* For $x \leq \chi_{k-1} - \frac{1}{K^2}$, we have

$$
\begin{aligned}
|\ell_k(x)| \leq & \sigma\left(\frac{x - \chi_{k-1}}{\gamma}\right) \\
\leq & \sigma\left(-\frac{1}{K^2\gamma}\right) \\
= & \frac{1}{1 + \exp(K)}.
\end{aligned}
$$

Also, for $x \geq \chi_k + \frac{1}{K^2}$, we have

$$
\begin{aligned}
|\ell_k(x)| \leq & 1 - \sigma\left(\frac{x - \chi_k}{\gamma}\right) \\
\leq & 1 - \sigma\left(\frac{1}{K^2\gamma}\right) \\
= & \frac{1}{1 + \exp(K)}.
\end{aligned}
$$

Hence, we obtain

$$
\begin{aligned}
\mathbb{E}_{\mathcal{X}_j}[\ell_k(X_j)\mathbb{I}(X_j \notin \Omega_k)] \leq & \mathbb{P}\left(X_j \in \left(\chi_{k-1} - \frac{1}{K^2}, \chi_{k-1}\right) \cup \left(\chi_k, \chi_k + \frac{1}{K^2}\right)\right) + \frac{1}{1 + \exp(K)} \\
< & \frac{3p_U}{K^2}.
\end{aligned}
$$

Also, for $x \in \left[\chi_{k-1} + \frac{1}{K^2}, \chi_k - \frac{1}{K^2}\right]$, we have

$$
\begin{aligned}
\ell_k(x) \geq & \sigma\left(\frac{x - \chi_{k-1}}{\gamma}\right) - \sigma\left(\frac{x - \chi_k}{\gamma}\right) \\
\geq & \sigma(K) - \sigma(-K) \\
> & \frac{1}{2}
\end{aligned}
$$

for sufficiently large $K$. Hence, we obtain

$$
\begin{aligned}
\mathbb{E}_{\mathcal{X}_j}[\ell_k(X_j)\mathbb{I}(X_j \in \Omega_k)] > & \mathbb{P}\left(X_j \in \left[\chi_{k-1} + \frac{1}{K^2}, \chi_k - \frac{1}{K^2}\right]\right) \cdot \frac{1}{2} \\
> & \frac{1}{3K}.
\end{aligned}
$$

$\square$

*Lemma A.2.* We define

$$
\begin{aligned}
\eta := & -\mathbb{E}_{\mathcal{X}_j}[\ell_{T,T}(X_j)](\rho_T - \rho_{T-1}), \\
\tau := & \sum_{t=1}^{T-1} \mathbb{E}_{\mathcal{X}_j}[\ell_{t,T}(X_j)](\rho_T - \rho_{T-1}), \\
\kappa_t := & \rho_t - \eta, \qquad\qquad\qquad\qquad\qquad t \in [T-1].
\end{aligned}
$$

Then, we obtain (5) by

$$\sum_{t=1}^{T-1} \kappa_t \ell_{t,T-1}(x) + [\eta \cdot c_1(x|\chi_{T-1}, \gamma) + \tau \cdot c_2(x|\chi_{T-1}, \gamma)]$$

$$= \sum_{t=1}^{T-2} (\rho_t - \eta)\ell_{t,T}(x) + (\rho_{T-1} - \eta) \cdot \sigma\left(\frac{x - \chi_{T-2}}{\gamma}\right) + (\tau - \eta) \cdot \sigma\left(\frac{x - \chi_{T-1}}{\gamma}\right) + \eta$$

$$= \sum_{t=1}^{T-2} (\rho_t - \eta)\ell_{t,T}(x) + (\rho_{T-1} - \eta) \cdot \left(\sigma\left(\frac{x - \chi_{T-2}}{\gamma}\right) - \sigma\left(\frac{x - \chi_{T-1}}{\gamma}\right)\right)$$

$$\quad + (\rho_{T-1} - \eta) \cdot \sigma\left(\frac{x - \chi_{T-1}}{\gamma}\right) + (\rho_T - \rho_{T-1}) \cdot \sigma\left(\frac{x - \chi_{T-1}}{\gamma}\right) + \eta$$

$$= \sum_{t=1}^{T-1} (\rho_t - \eta)\ell_{t,T}(x) + (\rho_T - \eta) \cdot \sigma\left(\frac{x - \chi_{T-1}}{\gamma}\right) + \eta$$

$$= \sum_{t=1}^{T} (\rho_t - \eta)\ell_{t,T}(x) + \eta$$

$$= \sum_{t=1}^{T} \rho_t \ell_{t,T}(x),$$

where the second equality holds by

$$\tau - \eta = \sum_{t=1}^{T} \mathbb{E}_{\mathcal{X}_j}[c_t(X_j)](\rho_T - \rho_{T-1}) = \rho_T - \rho_{T-1}.$$

Also, since we have

$$\mathbb{E}_{\mathcal{X}_j}[\eta \cdot c_1(x|\chi_{T-1}, \gamma) + \tau \cdot c_2(x|\chi_{T-1}, \gamma)]$$
$$= \eta \cdot \left(1 - \mathbb{E}_{\mathcal{X}_j}[\ell_{T,T}(X_j)]\right) + \tau \cdot \mathbb{E}_{\mathcal{X}_j}[\ell_{T,T}(X_j)]$$
$$= \eta + \mathbb{E}_{\mathcal{X}_j}[\ell_{T,T}(X_j)](\tau - \eta)$$
$$= -\mathbb{E}_{\mathcal{X}_j}[\ell_{T,T}(X_j)](\rho_T - \rho_{T-1}) + \mathbb{E}_{\mathcal{X}_j}[\ell_{T,T}(X_j)](\rho_T - \rho_{T-1})$$
$$= 0,$$

we obtain (6) by (4) and (5). $\qquad\square$

**Case of** $d = 2$**.** The proof for the case of $d = 2$ is too complex to express by mathematical notation, so we explain the process of decomposing the ensemble function $f_{\mathcal{E}}$ into ANOVA-NODTs, which is the most important part of Theorem 3.3, using a top example. Consider the ensemble function $f_{\mathcal{E}}(x_1, x_2)$ which is defined as below, with 9 height parameters: We explain the process of decomposing $f_{\mathcal{E}}$ into 4 ANOVA-NODTs in 2 steps.

$$
\begin{aligned}
f_{\mathcal{E}}(x_1, x_2) = {} & \beta_{11}\left(1 - \sigma\left(\frac{x_1 - b_{11}}{\gamma_{11}}\right)\right)\left(1 - \sigma\left(\frac{x_2 - b_{12}}{\gamma_{12}}\right)\right) \\
& + \beta_{21}\left(\sigma\left(\frac{x_1 - b_{11}}{\gamma_{11}}\right) - \sigma\left(\frac{x_1 - b_{21}}{\gamma_{21}}\right)\right)\left(1 - \sigma\left(\frac{x_2 - b_{12}}{\gamma_{12}}\right)\right) \\
& + \beta_{31}\sigma\left(\frac{x_1 - b_{21}}{\gamma_{21}}\right)\left(1 - \sigma\left(\frac{x_2 - b_{12}}{\gamma_{12}}\right)\right) \\
& + \beta_{12}\left(1 - \sigma\left(\frac{x_1 - b_{11}}{\gamma_{11}}\right)\right)\left(\sigma\left(\frac{x_2 - b_{12}}{\gamma_{12}}\right) - \sigma\left(\frac{x_2 - b_{22}}{\gamma_{22}}\right)\right) \\
& + \beta_{22}\left(\sigma\left(\frac{x_1 - b_{11}}{\gamma_{11}}\right) - \sigma\left(\frac{x_1 - b_{21}}{\gamma_{21}}\right)\right)\left(\sigma\left(\frac{x_2 - b_{12}}{\gamma_{12}}\right) - \sigma\left(\frac{x_2 - b_{22}}{\gamma_{22}}\right)\right) \\
& + \beta_{32}\sigma\left(\frac{x_1 - b_{21}}{\gamma_{21}}\right)\left(\sigma\left(\frac{x_2 - b_{12}}{\gamma_{12}}\right) - \sigma\left(\frac{x_2 - b_{22}}{\gamma_{22}}\right)\right) \\
& + \beta_{13}\left(1 - \sigma\left(\frac{x_1 - b_{11}}{\gamma_{11}}\right)\right)\sigma\left(\frac{x_2 - b_{22}}{\gamma_{22}}\right) \\
& + \beta_{23}\left(\sigma\left(\frac{x_1 - b_{11}}{\gamma_{11}}\right) - \sigma\left(\frac{x_1 - b_{21}}{\gamma_{21}}\right)\right)\sigma\left(\frac{x_2 - b_{22}}{\gamma_{22}}\right) \\
& + \beta_{33}\sigma\left(\frac{x_1 - b_{21}}{\gamma_{21}}\right)\sigma\left(\frac{x_2 - b_{22}}{\gamma_{22}}\right)
\end{aligned}
$$

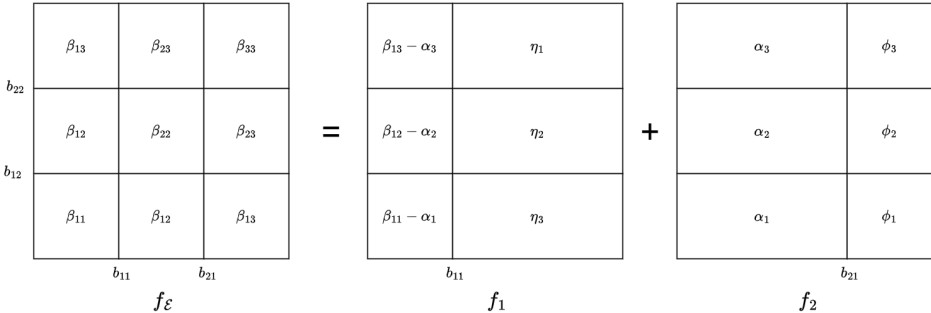

Figure 5: **Decomposition** $f_{\mathcal{E}}$ **into** $f_1$ **and** $f_2$. Each cell represents the support created by $\{b_{12}, b_{22}, b_{12}, b_{21}\}$ and $\beta_{ij}$ is the height parameter corresponding to each cell.

**Step 1) Decomposition of** $f_{\mathcal{E}}$**.** As shown in Figure 5, $f_{\mathcal{E}}$ can be decomposed into $f_1$ and $f_2$ which satisfy the sum-to-zero condition, and $f_1$ and $f_2$ are defined as

$$
\begin{aligned}
f_1(x_1, x_2) = {} & (\beta_{11} - \alpha_1)\left(1 - \sigma\left(\frac{x_1 - b_{11}}{\gamma_{11}}\right)\right)\left(1 - \sigma\left(\frac{x_2 - b_{12}}{\gamma_{12}}\right)\right) \\
& + \eta_1\sigma\left(\frac{x_1 - b_{11}}{\gamma_{11}}\right)\left(1 - \sigma\left(\frac{x_2 - b_{12}}{\gamma_{12}}\right)\right) \\
& + (\beta_{12} - \alpha_2)\left(1 - \sigma\left(\frac{x_1 - b_{11}}{\gamma_{11}}\right)\right)\left(\sigma\left(\frac{x_2 - b_{12}}{\gamma_{12}}\right) - \sigma\left(\frac{x_2 - b_{22}}{\gamma_{22}}\right)\right) \\
& + \eta_2\sigma\left(\frac{x_1 - b_{11}}{\gamma_{11}}\right)\left(\sigma\left(\frac{x_2 - b_{12}}{\gamma_{12}}\right) - \sigma\left(\frac{x_2 - b_{22}}{\gamma_{22}}\right)\right) \\
& + (\beta_{13} - \alpha_3)\left(1 - \sigma\left(\frac{x_1 - b_{11}}{\gamma_{11}}\right)\right)\sigma\left(\frac{x_2 - b_{22}}{\gamma_{22}}\right) \\
& + \eta_3\sigma\left(\frac{x_1 - b_{11}}{\gamma_{11}}\right)\sigma\left(\frac{x_2 - b_{22}}{\gamma_{22}}\right)
\end{aligned}
$$

and

$$f_2(x_1, x_2) = \alpha_1 \left( 1 - \sigma\left( \frac{x_1 - b_{21}}{\gamma_{21}} \right) \right) \left( 1 - \sigma\left( \frac{x_2 - b_{12}}{\gamma_{12}} \right) \right)$$

$$+ \phi_1 \sigma\left( \frac{x_1 - b_{21}}{\gamma_{21}} \right) \left( 1 - \sigma\left( \frac{x_2 - b_{12}}{\gamma_{12}} \right) \right)$$

$$+ \alpha_2 \left( 1 - \sigma\left( \frac{x_1 - b_{21}}{\gamma_{21}} \right) \right) \left( \sigma\left( \frac{x_2 - b_{12}}{\gamma_{12}} \right) - \sigma\left( \frac{x_2 - b_{22}}{\gamma_{22}} \right) \right)$$

$$+ \phi_2 \sigma\left( \frac{x_1 - b_{21}}{\gamma_{21}} \right) \left( \sigma\left( \frac{x_2 - b_{12}}{\gamma_{12}} \right) - \sigma\left( \frac{x_2 - b_{22}}{\gamma_{22}} \right) \right)$$

$$+ \alpha_3 \left( 1 - \sigma\left( \frac{x_1 - b_{21}}{\gamma_{21}} \right) \right) \sigma\left( \frac{x_2 - b_{22}}{\gamma_{22}} \right)$$

$$+ \phi_3 \sigma\left( \frac{x_1 - b_{21}}{\gamma_{21}} \right) \sigma\left( \frac{x_2 - b_{22}}{\gamma_{22}} \right)$$

where $\alpha_i = -(\beta_{3i} - \beta_{2i})\mathbb{E}\left[ \sigma\left( \frac{X_1 - b_{21}}{\gamma_{21}} \right) \right]$, $\phi_i = (\beta_{3i} - \beta_{2i})\mathbb{E}\left[ 1 - \sigma\left( \frac{X_1 - b_{21}}{\gamma_{21}} \right) \right]$ and $\eta_i = \beta_{2i} - \alpha_i$ for $i = 1, 2, 3$. It is easy to check that $f_1$ and $f_2$ satisfy the sum-to-zero condition.

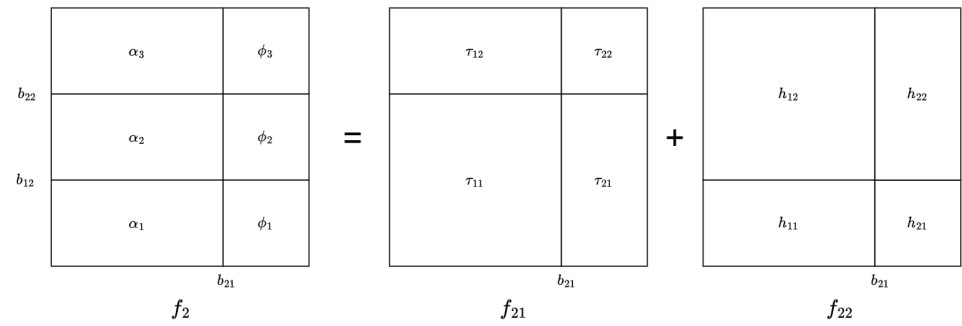

Figure 6: **Decomposition** $f_2$ **into** $f_{21}$ **and** $f_{22}$. Each cell represents the support created by $\{b_{12}, b_{22}, b_{21}\}$ and $\beta_{ij}$ is the height parameter corresponding to each cell.

**Step 2) Decomposition of** $f_2$. Similar to step 1, $f_2$ can be decomposed into $f_{22}$ and $f_{12}$, as described in Figure 6. Note that $f_{22}$ and $f_{12}$ are ANOVA-NODTs which are defined as

$$f_{21}(x_1, x_2) = \tau_{11} \left( 1 - \sigma\left( \frac{x_1 - b_{21}}{\gamma_{21}} \right) \right) \left( 1 - \sigma\left( \frac{x_2 - b_{22}}{\gamma_{22}} \right) \right)$$

$$+ \tau_{21} \sigma\left( \frac{x_1 - b_{21}}{\gamma_{21}} \right) \left( 1 - \sigma\left( \frac{x_2 - b_{22}}{\gamma_{22}} \right) \right)$$

$$+ \tau_{12} \left( 1 - \sigma\left( \frac{x_1 - b_{21}}{\gamma_{21}} \right) \right) \sigma\left( \frac{x_2 - b_{22}}{\gamma_{22}} \right)$$

$$+ \tau_{22} \sigma\left( \frac{x_1 - b_{21}}{\gamma_{21}} \right) \sigma\left( \frac{x_2 - b_{22}}{\gamma_{22}} \right)$$

where $\tau_{11} = -\mathbb{E}\left[ \sigma\left( \frac{X_2 - b_{22}}{\gamma_{22}} \right) \right] (\alpha_3 - \alpha_2)$ and $\tau_{12}, \tau_{21}, \tau_{22}$ are uniquely determined by sum-to-zero condition, and

$$f_{22}(x_1, x_2) = h_{11} \left( 1 - \sigma\left( \frac{x_1 - b_{21}}{\gamma_{21}} \right) \right) \left( 1 - \sigma\left( \frac{x_2 - b_{12}}{\gamma_{12}} \right) \right)$$

$$+ h_{21} \sigma\left( \frac{x_1 - b_{21}}{\gamma_{21}} \right) \left( 1 - \sigma\left( \frac{x_2 - b_{12}}{\gamma_{12}} \right) \right)$$

$$+ h_{12} \left( 1 - \sigma\left( \frac{x_1 - b_{21}}{\gamma_{21}} \right) \right) \sigma\left( \frac{x_2 - b_{12}}{\gamma_{12}} \right)$$

$$+ h_{22} \sigma\left( \frac{x_1 - b_{21}}{\gamma_{21}} \right) \sigma\left( \frac{x_2 - b_{12}}{\gamma_{12}} \right)$$

where $h_{11} = \alpha_1 - \tau_{11}$ and $h_{12}, h_{21}, h_{22}$ are uniquely determined by sum-to-zero condition. Also, another function $f_1$ can be decomposed into two ANOVA-NODTs in the same way as $f_2$ is decomposed.

**Case of** $d \geq 3$. Similar to the toy example at $d = 2$, the ensemble function $f_{\mathcal{E}}$ can also be decomposed into ANOVA-NODTs when $d \geq 3$.

# B  DETAILS FOR ANOVA-NODT

In this section, we describes the specific form of ANOVA-NODT $h^S$ for $S \subseteq [p]$.

## B.1  ANOVA-NODT FOR THE MAIN EFFECT.

Without loss of generality, we assumes that $S = \{i\}$. We can express ANOVA-NODT as:

$$h^i(x_i) = \beta_{1,i}c_1(x_i|b_{1,i}, \gamma_{1,i}) + \beta_{2,i}c_2(x_i|b_{1,i}, \gamma_{1,i}), \tag{7}$$

where $(\beta_{1,i}, \beta_{2,i})'$ are the height parameters of the terminal nodes,

$$c_1(x|b_{1,i}, \gamma_{1,i}) = 1 - entmax_\nu\left(\left(\frac{x - b_{1,i}}{\gamma_{1,i}}, 0\right)'\right)_1.$$

$$c_2(x|b_{1,i}, \gamma_{1,i}) = 1 - c_1(x|b_{1,i}, \gamma_{1,i})$$

and $b_{1,i}$ and $\gamma_{1,i}$ are learnable parameters. To satisfy the sum-to-zero condition, we require

$$\mathbb{E}[h^i(X_i)] = \beta_{1,i}\mathbb{E}[c_1(X_i|b_{1,i}, \gamma_{1,i})] + \beta_{2,i}\mathbb{E}[c_2(X_i|b_{1,i}, \gamma_{1,i})]$$
$$= 0$$

Without loss of generality, for a given $\theta_i \in \mathbb{R}$, let $\beta_{1,i} = \theta_i$. Then, we have $\beta_{2,i} = -I_{1,i}^{(b_{1,i}, \gamma_{1,i})}\theta_i$, where $I_{1,i}^{(b_{1,i}, \gamma_{1,i})} = -\frac{\mathbb{E}[c_1(X_i|b_{1,i}, \gamma_{1,i})]}{\mathbb{E}[c_2(X_i|b_{1,i}, \gamma_{1,i})]}$.

Therefore, the equation (7) is expressed as

$$h^i(x_i) = \theta_i c_1(x_i|b_{1,i}, \gamma_{1,i}) + \theta_i I_{1,i}^{(b_{1,i}, \gamma_{1,i})} c_2(x_i|b_{1,i}, \gamma_{1,i}),$$

where $\theta_i, b_{1,i}$ and $\gamma_{1,i}$ are learnable free parameters.

**Monotone constraint.**  $c_1(x_i|b_{1,i}, \gamma_{1,i})$ is a decreasing function with respect to $x_i$, and accordingly, $c_2(x_i|b_{1,i}, \gamma_{1,i})$ is an increasing function with respect to $x_i$. Therefore, if $\theta_i$ is greater than 0, then $h^i(x_i)$ becomes a decreasing function with respect to $x_i$ since $I_{1,i}^{(b_{1,i}, \gamma_{1,i})}$ is always less than 0.

## B.2  ANOVA-NODT FOR SECOND ORDER INTERACTION.

Without loss of generality, we consider the component $S = \{i, k\}$. For simplicity, we denote $c_h(x_i|b_{z,(i,k)}, \gamma_{z,(i,k)})$ as $c_{h,z}(x_i)$. Then, for the component $S$, ANOVA-NODT can be expressed as:

$$h^{(i,k)}(x_i, x_k) = \beta_{(1,1)',(i,k)}c_{1,1}(x_i)c_{1,2}(x_k)$$
$$+ \beta_{(1,2)',(i,k)}c_{1,1}(x_i)c_{2,2}(x_k)$$
$$+ \beta_{(2,1)',(i,k)}c_{2,1}(x_i)c_{1,2}(x_k)$$
$$+ \beta_{(2,2)',(i,k)}c_{2,1}(x_i)c_{2,2}(x_k),$$

where $(\beta_{(1,1)',(i,k)}, \beta_{(1,2)',(i,k)}, \beta_{(2,1)',(i,k)}, \beta_{(2,2)',(i,k)})$ are the height parameter vector of the terminal nodes and

$$c_{1,t}(x) = 1 - entmax_\nu\left(\left(\frac{x - b_{t,(i,k)}}{\gamma_{t,(i,k)}}, 0\right)'\right)_1$$

$$c_{2,t}(x) = 1 - c_{1,t}(x)$$

for $t = 1, 2$. To satisfy sum-to-zero condition, $h^{i,k}$ has to meet the followings: for $x_k \in \mathcal{X}_k$,

$$\mathbb{E}[h^{(i,k)}(X_i, x_k)] = \beta_{(1,1)',(i,k)}c_{1,2}(x_k)\mathbb{E}[c_{1,1}(X_i)]$$
$$+ \beta_{(1,2)',(i,k)}c_{2,2}(x_k)\mathbb{E}[c_{1,1}(X_i)]$$
$$+ \beta_{(2,1)',(i,k)}c_{1,2}(x_k)\mathbb{E}[c_{2,1}(X_i)]$$
$$+ \beta_{(2,2)',(i,k)}c_{2,2}(x_k)\mathbb{E}[c_{2,1}(X_i)]$$
$$= 0$$

and for $x_i \in \mathcal{X}_i$,

$$
\begin{aligned}
\mathbb{E}[h^{(i,k)}(x_i, X_k)] &= \beta_{(1,1)',(i,k)} c_{1,1}(x_i) \mathbb{E}[c_{1,2}(X_k)] \\
&\quad + \beta_{(1,2)',(i,k)} c_{1,1}(x_i) \mathbb{E}[c_{2,2}(X_k)] \\
&\quad + \beta_{(2,1)',(i,k)} c_{2,1}(x_i) \mathbb{E}[c_{1,2}(X_k)] \\
&\quad + \beta_{(2,2)',(i,k)} c_{2,1}(x_i) \mathbb{E}[c_{2,2}(X_k)] \\
&= 0.
\end{aligned}
$$

The following conditions are a rephrase of the above conditions:

$$
\begin{aligned}
\beta_{(1,1)',(i,k)} \mathbb{E}[c_{1,1}(X_i)] + \beta_{(2,1)',(i,k)} \mathbb{E}[c_{2,1}(X_i)] &= 0 \\
\beta_{(1,2)',(i,k)} \mathbb{E}[c_{1,1}(X_i)] + \beta_{(2,2)',(i,k)} \mathbb{E}[c_{2,1}(X_i)] &= 0 \\
\beta_{(1,1)',(i,k)} \mathbb{E}[c_{1,2}(X_k)] + \beta_{(1,2)',(i,k)} \mathbb{E}[c_{2,2}(X_k)] &= 0 \\
\beta_{(2,1)',(i,k)} \mathbb{E}[c_{1,2}(X_k)] + \beta_{(2,2)',(i,k)} \mathbb{E}[c_{2,2}(X_k)] &= 0
\end{aligned}
$$

For $\theta_{i,k} \in \mathbb{R}$, let $\beta_{(1,1)',(i,k)} = \theta_{i,k}$. Then, ANOVA-NODT $h^{(i,k)}(x_i, x_k)$ can be expressed as

$$
\begin{aligned}
h^{(i,k)}(x_i, x_k) &= \theta_{i,k} c_{1,1}(x_i) c_{1,2}(x_k) \\
&\quad + \theta_{i,k} I_{1,(i,k)} c_{1,1}(x_i) c_{2,2}(x_k) \\
&\quad + \theta_{i,k} I_{2,(i,k)} c_{2,1}(x_i) c_{1,2}(x_k) \\
&\quad + \theta_{i,k} I_{3,(i,k)} c_{2,1}(x_i) c_{2,2}(x_k)
\end{aligned}
$$

where

$$
\begin{aligned}
I_{1,(i,k)} &= -\frac{\mathbb{E}[c_{1,2}(X_k)]}{\mathbb{E}[c_{2,2}(X_k)]} \\
I_{2,(i,k)} &= -\frac{\mathbb{E}[c_{1,1}(X_i)]}{\mathbb{E}[c_{2,1}(X_i)]} \\
I_{3,(i,k)} &= I_{1,(i,k)} I_{2,(i,k)}.
\end{aligned}
$$

and $\theta_{i,k}$ as well as $b_{1,(i,k)}, \gamma_{1,(i,k)}$ and $b_{2,(i,k)}, \gamma_{2,(i,k)}$ are learnable free parameters.

## C  DETAILS OF THE EXPERIMENTS

All models except XGB were trained via Adam optimizer. Likewise in Popov et al. (2019), we set $\nu = 1.5$ for $entmax_\nu$ in all the experiments. All experiments are run with RTX 3090, RTX 4090, and 24GB memory.

### C.1  DETAILS FOR SYNTHETIC DATASETS

Table 6: **Test suite of synthetic functions.**

| $f^{(1)}$ | $Y = 10X_1 + 10X_2 + 20(X_3 - 0.3)(X_3 - 0.6) + 20X_4 + 5X_5 + 10\sin(\pi X_1 X_2) + \epsilon$ |
|---|---|
| $f^{(2)}$ | $Y = \pi^{X_1 X_2}\sqrt{2X_3} - \sin^{-1}(X_4) + \log(X_3 + X_5) - \frac{X_9}{X_{10}}\sqrt{\frac{X_7}{X_8}} - X_2 X_7 + \epsilon$ |
| $f^{(3)}$ | $Y = \exp|X_1 - X_2| + |X_2 X_3| - X_3^{2|X_4|} + \log(X_4^2 + X_5^2 + X_7^2 + X_8^2) + X_9 + \frac{1}{1+X_{10}^2} + \epsilon$ |

Table 7: **Distribution of input features in synthetic functions.**

| $f^{(1)}$ | $X_1, X_2, X_3, X_4, X_5 \sim^{iid} U(0,1)$ |
|---|---|
| $f^{(2)}$ | $X_1, X_2, X_3, X_6, X_7, X_9 \sim^{iid} U(0,1)$ and $X_4, X_5, X_8, X_{10} \sim^{iid} U(0.6,1)$. |
| $f^{(3)}$ | $X_1, X_2, X_3, X_4, X_5, X_6, X_7, X_8, X_9, X_{10} \sim^{iid} U(-1,1)$ |

The synthetic function $f^{(1)}$ is a slightly modified version of Friedman's synthetic function used in Chipman et al. (2010). $f^{(2)}$ and $f^{(3)}$ are taken from the synthetic functions used in the interaction detection experiments in Tsang et al. (2017). We generate 15K data samples from the distribution in the Table 7 and functions in the Table 6. Also, we divide them into train, validation and test datasets with ratio 0.7, 0.1 and 0.2, respectively. For all of the synthetic functions, the number of trees for component $S$, $K_S$, is set to 30.

### C.2  DETAILS OF THE EXPERIMENTS WITH REAL DATASETS.

Table 8: **Descriptions of real datasets.**

| Dataset | Size | Number of features | Problem | Number of Class |
|---|---|---|---|---|
| CALHOUSING | 21k | 8 | Regression | - |
| WINE | 4k | 11 | Regression | - |
| ONLINE | 40k | 58 | Regression | - |
| ABALONE | 4k | 10 | Regression | - |
| FICO | 10k | 23 | Classification | 2 |
| CHURN | 7k | 39 | Classification | 2 |
| CREDIT | 284k | 30 | Classification | 2 |
| LETTER | 20k | 16 | Classification | 2 |
| DRYBEAN | 13k | 16 | Classification | 7 |
| MICROSOFT | 960k | 136 | Regression | - |
| YAHOO | 700k | 699 | Regression | - |
| MADELON | 2.6k | 500 | Classification | 2 |
| CELEBA | 200K | | Classification | 2 |

Table 9: $K_S$ in **ANOVA-N$^1$ODE and ANOVA-N$^2$ODE**

| Dataset | ANOVA-N$^1$ODE | ANOVA-N$^2$ODE |
|---|---|---|
| CALHOUSING | 10 | 10 |
| WINE | 100 | 10 |
| ONLINE | 10 | 10 |
| ABALONE | 50 | 10 |
| FICO | 30 | 30 |
| CHURN | 10 | 10 |
| CREDIT | 10 | 5 |
| LETTER | 50 | 10 |
| DRYBEAN | 50 | 100 |
| MICROSOFT | 10 | 10 |
| YAHOO | 10 | 10 |
| MADELON | 10 | 40 |

**Implementation of baseline model.** We conduct experiments for all baseline models (NAM, NBM, NODE-GAM, NODE) using the official source code. For XGB, we utilize the xgboost package (Chen & Guestrin, 2016) for our experiments.

**Data descriptions.** Table 8 summarizes the descriptions of 9 real datasets we analyze in the numerical studies.

**Data preprocessing.** Minimax scaling is applied to NAM and NBM, while standard scaling was used for NODE-GAM, NODE, and XGB. For ANOVA-NODE, transformation using quantiles of a uniform distribution is performed to satisfy sum-to-zero condition stably during training. Additionally, all categorical features is encoded using one-hot encoding.

**Learning rate.** For all models except XGB, we set the learning rate of Adam optimizer as 5e-3 and batch size is 4096. We find the optimal learning rate of XGB via grid search.

**Model hyperparameters.** Table 9 presents the number of NODTs $K_S$ used in ANOVA-N$^1$ODE and ANOVA-N$^2$ODE for real datasets. In NA$^1$M, the dimensions of the hidden layers of each component consists of [64,32,16] for MICROSOFT, YAHOO and MADELON, and [64,64,32] for other datasets. In N$A^2$M, the hidden layers consist of [64,32,16] for the ONLINE, CREDIT and DRYBEAN datasets, [64,16,8] for MICROSOFT, YAHOO and MADELON, and [64,64,32] for the other datasets.

For XGB, NODE, and NODE-GAM, we randomly split the train, validation and test data into the ratio 70/10/20 and evaluated its performance on the validation dataset using the model trained on the train dataset. We repeated this process 10 times with randomly split data, resulting in 10 prediction performance values for the validation datasets. Then, we selected the optimal hyper-parameters by the grid search based on the average of the prediction performance values for the validation datasets.

Finally, with the optimal hyper-parameters selected earlier, we fixed the model's hyper-parameters and used the 10 train-test dataset pairs obtained from the previous data splitting to train the model on the train datasets and evaluate its performance on the test datasets.

For XGB, the range of hyper-parameters for the grid search is as below.

- The number of tree : {50,100,200,300,400,500,600,700,800,900,1000}
- max depth : {3 , 5 , 7}
- learning rate : {0.0001, 0.005, 0.01, 0.05 , 0.1}

For NODE, NODE-GA$^1$M and NODE-GA$^2$M, the range of hyper-parameters for the grid search is as below.

- The number of layer : {2, 4, 8}
- tree depth : {6, 8}
- The number of trees in each layer : {256, 512}

**Selected components by NID for high-dimensional datasets.** Table 10 presents the number of components used in training ANOVA-$N^2$ODE and baseline models. All main effects are used, and the second order interactions are selected using NID (Tsang et al., 2017). For MICROSOFT, 300 second order interactions are used; 500 for YAHOO, 500; and 300 for MADELON.

Table 10: **The number of components used in training ANOVA-$N^2$ODE, NA$^2$M, and NB$^2$M.**

| Dataset | MICROSOFT | YAHOO | MADELON |
|---|---|---|---|
| # of selected components | 136(Main) + 300(2nd) | 699(Main) + 500(2nd) | 500(Main) + 300(2nd) |

## C.3 EXPERIMENT DETAILS FOR IMAGE DATASET.

For CELEBA image dataset, we use the joint bottleneck model in Koh et al. (2020). The main idea of the joint bottleneck model (Koh et al., 2020) is not to directly input the embedding vector derived from image data through a CNN into a classifier for classification. Instead, CNN predicts given concepts (attributes) for the image, and the predicted values for these concepts are then used as an input of classifier. In Koh et al. (2020), they used DNN as a classifier which is a black box model. In this paper, we replace DNN with ANOVA-NODE, NAM, NBM and NODE-GAM. For CNN, we linear heads on the bottom of the pretrained ResNet18.

All models are trained via the Adam optimizer with a 1e-3 learning rate and the batch size for training is 256. In ANOVA-$N^1$ODE and ANOVA-$N^2$ODE, $K_S$ is set to 10 and 3, respectively. In NA$^1$M and NA$^2$M, we use a neural network consisting of 3 hidden layer with sizes (64, 16,8) and (64,4,2), respectively. In NB$^1$M and NB$^2$M, we use 100 basis neural networks consisting of 3 hidden layer with sizes (256,128,128) and (128,64,64) for the basis model, respectively. Also, for NODE-GA$^1$M and NODE-GA$^2$M, the number of trees is set to 125 and 50, respectively, and the depth and the number of layers are set to be 6 and 4, respectively.

# D ABLATION STUDIES.

## D.1 THE CHOICE OF THE NUMBER OF TREES IN ANOVA-NODE.

Table 11 presents the averages and standard deviations of prediction performance of ANOVA-NODE based on 10 randomly sampled data of ABALONE datasets for various values of the number of trees $K_S$. We observe that $K_S$ around 50 yields the best results for ANOVA-N$^1$ODE and $K_S$ around 10 for ANOVA-N$^2$ODE. We report that similar results are obtained for other datasets.

Table 11: **Results of prediction performance for various numbers of trees.**

| The number of trees for each component | 1 | 5 | 10 | 50 | 100 |
|---|---|---|---|---|---|
| ANOVA-N$^1$ODE | 2.176 (0.09) | 2.163 (0.08) | 2.160 (0.09) | 2.135 (0.09) | 2.159 (0.08) |
| ANOVA-N$^2$ODE | 2.103 (0.08) | 2.102 (0.08) | 2.087 (0.08) | 2.105 (0.08) | 2.122 (0.08) |

## D.2 IMPACT OF THE INITIAL VALUES OF MODEL PARAMETERS TO STABILITY

We investigate how the choice of initial values of the model parameters affects the stability of the estimated components by ANOVA-NODE, NAM and NBM by analyzing a synthetic dataset generated from $f^{(1)}$.

We conducted 5 trials on the same train/test/validation dataset, and the results are presented in Figure 7 ,8 and 9. We observe that NA$^2$M and NB$^2$M frequently estimate the true function inaccurately. In contrast, ANOVA-N$^2$ODE consistently estimates the components accurately regardless of the choice of the initial values.

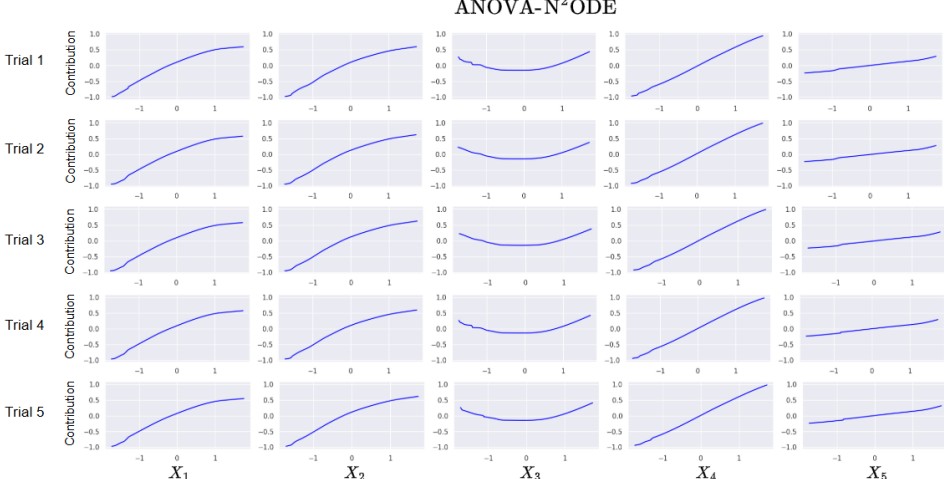

Figure 7: **Plots of the functional relations of the main effects in ANOVA-N$^2$ODE on synthetic datasets generated from $f^{(1)}$.**

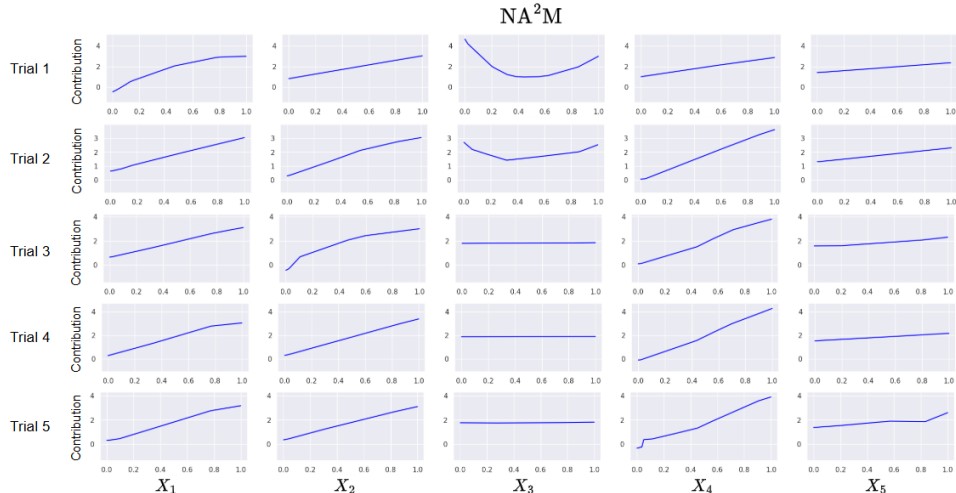

Figure 8: **Plots of the functional relations of the main effects in NA$^2$M on synthetic datasets generated from $f^{(1)}$.**

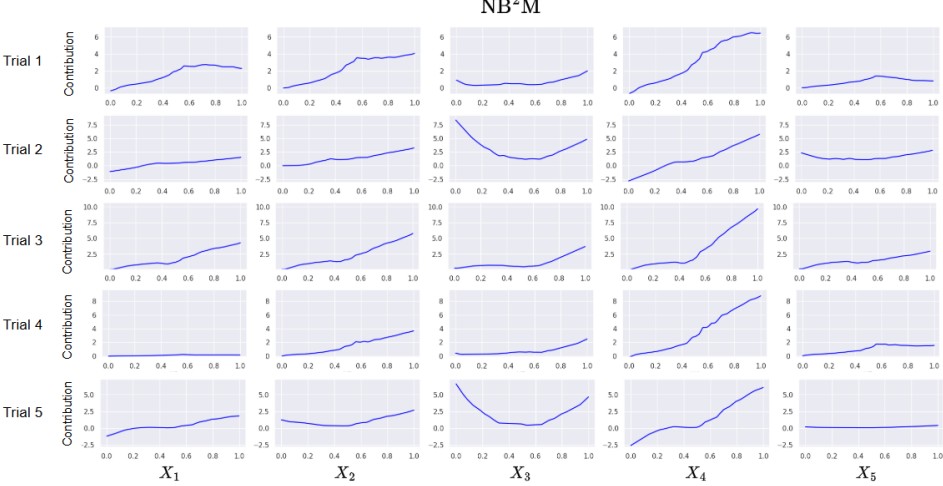

Figure 9: **Plots of the functional relations of the main effects in NB$^2$M on synthetic datasets generated from $f^{(1)}$.**

# E    ILLUSTRATION OF INTERPRETABILITY OF ANOVA-NODE.

## E.1    ILLUSTRATION OF INTERPRETABILITY

We consider the two concepts of interpretation: Local and Global which are roughly defined as:

**Local Interpretation:**    Information about how each feature of a given datum affects the prediction. SHAP is a notable example of local interpretation. For the functional ANOVA model, the predictive values of each component at a given datum would be considered as local interpretation.

**Global Interpretation:**    Information about how each feature is related to the final prediction model. The importance scores of each feature (e.g. global SHAP (Molnar, 2020)) and the functional relations between each feature and the prediction model (e.g. the dependency plot of SHAP (Molnar, 2020) are examples of global interpretation. For the functional ANOVA model, the importance score, which can be defined as the $l_1$ norm of the corresponding component as is done in Section 4.2, and the functional relation identified by the functional form of each component are two tools for global interpretation.

### E.1.1    ILLUSTRATION OF INTERPRETABILITY ON CALHOUSING DATASET.

Table 12: **Feature descriptions of CALHOUSING dataset.**

| Feature name | Index | Description | Feature type |
|---|---|---|---|
| MedInc | 1 | Median income in block | Numerical |
| HouseAge | 2 | Median house age in block | Numerical |
| AveRooms | 3 | Average number of rooms | Numerical |
| AveBedrms | 4 | Average number of bedrooms | Numerical |
| Population | 5 | Population in block | Numerical |
| AveOccup | 6 | Average house occupancy | Numerical |
| Latitude | 7 | Latitude of house block | Numerical |
| Longitude | 8 | Longitude of house block | Numerical |

**Local Interpretation on CALHOUSING dataset.**    We conduct an experiment on CALHOUSING (Pedregosa et al., 2011a) dataset to illustrate local interpretation of ANOVA-N$^1$ODE. Note that ANOVA-N$^1$ODE is given as

$$\hat{f}_{\text{ANOVA-N}^1\text{ODE}}(\mathbf{x}) = \sum_{j=1}^{8} \hat{f}_j(x_j).$$

Thus, it is reasonable to treat $\hat{f}_j(x_j)$ as the contribution of $x_j$ to $\hat{f}(\mathbf{x})$. In fact, we have seen in Section 3.1 that this contribution is equal to SHAP (Lundberg, 2017). As an illustration, for a given datum

$$\mathbf{x} = (-0.2378, -0.4450, 0.0036, -0.1531, 0.3814, -0.067, 0.5541, -0.1111)^\top,$$

the contributions of each feature to $\hat{f}(\mathbf{x})$ are

$$(\hat{f}_1, ..., \hat{f}_8) = (-4.9900, 0.3278, -0.0456, 0.4432, -0.1730, 2.7521, -11.6190, 6.5184).$$

That is, the 7th variable contributes most to the prediction value of $\hat{f}(\mathbf{x})$, which can be interpreted as 'the housing price is low because the latitude is not good'.

**Global Interpretation on CALHOUSING dataset.**    Figure 10 and Table 13 present the functional relations of each input feature to the prediction model learned by ANOVA-NO$^1$DE and their importance scores. From these results, we can see that the location is the most important features and the housing price on the south-west area is the most expensive.

Table 14 describes the 10 most important components with descending order of the importance scores of ANOVA-NO$^2$DE normalized by the maximum importance score. The results are bit different from those of ANOVA-NO$^1$DE. In particular, the interaction between 'latitude' and 'longitude' emerges as a new important feature while the main effects of 'latitude' and 'longitude' become less important.

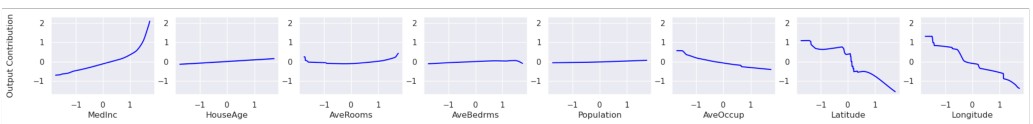

Figure 10: **Plots of the functional relations of the main effects in ANOVA-N$^1$ODE on CALHOUSING dataset.**

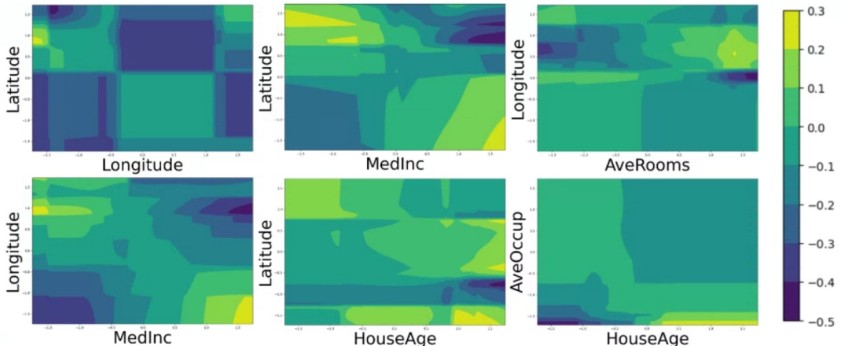

Figure 11: **Contour plots of the functional relations of the interactions in ANOVA-N$^2$ODE on CALHOUSING dataset.**

Table 13: **Importance scores of ANOVA-N$^1$ODE on CALHOUSING dataset.**

| Feature index | 7 | 8 | 1 | 6 | 3 | 2 | 4 | 5 |
|---|---|---|---|---|---|---|---|---|
| Importance score | 1.000 | 0.906 | 0.564 | 0.284 | 0.107 | 0.093 | 0.057 | 0.049 |

Table 14: **Importance scores of ANOVA-N$^2$ODE on CALHOUSING dataset.**

| Feature index | 6 | (7,8) | (1,7) | (3,8) | 7 | (1,8) | (4,8) | (2,7) | (1,5) | 8 |
|---|---|---|---|---|---|---|---|---|---|---|
| Importance score | 1.000 | 0.347 | 0.324 | 0.268 | 0.258 | 0.247 | 0.212 | 0.194 | 0.193 | 0.178 |

E.1.2 INTERPRETABILITY AND PREDICTION PERFORMANCE ON CELEBA DATASET.

**Prediction performance with and without the monotone constraint.** Table 15 presents the prediction performances of two estimates of ANOVA-NODE with and without monotone constraint. Prediction performances are similar regardless of the monotone constraint but interpretation of the estimated model can be quite different which is discussed in the followings.

**Global interpretation on CELEBA dataset.** Table 16 gives the the importance scores (normalized by of the maximum important score) of 3 components obtained by ANOVA-N$^1$ODE on a randomly sampled data from CELEBA dataset.

**Local interpretation on CELEBA dataset.** In Table 17, we observe that Image 2-1 of Figure 12 is correctly classified when the monotone constraint is applied, whereas it is misclassified without the monotone constraint. Despite Image 2-1 of Figure 12 having 'No Beard', 'Heavy Makeup', and 'Wearing Lipstick', the scores for these features makes the probability of male increase. However, ANOVA-N$^1$ODE with the monotone constraint does not provide these unreasonable interpretations and classifies the image correctly.

In Image 2-2 of Figure 12, we observe that ANOVA-N$^1$ODE with the monotone condition assigns a negative score to 'No Beard' that increases the probability of being classified as female. However, ANOVA-N$^1$ODE without the monotone condition assigns a positive score to 'No Beard' that increases the probability of being classified as male, even though 'No Beard' is present.

Note that we can understand why ANOVA-N$^1$ODE with the monotone constraint classifies Image 2-2 of Figure 12 incorrectly because there is no bear in the image. In contrast, it is not easy to understand why ANOVA-N$^1$ODE without the monotone constraint classifies Image 2-1 of Figure 12 incorrectly. That is, imposing the monotone constraint is helpful to learn more reasonably interpretable models.

Table 15: **Results of prediction performance of ANOVA-NODE with and without the monotone constraint.**

|  | Measure | ANOVA-N$^1$ODE | ANOVA-N$^2$ODE |
|---|---|---|---|
| Without Monotone constraint | Accuracy ↑ | 0.985 (0.001) | 0.986 (0.001) |
| With Monotone constraint | Accuracy ↑ | 0.984 (0.001) | 0.985 (0.001) |

**Comparison with baseline models in terms of prediction performance.** In Table 18, we observe that the prediction performances of ANOVA-N$^1$ODE and ANOVA-N$^2$ODE are comparable or superior to their competitors.

**Attributes to which monotone constraints are applied.** For attributes 'Bald', 'Big Nose', 'Goatee' and 'Mustache', we apply the increasing monotone constraint, while for attributes 'Arched Eyebrows', 'Attractive', 'Heavy Makeup', 'No Beard', 'Wearing Earrings', 'Wearing Lipstick', 'Wearing Necklace', 'Wearing Necktie', we used the decreasing monotone constraint.

Table 16: **Importance scores for the 3 important components.**

| Components | Monotone | No Beard | Wearing Lipstick | Heavy Makeup |
|---|---|---|---|---|
| Score | X | 0.794 | 0.465 | 0.210 |
| Score | O | 0.757 | 0.738 | 0.227 |

Table 17: **Results of local interpretation with and without the monotone constraint.**

| Image index | Monotone | Heavy Makeup | No beard | Wearing Lipstick | classified label | True label |
|---|---|---|---|---|---|---|
| 2-1 | X | 0.030 | 0.035 | 0.093 | male | female |
| 2-1 | O | -0.080 | -0.161 | -0.106 | female | female |
| 2-2 | X | 0.036 | 0.104 | 0.095 | male | male |
| 2-2 | O | -0.081 | -0.183 | -0.106 | female | male |

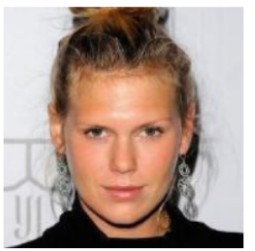 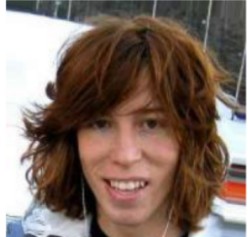

Image 2-1      Image 2-2

Figure 12: **Misclassified two images.**

Table 18: **Accuracies (standard deviations) on CELEBA dataset.**

| ANOVA-N$^1$ODE | NODE-GA$^1$M | NA$^1$M | NB$^1$M | ANOVA-N$^2$ODE | NODE-GA$^2$M | NA$^2$M | NB$^2$M |
|---|---|---|---|---|---|---|---|
| 0.985 (0.001) | 0.981 (0.006) | 0.982 (0.002) | 0.980 (0.002) | **0.986** (0.001) | 0.981 (0.006) | **0.986** (0.001) | 0.980 (0.002) |

# F ADDITIONAL EXPERIMENTS FOR THE STABILITY OF ANOVA-NODE.

## F.1 STABILITY OF THE ESTIMATED COMPONENTS ON VARIATIONS OF TRAINING DATA

We investigate the stability of components estimated by ANOVA-N$^1$ODE and ANOVA-N$^2$ODE when training data vary. We use CALHOUSING (Pedregosa et al., 2011a) and WINE (Cortez et al., 2009) datasets and compare ANOVA-NODE with NAM and NBM.

**Experiment for CALHOUSING dataset.** Figures 13, 14 and 15 present the plots of the functional relations of the main effects estimated by ANOVA-N$^1$ODE, NA$^1$M, and NB$^1$M for 5 randomly sampled training datasets. Figures 13, 14 and 15 present the plots of the functional relations of the main effects estimated by ANOVA-N$^2$ODE, NA$^2$M, and NB$^2$M for 5 randomly sampled training datasets. We observe that the 5 main components estimated by ANOVA-N$^1$ODE and ANOVA-N$^2$ODE are relatively much more stable compared to NAM and NBM. Note that as seen in Figure 17, we observe that in NA$^2$M, some components are estimated as a constant function, which would be partly because the main effects are absorbed into the second order interactions.

**Experiment for WINE dataset.** Figures 19, 20 and 21 present the plots of the functional relations of the main effects estimated by ANOVA-N$^1$ODE, NA$^1$M, and NB$^1$M for 5 randomly sampled training datasets. Figures 22, 23 and 24 present the plots of the functional relations of the main effects estimated by ANOVA-N$^2$ODE, NA$^2$M, and NB$^2$M for 5 randomly sampled training datasets. The results are similar to those of CALHOUSING dataset.

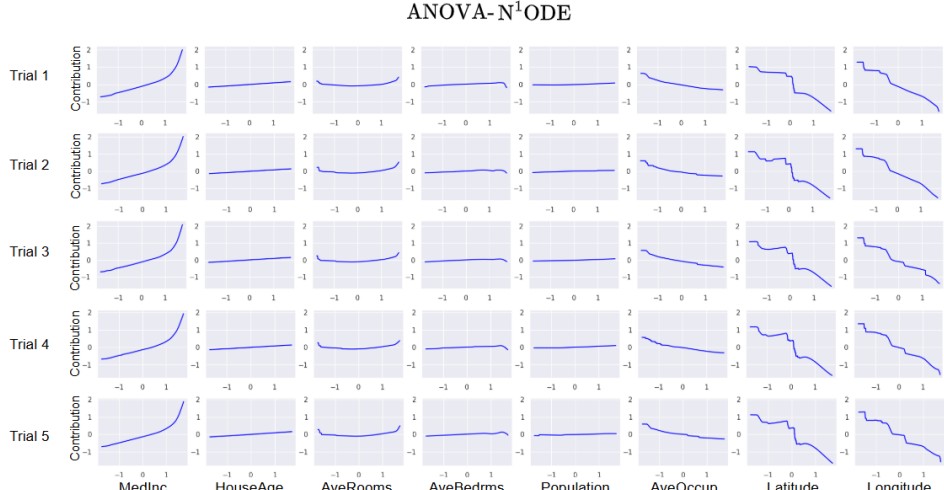

Figure 13: **Plots of the functional relations of the main effects in ANOVA-N$^1$ODE on CALHOUSING dataset.**

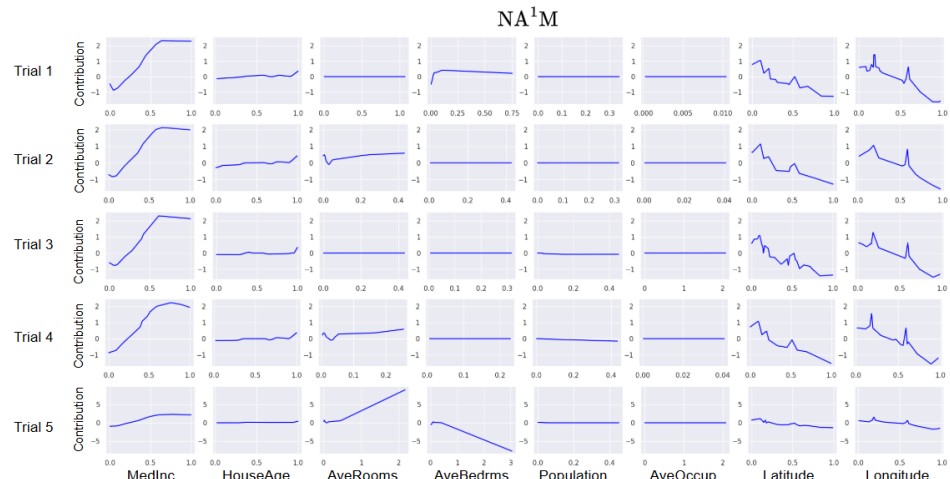

Figure 14: **Plots of the functional relations of the main effects in NA$^1$M on CALHOUSING dataset.**

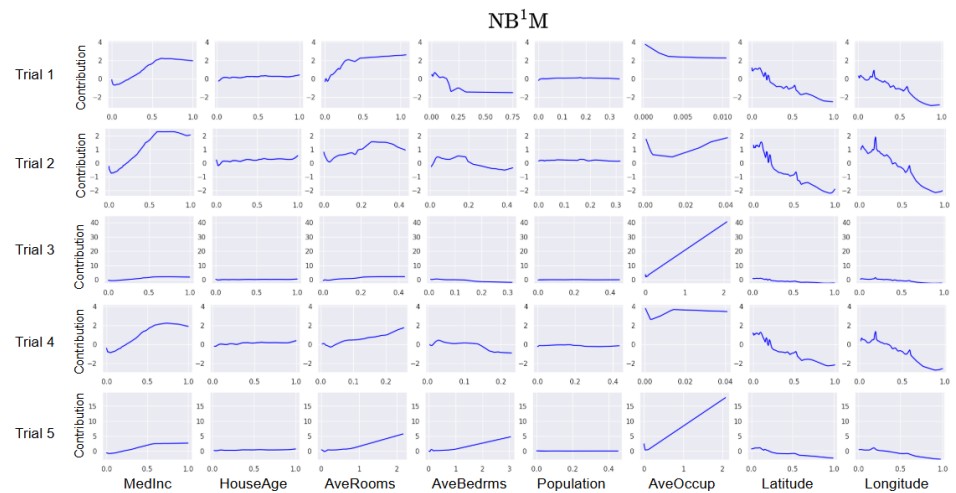

Figure 15: **Plots of the functional relations of the main effects in NB$^1$M on CALHOUSING dataset.**

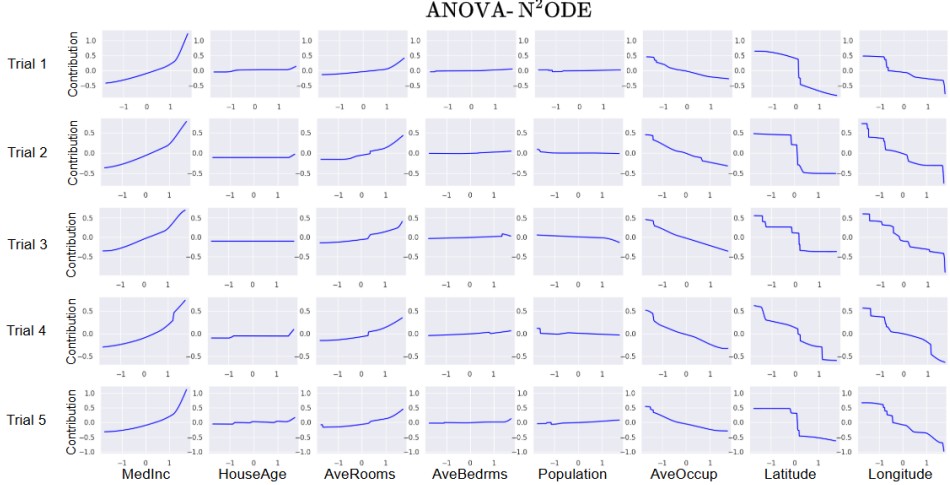

Figure 16: **Plots of the functional relations of the main effects in ANOVA-N$^2$ODE on CALHOUSING dataset.**

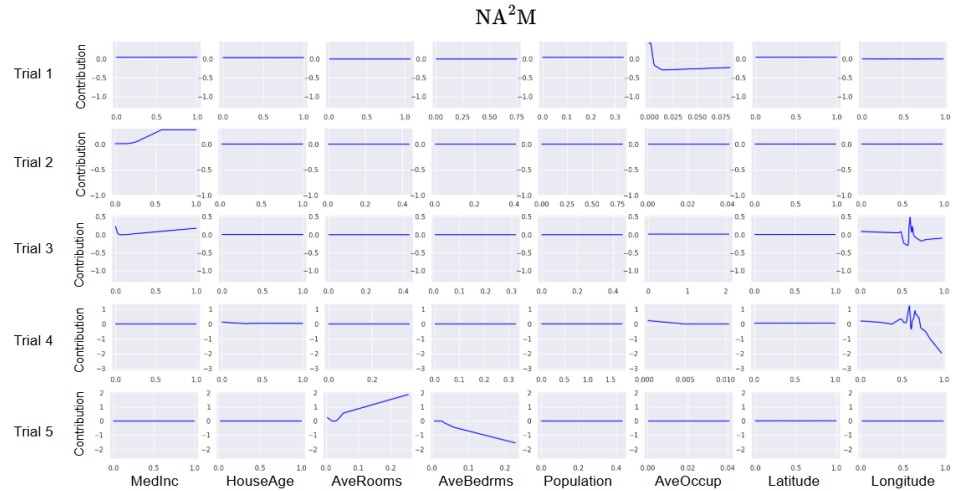

Figure 17: **Plots of the functional relations of the main effects in NA$^2$M on CALHOUSING dataset.**

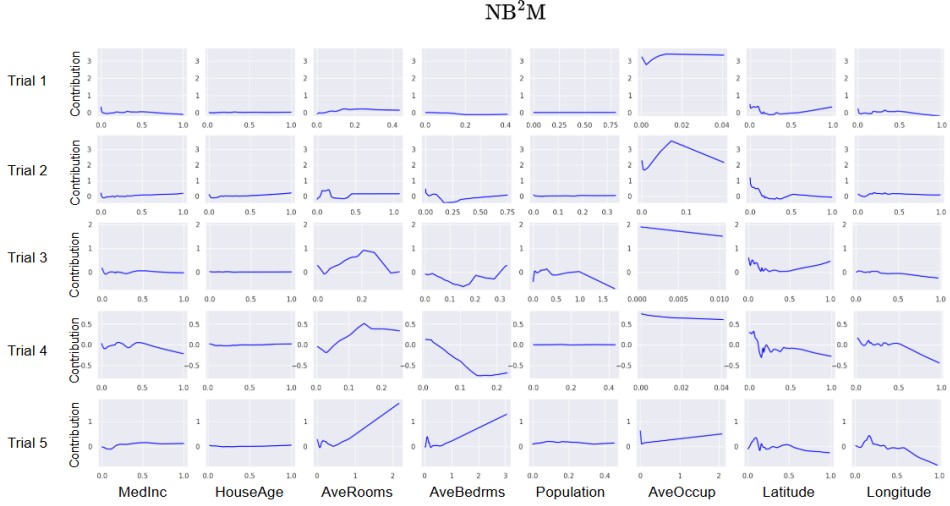

Figure 18: **Plots of the functional relations of the main effects in NB$^2$M on CALHOUSING dataset.**

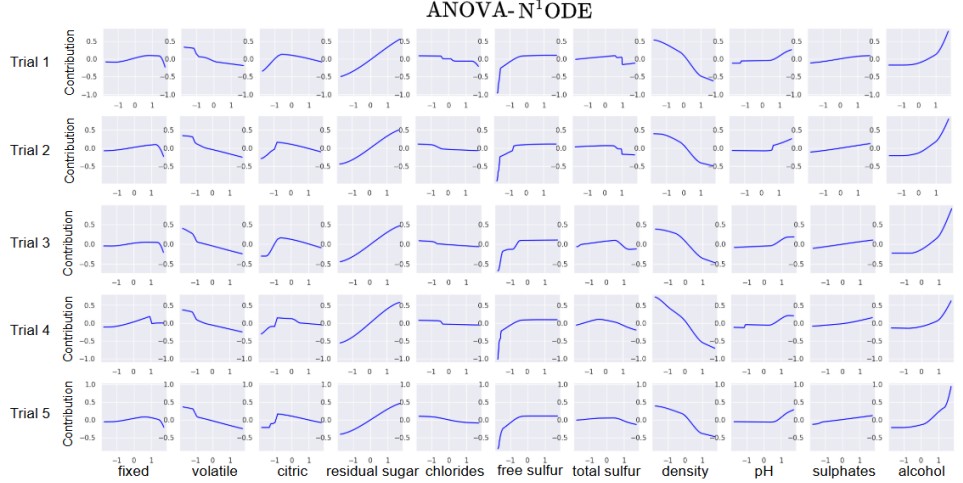

Figure 19: **Plots of the functional relations of the main effects in ANOVA-N$^1$ODE on WINE dataset.**

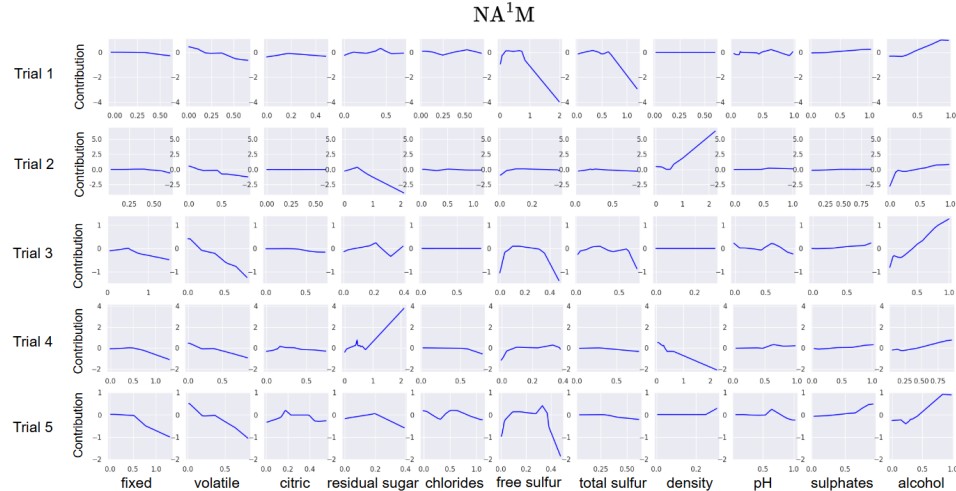

Figure 20: **Plots of the functional relations of the main effects in NA$^1$M on WINE dataset.**

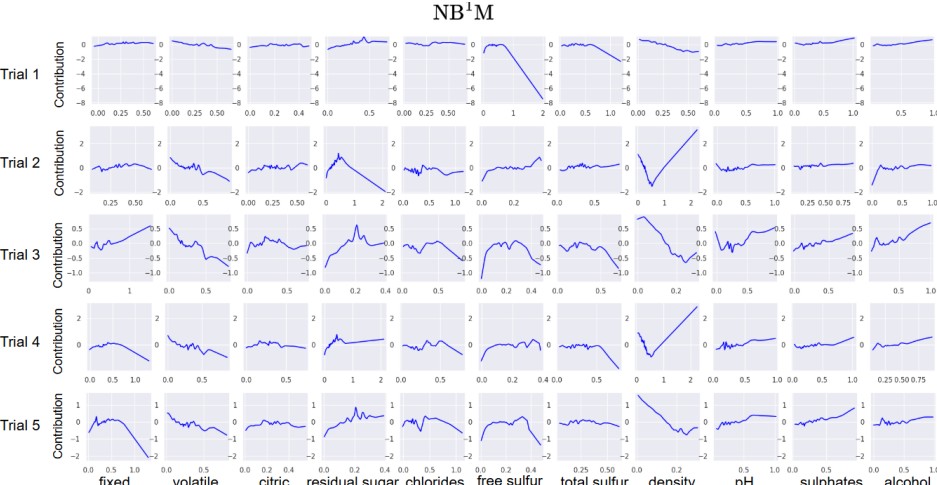

Figure 21: **Plots of the functional relations of the main effects in NB$^1$M on WINE dataset.**

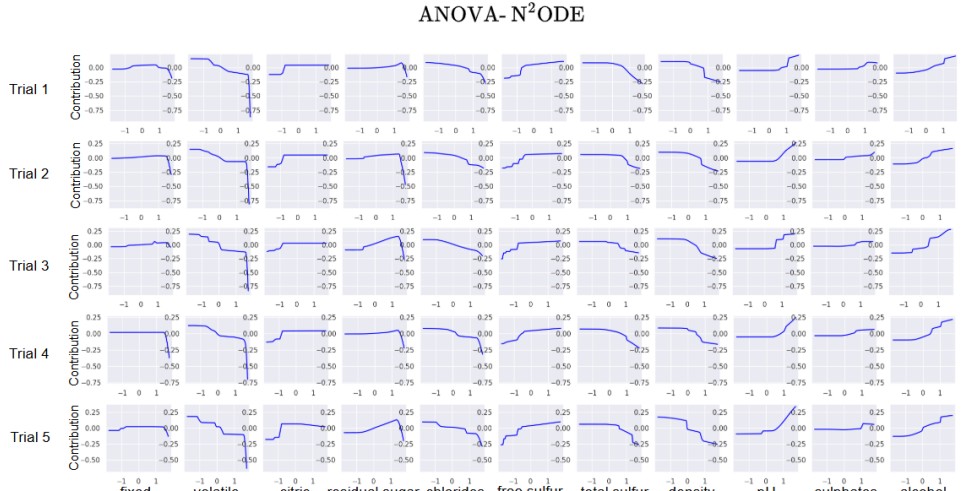

Figure 22: **Plots of the functional relations of the main effects in ANOVA-N$^2$ODE on WINE dataset.**

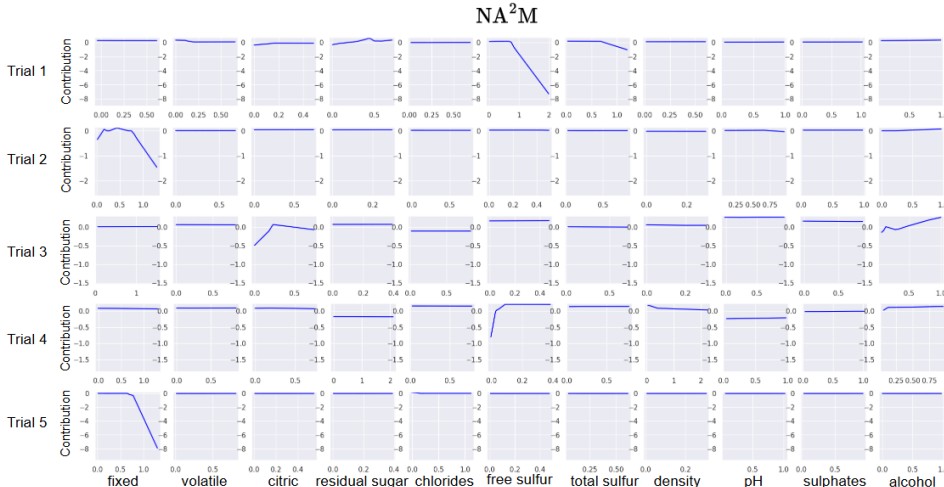

Figure 23: **Plots of the functional relations of the main effects in NA$^2$M on WINE dataset.**

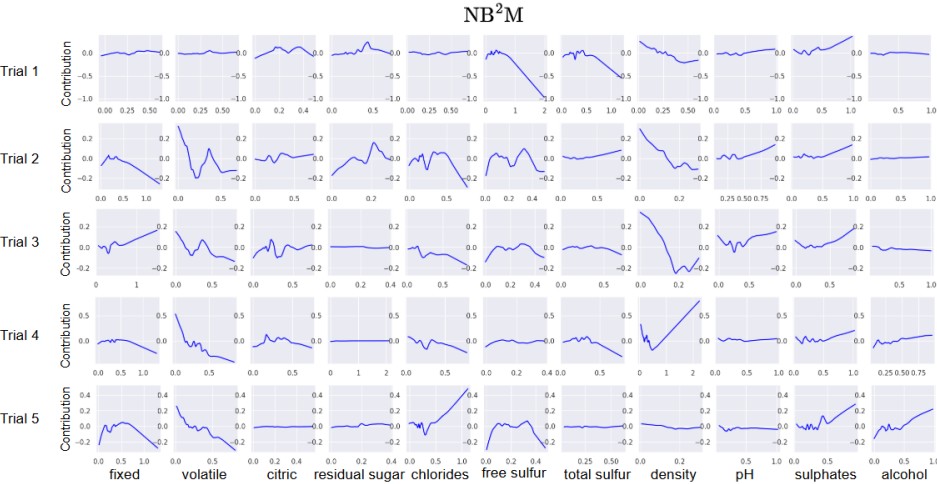

Figure 24: **Plots of the functional relations of the main effects in NB$^2$M on WINE dataset.**

## F.2 STABILITY OF ANOVA-SHAP ON CALHOUSING DATASET

We conduct an experiment to evaluate the stability of ANOVA-SHAP on CALHOUSING dataset. We calculate the global importance of features for 10 trials using the $l_1$ norm of the ANOVA-SHAP values defined in (2). Finally, we compute the stability score of ANOVA-SHAP as the average of Hamming distance between the global importance ranks across all pairs in the trials. Table 19 presents the results of stability scores of ANOVA-SHAP which are normalized by the that of ANOVA-N$^1$ODE (ANOVA-N$^2$ODE). We confirm that our model provides significantly more stable ANOVA-SHAP interpretations compared to other baseline models.

Table 19: **Results of average of Hamming distance.** A smaller distance indicates that the interpretation of ANOVA-SHAP is more stable.

| Model | ANOVA-N$^1$ODE | NA$^1$M | NB$^1$M |
|---|---|---|---|
| Average of Hamming distance | **1.000** | 6.188 | 2.408 |
| Model | ANOVA-N$^2$ODE | NA$^2$M | NB$^2$M |
| Average of Hamming distance | **1.000** | 5.157 | 2.663 |

## F.3 SCATTER PLOTS OF STABILITY SCORE

In this section, we present the scatter plots of the stability score on WINE dataset. It is obvious that ANOVA N$^1$ODE as well as ANOVA N$^2$ODE are more stable in estimation of the components than NAM and NBM.

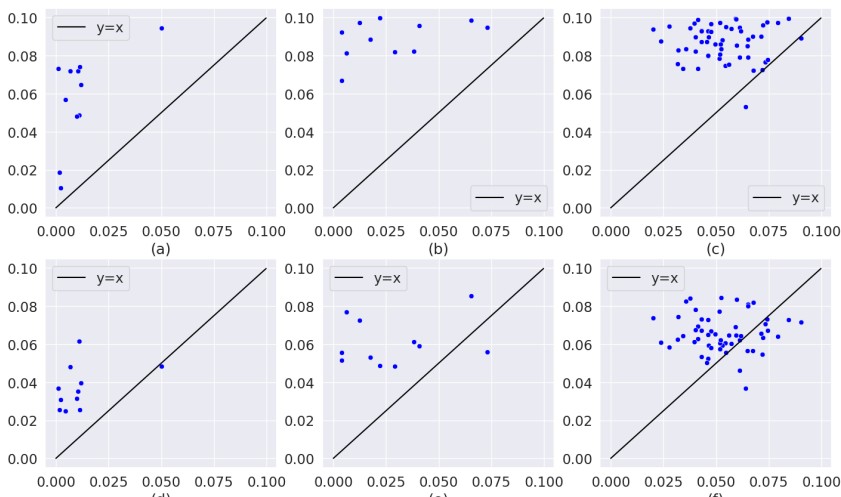

Figure 25: **Scatter plots of the stability scores on WINE dataset.** Figures (a) and (d) are the scatter plots for the stability scores of the main effects, where the x-axis is the stability score of ANOVA-N$^1$ODE and the y-axis is the stability scores of NA$^1$M in (a) and NB$^1$M in (d). Figures (b) and (e) are the scatter plots for the stability scores of the main effects, where the x-axis is the stability score of ANOVA-N$^2$ODE, and the y-axis is the stability score of NA$^2$M in (b) and NB$^2$M in (e). Figures (c) and (f) compare the stability scores of the second order interactions of ANOVA-N$^2$ODE to those of NA$^2$M and NB$^2$M. Each dot in the scatter plots corresponds to each component.

# G ADDITIONAL EXPERIMENTS ON HIGH-DIMENSIONAL DATASETS

## G.1 RESULTS OF UNNORMALIZED STABILITY SCORE.

Table 20 presents the original stability scores $\mathcal{SC}(f)/|S|$ of the normalized stability scores presented in Table 1.

Table 20: **Results of stability scores in each model on the real datasets.**

| Dataset | ANOVA-N$^1$ODE | NA$^1$M | NB$^1$M | ANOVA-N$^2$ODE | NA$^2$M | NB$^2$M |
|---|---|---|---|---|---|---|
| CALHOUSING | 0.012 | 0.045 | 0.039 | 0.035 | 0.071 | 0.075 |
| WINE | 0.011 | 0.058 | 0.043 | 0.049 | 0.087 | 0.065 |
| ONLINE | 0.031 | 0.076 | 0.054 | 0.052 | 0.072 | 0.072 |
| ABALONE | 0.008 | 0.013 | 0.026 | 0.028 | 0.047 | 0.038 |
| FICO | 0.035 | 0.046 | 0.046 | 0.048 | 0.089 | 0.075 |
| CHURN | 0.017 | 0.027 | 0.047 | 0.047 | 0.089 | 0.080 |
| CREDIT | 0.021 | 0.069 | 0.025 | 0.036 | 0.089 | 0.053 |
| LETTER | 0.017 | 0.022 | 0.014 | 0.026 | 0.075 | 0.081 |
| DRYBEAN | 0.028 | 0.074 | 0.070 | 0.053 | 0.088 | 0.081 |

## G.2 EXTENSION TO NEURAL BASIS MODEL

Similarly to NBM (Radenovic et al., 2022), we can consider extension of ANOVA-NODE using ANOVA-NODTs as basis functions. We call this extension model as NBM-NODE. Consider basis ANOVA-NODTs i.e., $\{h_k(x|\phi_k) : \mathbb{R} \to \mathbb{R}, k = 1, ..., B\}$, then the NBM-N$^1$ODE $f_{\text{NBM-N}^1\text{ODE}}(\mathbf{x})$ is defined as

$$f_{\text{NBM-N}^1\text{ODE}}(\mathbf{x}) = \sum_{j=1}^{p} f_{\text{NBM-N}^1\text{ODE}}^j(x_j) w_j \tag{8}$$

where $f_{\text{NBM-N}^1\text{ODE}}^j(x_j) = \sum_{k=1}^{B} h_k(x_j|\phi_k)a_{jk}$ for $j = 1, ..., p$ and and $h_k(x|\phi_k)$ satisfy the sum-to-zero condition with respect to the uniform distribution for $\mu_j$. NBM-N$^1$ODE can be easily extended to NBM-N$^2$ODE in a similar way as Radenovic et al. (2022).

Figures 26 and 27 show the plots of the functional relations of the main effects estimated by NBM-N$^1$ODE on the WINE dataset and the CALHOUSING dataset Table 21 shows the prediction performance of NBM-N$^1$ODE, and Table 22 presents the results of stability scores normalized by the that of ANOVA-N$^1$ODE. We observe that NBM-N$^1$ODE also exhibits similar prediction performance and stability to ANOVA-N$^1$ODE.

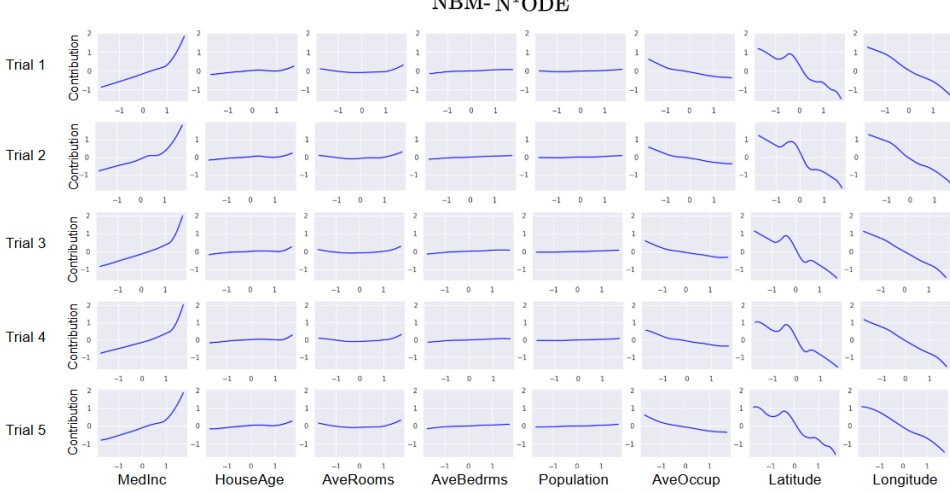

Figure 26: **Plots of the functional relations of the main effect estimated by NBM-N$^1$ODE on 5 randomly sampled training data from CALHOUSING dataset.**

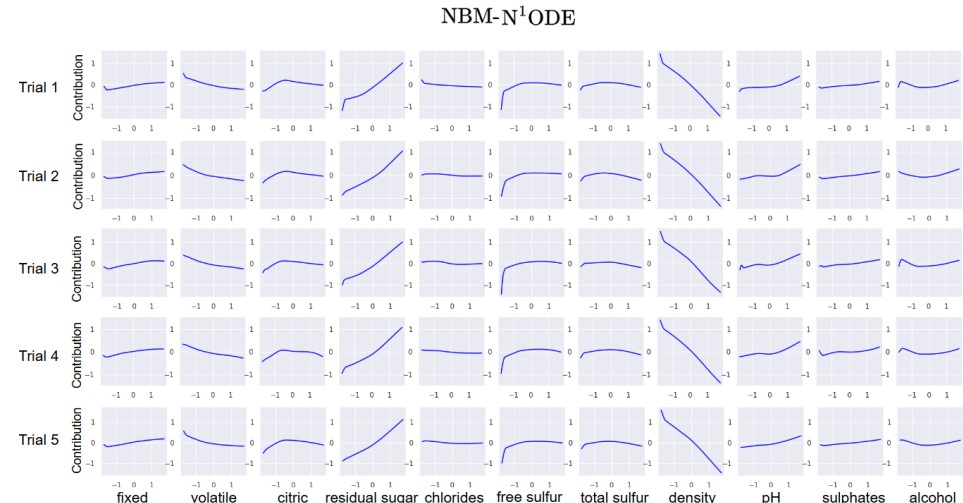

Figure 27: **Plots of the functional relations of the main effects estimated by NBM-N$^1$ODE on 5 randomly sampled training data from WINE dataset.**

Table 21: **Results of prediction performance.**

|  | CALHOUSING | WINE |
| --- | --- | --- |
| NBM-N$^1$ODE | 0.604 (0.001) | 0.720 (0.02) |
| ANOVA-N$^1$ODE | 0.614 (0.001) | 0.725 (0.02) |

Table 22: **Results of stability score.**

|  | ANOVA-N$^1$ODE | NBM-N$^1$ODE | NA$^1$M | NB$^1$M |
| --- | --- | --- | --- | --- |
| CALHOUSING | 1.000 | **0.750** | 3.750 | 3.250 |
| WINE | **1.000** | 1.571 | 5.273 | 3.364 |

# H  ADDITIONAL EXPERIMENTS FOR COMPONENT SELECTION

Table 23 presents the averages and standard deviations of the prediction performance of the models used in the component selection experiment. The three models perform similarly.

Table 23: **The results of prediction performance.** We report the averages and standard deviations of RMSEs of ANOVA-N$^2$ODE, NA$^2$M and NB$^2$M on 10 synthetic datasets generated from $f^{(1)}$, $f^{(4)}$ and $f^{(3)}$.

| | | | GA$^2$M | |
| :---: | :---: | :---: | :---: | :---: |
| Synthetic function | Measure | ANOVA N$^2$ODE | NA$^2$M | NB$^2$M |
| $f^{(1)}$ | RMSE ↓ | 3.483 (0.03) | **3.474** (0.03) | 3.511 (0.03) |
| $f^{(2)}$ | RMSE ↓ | 0.076 (0.001) | 0.088 (0.005) | **0.075** (0.001) |
| $f^{(3)}$ | RMSE ↓ | 0.161 (0.003) | 0.183 (0.016) | **0.137** (0.003) |

# I ADDITIONAL EXPERIMENTS FOR PREDICTION PERFORMANCE OF DECISION TREE

Table 24 presents the averages and standard deviations of the prediction performance of decision tree (Breiman, 2017) for 10 trials. We implemented a decision tree by using the scikit-learn python package (Pedregosa et al., 2011b) and turned by using the optuna python package based on below range of hyper-parameters.

- Range of max depth = [2 ,12]
- Range of min_samples_leaf = [2,10]
- Range of min_samples_split = [2,10]
- Range of max_leaf_nodes = [2,10]

Table 24: **Results of the prediction performance in decision tree and ANOVA-NODE.**

| Dataset | Measure | Decision Tree | ANOVA-N$^1$ODE | ANOVA-N$^2$ODE |
|---------|---------|---------------|----------------|----------------|
| CALHOUSING | RMSE ↓ | 0.671 ( 0.02 ) | 0.614 ( 0.01 ) | 0.512 ( 0.01 ) |
| WINE | RMSE ↓ | 0.811 ( 0.03 ) | 0.725 ( 0.02 ) | 0.704 ( 0.02 ) |
| ONLINE | RMSE ↓ | 1.119 ( 0.26 ) | 1.111 ( 0.25 ) | 1.111 ( 0.25 ) |
| ABALONE | RMSE ↓ | 2.396 ( 0.08 ) | 2.135 ( 0.09 ) | 2.087 ( 0.08 ) |
| FICO | AUROC ↑ | 0.704 ( 0.02 ) | 0.799 ( 0.007 ) | 0.800 ( 0.007 ) |
| CHURN | AUROC ↑ | 0.676 ( 0.03 ) | 0.839 ( 0.012 ) | 0.842 ( 0.012 ) |
| CREDIT | AUROC ↑ | 0.890 ( 0.02 ) | 0.983 ( 0.005 ) | 0.984 ( 0.006 ) |
| LETTER | AUROC ↑ | 0.745 ( 0.001 ) | 0.900 ( 0.003 ) | 0.984 ( 0.001 ) |
| DRYBEAN | AUROC ↑ | 0.975 ( 0.0002 ) | 0.995 ( 0.001 ) | 0.997 ( 0.001 ) |

## J    ADDITIONAL EXPERIMENTS FOR RUNTIME ON VARIOUS DATASETS.

We conducted additional experiments to assess the improvement in scalability. We consider $NA^1M$, which has 3 hidden layers with 16, 16, and 8 units; 10 basis DNNs for $NB^1M$, which has 3 hidden layers with 32, 16, and 16 units; 10 trees for each component in ANOVA-$N^1$ODE; and 10 basis functions in NBM-$N^1$ODE. Table 25 presents the results of runtime in $NA^1M$, $NB^1M$, ANOVA-$N^1$ODE, and NBM-$N^1$ODE on ABALONE, CALHOUSING, and ONLINE datasets. Note that our computational environment consists of RTX 3090 and RTX 4090.

Table 25: **Results of runtime in $NA^1M$, $NB^1M$, ANOVA-$N^1$ODE, and NBM-$N^1$ODE.**

| Dataset | Size of dataset | # of features | $NA^1M$ | $NB^1M$ | ANOVA-$N^1$ODE | NBM-$N^1$ODE |
|---|---|---|---|---|---|---|
| ABALONE | 4K | 10 | 6.6 sec | 3.0 sec | 4.2 sec | 1.5 sec |
| CALHOUSING | 21K | 8 | 14.1 sec | 4.1 sec | 9,7 sec | 3.5 sec |
| ONLINE | 40K | 58 | 68 sec | 15.6 sec | 70 sec | 9,8 sec |

# K ADDITIONAL EXPERIMENTS FOR APPLICABILITY OF ANOVA-SHAP

As explained in Section 3.1, ANOVA-SHAP value of the estimated ANOVA-NODE model can be easily calculated from the estimated components. To evaluate the similarity between ANOVA-SHAP and SHAP without feature independence, we compare ANOVA-SHAP and SHAP values of the estimated ANOVA-$N^2$ODE model on CALHOUSING dataset, where SHAP is computed using Deep-SHAP of Lundberg (2017). Note that the computation time of ANOVA-SHAP was approximately 1,600 times shorter than that of Deep-SHAP.

Figure 28 presents the boxplots of the absolute differences between ANOVA-SHAP and Deep-SHAP values at each data point for the 8 features, based on 1,000 data points which are randomly sampled from the test data, where ANOVA-SHAP values of NAM and NBM are calculated by the equation (2) with the estimated components by NA$^2$M and NB$^2$M, respectively. The absolute differences between ANOVA-SHAP and Deep-SHAP of ANOVA-NODE are distributed around zero which indicates that ANOVA-SHAP is a computationally efficient alternative to Deep-SHAP for ANOVA-NODE. In contrast, the boxplots for NA$^2$M and NB$^2$M, which either are far from zero or have large variations in many cases, imply that the formula (2) of ANOVA-SHAP is only applicable when the components satisfy the sum-to-zero condition. The results for stability of ANOVA-SHAP are given in Appendix F.2.

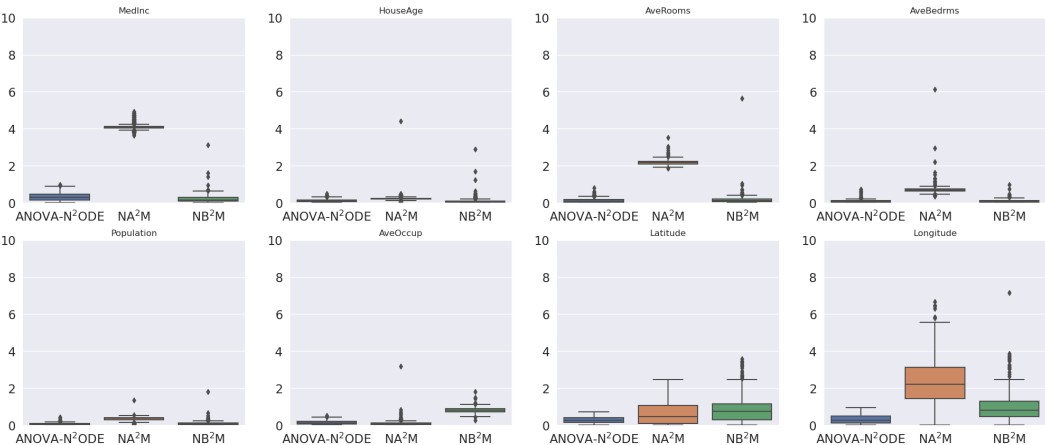

Figure 28: **Boxplots of the absolute differences between Deep-SHAP and ANOVA-SHAP values.**

## L  ANOVA-NODE WITHOUT SUM-TO-ZERO CONDITION

We investigate the performance of ANOVA-NODE without the sum-to-zero condition, which we denote GAM-NODE, by analyzing CALHOUSING and WINE datasets. In GAM-NODE, all of the heights at the terminal nodes of ANOVA-NODE are learnable parameters.

Table 26: **Comparison of ANOVA-NODE and GAM-NODE**. We report the averages of RMSE and stability score (normalized by the that of ANOVA-N$^1$ODE or NOVA-N$^2$ODE) for 10 trials.

| | ANOVA-N$^1$ODE | GAM-N$^1$ODE | ANOVA-N$^2$ODE | GAM-N$^2$ODE |
|---|---|---|---|---|
| CALHOUSING | 0.614 (**1.000**) | 0.580 (1.500) | 0.512 (**1.000**) | 0.502 (1.690) |
| WINE | 0.725 (**1.000**) | 0.713 (2.550) | 0.704 (**1.000**) | 0.690 (1.300) |

Table 26 presents (RMSE, stability score) of ANOVA-NODE and GAM-NODE based on 10 randomly selected datasets. Without the sum-to-zero condition, we observe increasing in the stability score. In particular, when the second order interactions are in the model, the main effects are estimated very unstably.

In Table 26, we observe that the prediction performance of GAN-NODE is (slightly) better than that of ANOVA-NODE. One reason could be that ANOVA-NODE is more vulnerable to the local minima problem. Further studies would be worth pursuing.

Figure 29 and 30 present the plots of the functional relations of the main effects on CALHOUSING and WINE dataset in GAM-N$^2$ODE. We observe that GAM-N$^2$ODE estimates the components more unstable compared to ANOVA-N$^2$ODE.

**Comparison between NODE-GAM and ANOVA-NODE.**  In NODE-GAM (Chang et al., 2021), the feature function $F^c$ of NODT at detph $c$ is a sparse weighted sum of input features by using $entmax_\nu$ and temperture parameter $T$. In other words, for a given depth $D$ and $c = 1, ..., D$, $F^c$ is defined as below.

$$F^c(\mathbf{x}) = \mathbf{x} \cdot entmax_\nu\left(\frac{\theta_F}{T}\right)$$

$$= \sum_{j=1}^{p} x_j w_j$$

where $\theta_F = (\theta_{F1}, ..., \theta_{Fp})^\top$ is a learnable parameter and $w_j = entmax_\nu\left(\frac{\theta_F}{T}\right)_j$. They expect the weights $\{w_1, ..., w_p\}$ to be trained as 0 or 1, but these weights may not all be 0 and 1. In other words, in NODE-GA$^1$M, NODTs may estimate the higher-order components rather than main effects. Therefore, it is difficult to consider NODE-GAM as the functional ANOVA model which decomposes a high-dimensional function into the sum of low-dimensional functions. Furthermore, in NODE-GAM, we can not obtain the estimated component function.

However, the feature function $F^c$ of ANOVA-NODT for component $S$ at depth $c$ uses only the input features corresponding to $S$. For $c = 1, ..., |S|$, we use feature funtion defined as

$$F^c(\mathbf{x}) = (\mathbf{x}_S)_c$$

ANOVA-NODE estimates component $f_S$ by an ensemble of ANOVA-NODTs corresponding to $S$. Therefore, ANOVA-NODE is a functional ANOVA model. Therefore, although both NODE-GAM and ANOVA-NODE utilize NODT, they are fundamentally different models.

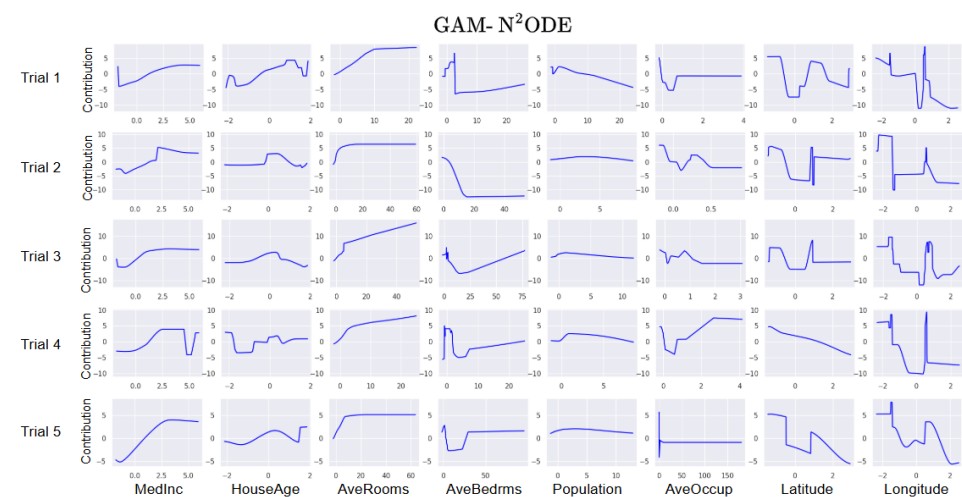

Figure 29: **Plots of the functional relations of the main effects on 5 randomly sampled training data from** CALHOUSING **datasets.**

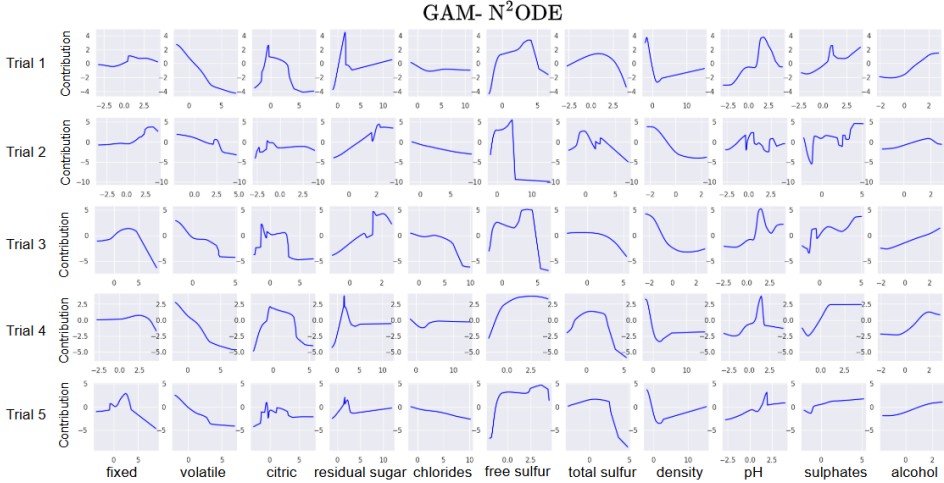

Figure 30: **Plots of the functional relations of the main effects on 5 randomly sampled training data from** WINE **datasets.**

## M ON THE POST-PROCESSING FOR THE SUM-TO-ZERO CONDITION

We have seen that NA$^2$M and NB$^2$M are competitive in prediction performance even though they are poor in estimating the components. There is a way to transform any estimate of a component to one that satisfies the sum-to-zero condition (Lengerich et al., 2020).

We consider a estimated GA$^d$M $\hat{f}(\mathbf{x}) = \beta_0 + \sum_{k=1}^d \sum_{S_k \subseteq [p]} \hat{f}_{S_k}(\mathbf{x}_{S_k})$ where $S_k$ is a component for $|S_k| = k$. We write $f_S$ instead of $f_S(\mathbf{x}_S)$ for notational simplicity since $\mathbf{x}$ is fixed. First of all, for component $S_d$, we can transform $\hat{f}_{S_d}$ into

$$\tilde{f}_{S_d} = \hat{f}_{S_d} + \sum_{k=1}^d \sum_{V \subseteq S_d, |V|=k} (-1)^{d-k} \int_{\mathcal{X}_V} \hat{f}_{S_d} d\Pi_{j \in V} \mu_j$$

where $\tilde{f}_{S_d}$ satisfies the sum-to-zero condition. Next, for $k = 1, ..., d$, we redefine $\hat{f}_{S_{d-k}}$ into

$$\hat{f}_{S_{d-k}} = \hat{f}_{S_{d-k}} - (-1)^{d-k} \int_{\mathcal{X}_{S_k}} \hat{f}_{S_d} d\Pi_{j \in S_k} \mu_j$$

where $S_{d-k} = S_d \backslash S_k$. If this process is performed sequentially for all components in order, all $\tilde{f}_{S_k}$ terms in $\hat{f}(\mathbf{x}) = \tilde{\beta}_0 + \sum_{k=1}^d \sum_{S_k \subseteq [p]} \tilde{f}_{S_k}(\mathbf{x}_{S_k})$ satisfy the sum-to-zero condition.

Let us consider performing post-processing for GA$^d$M on a given dataset. The computational order for post-processing of a single point $\mathbf{x}$ is $\mathcal{O}(dn^{d-1})$. Therefore, if post-processing is carried out for all data points, the computational order becomes $\mathcal{O}(dn^d)$. In other words, not only global interpretation (e.g., $l1$ norm, functional relation plots, etc.) but also local interpretation is practically infeasible. Furthermore, performing post-processing requires storing a dataset, which causes memory efficiency issues.

Additionally, in GA$^d$M, when post-processing is performed, and then calculating the ANOVA-SHAP $\phi_j(\mathbf{x})$ for a given point $\mathbf{x}$ requires a computational order of $\mathcal{O}(p^{d-1}dn^d)$, which is even more demanding.

Table 27 compares the stability scores of the main effects of 'Latitude' and 'Longitude' for ANOVA-N$^2$ODE, NA$^2$M and NB$^2$M, and Figure 31 draws the 5 functional relations of the main effects of 'Latitude' and 'Longitude' estimated by ANOVA-N$^2$ODE, NA$^2$M and NB$^2$M on 5 randomly sampled training data. It is observed that NA$^2$M and NB$^2$M are still unstable even after the post-processing, which indicates that instability in NAM and NBM is not only from unidentifiability but also instability of DNN.

The situation becomes different when we apply the post-processing to GAM-NODE. Table 28 presents the stability scores of ANOVA-N$^2$ODE and post-processed GAM-N$^2$ODE normalized by the stability score of ANOVA-N$^2$ODE, and Figure 32 presents the plots of the functional relations of the main effects estimated by GAM-N$^2$ODE on 5 randomly sampled training data. Interestingly, unlike NAM and NBM, it is observed that GAM-N$^2$ODE becomes more stable after post-processing.

The post-processing would not be a practically usable method since computation cost is too large for calculating the integration. The order of computation for the post-processing GAM-N$^d$ODE is $O(n^d)$, where $n$ is the number of data points to which the post-processing is applied.

Table 27: **Stability scores for 'Latitude' and 'Longitude'.**

| Model | ANOVA-N$^2$ODE | NA$^2$M | NB$^2$M |
|---|---|---|---|
| Latitude | 0.006 | 0.067 | 0.104 |
| Longitude | 0.015 | 0.094 | 0.103 |

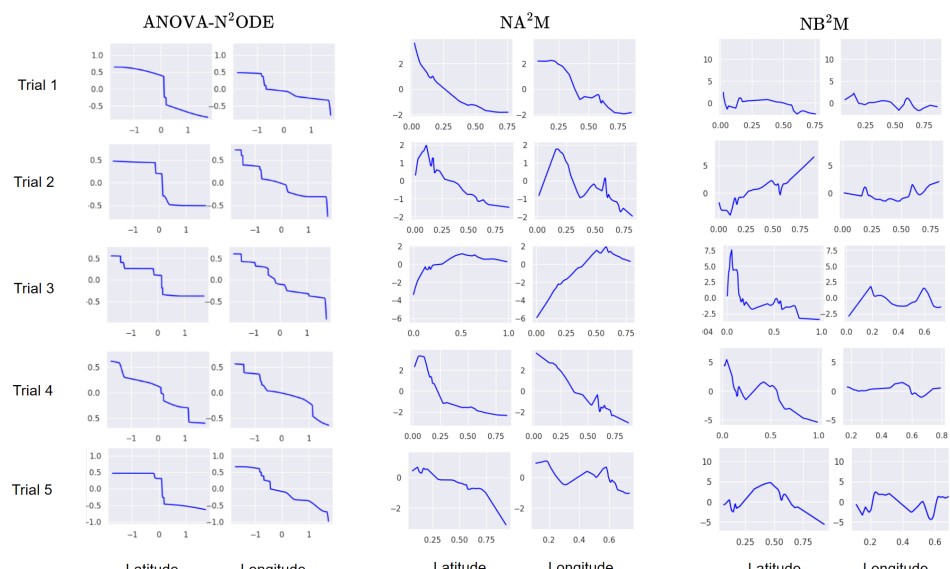

Figure 31: **Plots of the functional relations of 'Latitude' and 'Longitude'.**

Table 28: **Stability scores of the post-processed GAM-N$^2$ODE on CALHOUSING dataset.**

| Model | ANOVA-N$^2$ODE | post-processed GAM-N$^2$ODE |
|---|---|---|
| Stability score | **1.000** | 1.257 |

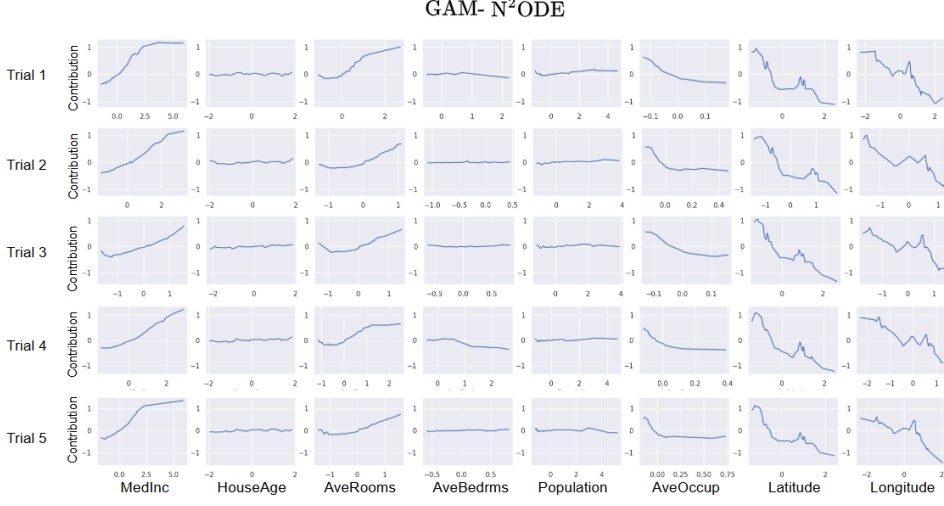

Figure 32: **Plots of the functional relations of the main effects in post-processed GAM-N$^2$ODE on CAL-HOUSING dataset.**

# N    DISCUSSION OF RELATED LITERATURE.

In this section, we describes the comparison between ANOVA-NODE and additive higher-order factorization machines (AHOFMs, Rügamer (2024)).

**Higher-Order Factorization Machines (HOFMs).**  Higher-Order Factorization Machines (HOFMs, Blondel et al. (2016)) are a Factorization Machine(FM) which considers up to higher-order interaction. Blondel et al. (2016) proposed an algorithm that efficiently learns HOFMs using the properties of the ANOVA kernel, even as the input dimension p increases.

**Additive higher-order factorization machines (AHOFMs).**  AHOFMs (Rügamer, 2024)) approximate each component in the functional ANOVA model using a spline basis representation. That is, for $j = 1, ..., p$, AHOFMs estimate $f_j$ by

$$f_j(x_j) = \sum_{m=1}^{M_j} B_{m,j}(x_j)\beta_{m,j}$$

where $\{B_{1,j}, ..., B_{M_j,j}\}$ are the basis functions and $\{b_{1,j}, ..., b_{M_j,j}\}$ are the coefficients. Furthermore, for high-order interaction $S$, AHOFM estimates $f_S$ via extended spline basis representation which used the tensor product spline. Finally, Rügamer (2024) proposed an algorithm for efficiently learning AHOFMs, which applies the method used in HOFMs to AHOFMs, taking higher-order interactions into account.

**Comparison between ANOVA-NODE and AHOFMs.**  We first describe a comparison of the scalability between AHOFMs and ANOVA-NODE, followed by the applicability of AHOFM to ANOVA-NODE, and finally, the differences between ANOVA-NODE and AHOFM.

If all higher-order interactions are considered, the method proposed by Rügamer (2024)) is more efficient in terms of scalability compared to ANOVA-NODE. However, the experiments in this paper show that considering interactions up to the second order is sufficient for real data. Moreover, when all higher-order interactions are considered in the functional ANOVA model, it is likely to be overfitted.

Since ANOVA-NODT is also a model represented by the spline basis representation of equation (2) in Rügamer (2024)($entmax$ can be viewed as basis function) factorization method can be used to improve the scalability of ANOVA-NODE. Applying the method used by AHOFMs to ANOVA-NODE seems like an interesting research topic, but in this paper, we proposed NBM-NODE in Appendix G.2, which enhances the scalability of ANOVA-NODE by utilizing the method by which Radenovic et al. (2022) improved the scalability of NAM (Agarwal et al. (2021)).

There are two main differences between ANOVA-NODE and AHOFMs: one is whether the basis function is learned, and the other is whether the sum-to-zero condition is satisfied. To be specific, AHOFMs do not learn the basis function, but ANOVA-NODE can be seen as learning the basis function by learning the parameters $\gamma$ and b in the $entmax$ function. Also, since AHOFMs do not satisfy the sum-to-zero condition, the estimated components are not identifiable. However, unlike AHOFMs, the estimated components by ANOVa-NODE are identifiable. That is, ANOVA-NODE provides more reliable interpretations than AHOFMs.

