# OpenReview forum: "ANOVA-NODE: An identifiable neural network for the functional ANOVA model for better interpretability"
_ICLR.cc/2025/Conference — Submitted to ICLR 2025_

### Official Review · Reviewer_yGHL · 2024-10-21

**Soundness:** 2
**Presentation:** 3
**Contribution:** 2
**Rating:** 5
**Confidence:** 4

**Summary:**

The paper introduces ANOVA-NODE, a novel model designed for better interpretability in AI by improving the stability of functional ANOVA model estimations. Traditional neural network models for ANOVA suffer from instability due to non-unique decompositions, but ANOVA-NODE addresses this by ensuring a unique functional decomposition through a modification of Neural Oblivious Decision Ensembles (NODE). The authors theoretically prove that ANOVA-NODE approximates smooth functions well and is related to SHAP values, a popular interpretability method. Experimental results demonstrate that ANOVA-NODE provides more stable and interpretable component estimates compared to existing methods.

**Strengths:**

- **Theoretical Contributions**: The authors provide a solid theoretical foundation, including proofs.
- **Innovative Solution**: The paper presents an innovative approach to addressing the unidentifiability problem in functional ANOVA models, offering stability in component estimation.
- **Experimental Validation**: While there are some concerns about the experimental setup (discussed below), the results demonstrate that ANOVA-NODE outperforms baseline models in both stability and prediction accuracy.
- **SHAP Link**: The connection between the sum-to-zero constraint and SHAP values offers a novel perspective on component-based interpretation, presenting an efficient alternative to existing SHAP calculation methods.

**Weaknesses:**

## Major Concerns / Questions
- **Motivation and Relevance**: Although the paper addresses the technical issue of unidentifiability in functional ANOVA models, it does not provide a clear explanation of why this problem is significant in practical applications. Including practical examples that demonstrate where current methods fail and how ANOVA-NODE offers concrete improvements would greatly enhance the impact of the paper. Although Appendix E discusses interpretability, the advantage of using ANOVA-NODE over existing methods (e.g., applying SHAP to other models) remains unclear. Clarification on this point would be beneficial.
- **Hyperparameter Tuning**: The ranges for tuning the hyperparameters and the selection process are not clearly described. Additionally, the chosen hyperparameters for the benchmarks raise concerns. For NODE, only five trees are used, which is an unusually small number, inconsistent with the original paper and other related works, where values like 1024 are more typical. Similarly, for XGBoost, both the tree depth and the number of estimators are surprisingly low, often limited to just 100 estimators with a depth of 3. Moreover, the learning rate is the same for all datasets except one. Although the authors state that the learning rate was chosen "by empirical trials," it is difficult to believe that this reflects a thorough tuning process. Other key hyperparameters are neither reported nor seemingly optimized, which likely affects the models' performance. Yet, the results based on these suboptimal hyperparameters are used to support claims that "ANOVA-NODE provides more stable estimation and interpretation of each component compared to the baseline models [...] **without losing prediction accuracy**" and that "ANOVA-NODE is the best." Given the current configuration, I find these claims unconvincing and poorly supported. To properly substantiate them, I strongly recommend a comprehensive hyperparameter tuning of the existing benchmarks, following best practices and relevant literature.
- **Code Availability**: The absence of the source code for review intensifies concerns about the evaluation setup, especially the selection of hyperparameters.


## Further Concerns / Questions
- **Baseline Comparison**: Including performance metrics for intrinsically interpretable models like linear or logistic regression and simple decision trees would provide a clearer baseline for comparison.
- **Related Work**: A section on related work is missing. The paper does not clearly distinguish ANOVA-NODE from existing methods. E.g. the authors claim that "The main idea of the proposed neural network is to model each component as the sum of special but simple neural networks that satisfy the sum-to-zero condition". What is the difference to NAM here?
- **Complexity of Presentation**: Certain sections, particularly those involving mathematical formulations (e.g., p.5), are dense and challenging to follow. More intuitive explanations and clearer descriptions of the model’s workings would improve accessibility.
- **Runtime and Efficiency**: The paper lacks information about the runtime of the proposed method and how it compares to existing benchmarks.
- **Performance Reporting**: The reporting of only normalized scores (Table 1) may be misleading. Providing unnormalized scores, at least in the appendix, would be more informative. Additionally, reporting only ranks in Table 2 can obscure performance differences. Including the averaged normalized distances to the best-performing model would strenghten the comparison.

**Questions:**

**Please see Weaknesses.**

### Further Questions:
- The performance of ANOVA-NODE is significantly lower than XGBoost for certain datasets (e.g., California Housing and Wine, Table 2). Is there an explanation for this comparatively poor performance?
- The results for the feature importance scores on California housing dataset (Tables 14 and 15) are confusing to me. Why is the importance of feature 6 very small for ANOVA-N1ODE while it has the by far highest importance for ANOVA-N2ODE?
- In the CBM example, how does the explanation of ANOVA-NODE differ from a simple linear model? Does it offer better performance? The results regarding monotone constraints also appear counterintuitive. Why is such a relation observed at all and how can enforcing monotonic constraints cause such a significant shift in results? A more detailed explanation is needed.

---

> ### Author Response · Authors · 2024-11-25
>
> We appreciate your feedback on our paper and have made every effort to address your insightful questions. We have uploaded the revised version of the paper, reflecting the reviewer's feedback.
>
> **Weakness 1. Motivation and Relevance.**
>
> In Section 3.1, the example which addresses the importance of identifiability was discussed.
> Moreover, the plots of the components estimated by ANOVA-NODE, which satisfies the sum-to-zero condition, and by NAM and NBM, which do not, are presented in Appendix F.1.
> While ANOVA-NODE estimates the components stably, NAM and NBM do not.
>
> Furthermore, in NA$^{2}$M, some main effects are estimated as a constant function, which would be partly
> because the main effects are absorbed into the second order interactions.
> However, such an issue does not occur in ANOVA-NODE.
>
> The advantages of ANOVA-NODE over SHAP are as follows.
>
>  **1.1. Functional relationship.**
>
> Unlike SHAP, ANOVA-NODE can deterministically identify the relationship between the components and the output.
> Therefore, in ANOVA-NODE, we can accurately assess the contribution of each component to the output.
>
> **1.2. Decomposition of components.**
>
> While SHAP provides an interpretation of the feature by considering all higher-order interactions that include the feature, ANOVA-NODE interprets each component individually, offering a more specific analysis.
> For example, consider $f(X_{1},X_{2}) =X_{1} +  X_{2} + X_{1}X_{2}$.
>
> Then, for given data points **X**$^{1}=(1,1)^{\top}$ and **X**$^{2}=(2,100)^{\top}$, then the SHAP value $\phi_{1}(f,$**X**$^{1})$ and $\phi_{1}(f,$**X**$^{2})$ is calculated as
>
> $ \phi_{1}(f,$**X**$^{1}) = {3\over 2} , \quad \phi_{1}(f,$**X**$^{2}) = 102 $
>
> In this case, while SHAP only provides an interpretation that combines both the main effect and interactions, ANOVA-NODE can provide interpretations for both the main effect and interactions via estimated component functions $(f_{1}, f_{1,2} )$.
>
> **1.3. Stability**
>
> We compare the stability of the local interpretation from SHAP and ANOVA-NODE on Wine dataset.
> As in the experiments of the paper, we randomly split the data, trained a XGB model with 800 trees, a learning rate of 0.005, and a max depth of 5, and calculated the SHAP values via Tree-SHAP, repeating this process a total of 10 times.
> We calculated the stability score, as defined in our paper, for the SHAP values from the 10 trials.
>
> **Table D.1 Results of stability score in ANOVA-N$^{1}$ODE + ANOVA-SHAP and  XGB + SHAP.**
> | Method | ANOVA-N$^{1}$ODE + ANOVA-SHAP | XGB + SHAP |
> |--------------------------------------------|-------|--------|
> | Stability score $\downarrow$ | 0.011 | 0.074  |
>
>
> Table D.1 presents the results of stability scores for local interpretations obtained using ANOVA-SHAP in ANOVA-NODE and Tree-SHAP in XGB.
> We observed that the local interpretation obtained using ANOVA-SHAP in the ANOVA-N$^{1}$ODE is approximately seven times more stable than that of SHAP in XGB.

---

> ### Author Response · Authors · 2024-11-25
>
> **Weakness 2. Hyperparameter Tuning.**
>
> As suggested by the reviewer, we additionally conducted an experiments of prediction performance for NODE and XGB by tuning the comprehensive hyper-parameters.
> We turned NODE by grid search over the following values.
>
>
> $\bullet$ num layers = [2,4,8]
>
> $\bullet$  tree depth = [6,8]
>
> $\bullet$ \# of trees in each layer = [256,512]
>
> Similarly, for XGB, we used below values for the grid search.
>
> $\bullet$ learning rate = [0.0001, 0.005, 0.01, 0.05 , 0.1]
>
> $\bullet$ max depth = [3, 5, 7]
>
> $\bullet$ \# of trees = [50,100,200,300,400,500,600,700,800,900,1000]
>
>
> **Table D.2 Results of prediction performance and hyper-parameters in XGB.**
> | Dataset | Measure | XGB's predicition performance | (\# of Tree , lr rate, max depth) |
> |--------------------------------------------|-------|--------|---|
> |Calhousing | RMSE $\downarrow$ | 0.452 (0.011)  | (900, 0.1, 5) |
> |Wine| RMSE $\downarrow$ | 0.635 (0.028)  | (800, 0.005, 5) |
> |Online| RMSE $\downarrow$ | 1.122 (0.26)  | (200, 0.001, 3) |
> |Abalone| RMSE $\downarrow$ | 2.164 (0.089)  | (800, 0.005, 5) |
> |FICO| AUROC $\uparrow$ | 0.796(0.008)  | (1000, 0.01, 3) |
> |Churn| AUROC $\uparrow$ | 0.846(0.012)  | (700, 0.01, 3) |
> |Credit| AUROC $\uparrow$ | 0.983(0.005)  | (400, 0.05, 3) |
> |Letter| AUROC $\uparrow$ | 0.997(0.001)  | (1000, 0.1, 7) |
> |Drybean| AUROC $\uparrow$ | 0.996(0.001)  | (600, 0.05, 3) |
>
>
> **Table D.3 Results of prediction performance and hyper-parameters in NODE.**
> | Dataset | Measure | NODE's predicition performance | (Depth , \# of layer, \# of tree in each layer) |
> |--------------------------------------------|-------|--------|---|
> |Calhousing | RMSE $\downarrow$ | 0.482 (0.014)  | (6, 2, 256) |
> |Wine| RMSE $\downarrow$ | 0.646 (0.027)  | (6, 2, 512) |
> |Online| RMSE $\downarrow$ | 1.112 (0.27)  | (8, 2, 512) |
> |Abalone| RMSE $\downarrow$ | 2.086 (0.092)  | (8, 2, 512) |
> |FICO| AUROC $\uparrow$ | 0.795 (0.008)  | (8, 4, 256) |
> |Churn| AUROC $\uparrow$ | 0.844 (0.013)  | (6, 4, 256) |
> |Credit| AUROC $\uparrow$ | 0.984 (0.009)  | (6, 4, 512) |
> |Letter| AUROC $\uparrow$ | 0.998 (0.001)  | (8, 8, 256) |
> |Drybean| AUROC $\uparrow$ | 0.996 (0.001)  | (6, 8, 256) |
>
> Tables D.2 and D.3 present the best prediction performance obtained through grid search for the XGB and NODE models, along with the corresponding hyper-parameters.
> Based on the results of the above experiments, we have revised the experimental results in the paper.
>
>
> **Weakness 3.Code Availability.**
>
> We have included the source code of ANOVA-NODE in the Supplementary Material.
>
>
> **Weakness 4. Baseline Comparison.**
>
> As suggested by the reviewer, we conducted additional experiment of prediction performance in simple decision tree by using sklearn python package.
> Table D.4 presents the results of prediction performance in simple decision tree.
> We add these results in Appendix I of the paper.
>
> **Table D.4 Results of prediction performance  in decision tree.**
> | Dataset | Measure | decision tree's predicition performance  |
> |--------------------------------------------|-------|--------|
> |Calhousing | RMSE $\downarrow$ | 0.730 (0.019)  |
> |Wine| RMSE $\downarrow$ | 0.871 (0.027)  |
> |Online| RMSE $\downarrow$ | 1.668 (0.28)  |
> |Abalone| RMSE $\downarrow$ | 2.992 (0.152) |
> |FICO| AUROC $\uparrow$ | 0.632 (0.008)  |
> |Churn| AUROC $\uparrow$ | 0.660 (0.016)  |
> |Credit| AUROC $\uparrow$ | 0.891 (0.014)  |
> |Letter| AUROC $\uparrow$ | 0.920 (0.004)  |
> |Drybean| AUROC $\uparrow$ | 0.947 (0.003)  |
>
>
> **Weakness 4. Related Work.**
>
> ANOVA-NODE estimates each component using the ANOVA-NODT.
> It is a modified version of NODT introduced in Section 2.3, which satisfies the sum-to-zero condition.
> On the other hand, NAM estimates each component using a deep neural network which does not satisfy sum-to-zero condition.
>
> **Weakness 5. Complexity of Presentation.**
>
> As pointed out by the reviewer, we revised the content corresponding to page 5 to make it clearer, more concise, and easier to understand in the revision.

---

> ### Author Response · Authors · 2024-11-25
>
> **Weakness 6. Runtime and Efficiency.**
>
> We conducted additional experiments on the runtime of NAM, NBM, ANOVA-NODE and NBM-NODE where NBM-NODE is an extended model of ANOVA-NODE, proposed in Appendix G.2 of this paper, which improves the scalability of ANOVA-N$^{1}$ODE by using basis functions, similar to the approach used in NBM.
> Note that in NBM-NODE, the number of ANOVA-NODT for basis does not depend on the number of components
>
> We conducted additional experiments to assess the improvement in scalability.
> We consider NA$^{1}$M, which has 3 hidden layers with 16, 16, and 8 units; 10 basis DNNs for NB$^{1}$M, which has 3 hidden layers with 32, 16, and 16 units; 10 trees for each component in ANOVA-N$^{1}$ODE; and 10 basis functions in NBM-N$^{1}$ODE.
>
> **Table D.5 Results of computaitonal complexity**
> | Model  | NA$^{1}$M| NB$^{1}$M| ANOVA-N$^{1}$ODE | NBM-N$^{1}$ODE|
> |---------------|-------|--------|--------|-------|
> | Run time| 6.6 sec | 3.0 sec  | 4.2 sec  | 1.5 sec |
> | \# of weight parameters | 4,490 | 1,113  | 670  | 170 |
>
> Table D.5 presents the runtime for each model when trained on a Abalone dataset and 100 epochs.
> Our computational environment consists of RTX 3090 and RTX 4090.
> We observed that our model has fewer weight parameters and takes less runtime compared to existing models (NAM and NBM).
>
>
> **Weakness 7. Performance Reporting.**
>
> As suggested by the reviewer, we report the unnormalized stability scores in Appendix G.2 of paper.
> The results reported in Table 2 are the AUROCs for component selection, not the ranks.
>
>
> **Question 1.**
>
> In particular, the black-box models XGB and NODE show good prediction performance, which is due to the presence of important higher-order interactions of degree 3 or higher in these two datasets (Calhousing and Wine dataset).
>
> **Question 2.**
>
> In Appendix E.1.1, ANOVA-NODE identified the important main effects in the following order: 7, 8, 1, 6, 3, 2, 4, 5.
> ANOVA-N$^{2}$ODE interpreted main effect 6 as the most important, because the effects of features 1, 7 and 8 in ANOVA-N$^{1}$ODE are decomposed into interactions such as (7,8), (1,7), (3,8), (1,8), (4,8), (2,7) and (1,5), whose importance scores are significant in ANOVA-N$^{2}$ODE.
> Therefore, the importance scores of main effects 1, 7, and 8 are relatively lower in ANOVA-N$^{2}$ODE.
>
> **Question 3.**
>
> If a linear model is used instead of ANOVA-NODE within the CBM framework, the interpretation of features for each image is limited to linear relationships, whereas using ANOVA-NODE allows for more flexible interpretations.
>
> The prediction performance results when using a linear model instead of ANOVA-N$^{1}$ODE in CBM are shown in Table D.6.
> The target variable is set as 'gender'.
>
> **Table D.6 Results of prediction performance in CBM + Linear and CBM + ANOVA-N$^{1}$ODE.**
> | Model  | CBM + Linear  | CBM + ANOVA-N$^{1}$ODE |
> |---------------|-------|--------|
> | Accuracy $\uparrow$ (std)  | 0.876 (0.006) | 0.985 (0.001) |
>
> We observed a decrease in prediction performance when using a linear model instead of ANOVA-N$^{1}$ODE in CBM.
> Additionally, the details of how the estimated function becomes monotone due to the monotone constraint are described in Appendix B.1.

---

> > ### Comment · Reviewer_yGHL · 2024-11-25
> > **Remaining Concerns**
> >
> > Thank you for your comprehensive response. I appreciate the effort you put into addressing the comments. However, I still have some remaining concerns, which I outline below:
> >
> > **W1:** While the clarification addresses some points, I believe the paper would benefit from a more detailed exposition of the motivation in the introduction. This would help to frame the work more effectively for the reader.
> >
> > **W2:** I appreciate the effort to conduct hyperparameter optimization (HPO). However, I suggest selecting the hyperparameter grids based on prior related work. For NODE, the grid is reasonably close to that suggested in the original paper. However, for XGB, the selected grid is suboptimal at best. Parameters such as `lambda` and `alpha` should at least be included in the tuning process, while the number of estimators could be fixed, as recommended in the literature `[1,2]`.
> >
> > **W3:** While I recommend providing code that reproduces the experiments and includes all relevant details, I acknowledge the value of being able to verify the overall method using the currently available code. Ensuring reproducibility remains an important aspect of scientific research.
> >
> > **W4:** The additional results are valuable; however, the hyperparameter configurations used and whether they were appropriately tuned remain unclear. Furthermore, the table would benefit from a direct comparison with the proposed method. At present, the separate, stand-alone table of results for decision trees makes it challenging to contextualize the findings.
> >
> > **W6:** Reporting runtimes for all datasets would be beneficial, even if relegated to the appendix. Relying on runtime data for a single (small) dataset provides limited insight. Nonetheless, the reported numbers suggest that runtime is not a significant issue overall.
> >
> > **W7:** The stability scores appear quite low across the board. Could the authors elaborate on whether stability is a common concern with most existing methods, or if this is specific to the proposed approach?
> >
> > `[1]` https://github.com/naszilla/tabzilla/blob/main/TabZilla/models/tree_models.py
> > `[2]` https://github.com/kathrinse/TabSurvey/blob/main/models/tree_models.py
> >
> > For now, I will retain my current rating, as I still have concerns regarding the evaluation. I believe the paper requires further improvements to meet the acceptance threshold for ICLR.

---

> ### Author Response · Authors · 2024-11-26
>
> We appreciate your valuable comments.
>
> **W1.**
>
> As suggested by the reviewer, a figure 1 corresponding to the motivation has been added to the introduction to enhance the clarity of the paper's contributions.
>
> **W2.**
>
> As suggested by the reviewer, we conducted additional experiments on the hyper-parameter optimization of XGB.
> As the first approach, we fixed the number of trees to 1000 and performed a grid search for the learning rate, max_depth, alpha, and lambda using the values below.
>
> $\bullet$ learning rate = [0.001, 0.005, 0.01, 0.05, 0.1]
>
> $\bullet$ max depth = [3, 5, 7]
>
> $\bullet$ alpha = [1e-6, 1e-4, 1e-2, 1.0]
>
> $\bullet$ lambda= [1e-6, 1e-4, 1e-2, 1.0]
>
> Table D.7 presents the results of prediction performance in XGB on the optimized hyper-parameters by the grid search.
>
> **Table D.7. Results of prediction performance in XGB on the optimized hyper-parameters by the grid search**
> | Dataset | XGB's prediction performance | learning rate, max depth , alpha, lambda  |
> |--------------------------------------------|-------|--------|
> | Calhousing | 0.450 (0.011) | 0.05, 5, 1.0, 1.0  |
> | Abalone | 2.165 (0.093) | 0..01, 5, 1e-6, 1.0  |
> | Wine | 0.640 (0.027) | 0.05, 5, 1.0, 1.0  |
> | Online| 1.189 (0.25) | 0.01, 5, 1.0, 1.0  |
> | FICO | 0.796 (0.009) | 0.01, 3, 1e-4, 1e-6  |
> | Churn | 0.846 (0.012) | 0.01, 3, 1.0, 1.0  |
> | Letter | 0.997 (0.0004) | 0.1, 5, 1.0, 1.0  |
> | Credit | 0.983 (0.005) | 0.05, 1e-2, 1e-2  |
> | Drybean | 0.997 (0.0004) | 0.1, 5, 1.0, 1.0 |
>
> As the second approach, we used the Optuna package in Python to tune the hyper-parameters of XGB.
> The ranges of hyper-parameters for Optuna are as follow.
>
> $\bullet$ The range of \# of tree = [100,1000]
>
> $\bullet$ The range of max depth = [2,12]
>
> $\bullet$ The range of learning rate = [0.001, 1.0]
>
> $\bullet$ The range of lambda= [1e-8, 1.0]
>
> $\bullet$ The range of alpha= [1e-8, 1.0]
>
> Table D.8 presents the results of prediction performance in XGB on the optimized hyper-parameters by the Optuna python package.
>
> **Table D.8. Results of prediction performance in XGB on the optimized hyper-parameters by the Optuna python package.**
> | Dataset | XGB's prediction performance | /# of tree, learning rate, max depth , alpha, lambda  |
> |--------------------------------------------|-------|--------|
> | Calhousing | 0.456 (0.012) | 400, 0.107, 7, 2e-7, 0.794   |
> | Abalone | 2.153 (0.089)  |344, 0.028, 3, 1e-4, 0.009   |
> | Wine |0.632 (0.031)  | 467, 0.056, 8, 1e-4, 0.047   |
> | Online|1.119 (0.25) |992, 0.021, 2, 0.006, 0.547   |
> | FICO | 0.794 (0.008) |566, 0.037, 2, 7e-7, 7e-5  |
> | Churn |0.846 (0.009)| 231, 0.056, 3, 7e-5, 2e-7 |
> | Letter |0.997 (0.0003)|950, 0.118, 10, 1e-6, 2e-5    |
> | Credit |0.983 (0.011)|398, 0.317, 7, 0.054, 0.013  |
> | Drybean |0.996 (0.0004) |478, 0.064, 2, 2e-7, 5e-8 |
>
>
> We observed that The results in Table D.2, D.7, and D.8 showed little difference in view of prediction performance.
> Moreover, for some datasets, we found that setting lambda and alpha to their default values of 0 and fine-tuning the number of trees, learning rate, and max depth through grid search yielded better results.
>
> **W3.**
>
> We have not only added the code as supplementary material, but also included explanations of the data preprocessing process for NAM, NBM, ANOVA-NODE, and NODE-GAM in Section 3 and Appendix C.2.
> Additionally, we have added an explanation of the challenges encountered during the optimization of ANOVA-NODE in Section 3.

---

> ### Author Response · Authors · 2024-11-26
>
> **W4.**
>
> The results in Table D.4 are from a Decision Tree model trained with the default hyper-parameters set in the sklearn package.
> Additionally, we turned hyper-parameter f the Decision Tree using the Optuna package in Python.
> We applied Optuna to the following hyper-parameter range.
>
> $\bullet$ Range of max depth = [2 ,12]
>
> $\bullet$ Range of min_samples_leaf = [2,10]
>
> $\bullet$ Range of min_samples_split = [2,10]
>
> $\bullet$ Range of max_leaf_nodes = [2,10]
>
>
> Table D.9 presents the results of prediction performance in turned decision tree by Optuna package in Python.
> Additionally, the results obtained from the Decision Tree could not be included in the table of results for other baseline models in the main text, as doing so would exceed the paper's page limit.
> We have replaced the results in Appendix I with the tuned results.
>
> **Table D.9. Results of prediction performance in decision tree on the optimized hyper-parameters by the Optuna python package.**
> | Dataset | decision tree's prediction performance | max depth, min_samples_leaf , min_samples_split ,  max_leaf_nodes |
> |--------------------------------------------|-------|--------|
> | Calhousing | 0.671 (0.02) | 12,3,7,10  |
> | Abalone | 2.396 (0.08)  |4,6,5,10  |
> | Wine |0.811 (0.025) |11,3,6,10  |
> | Online|1.119 (0.26) |2,4,8,2 |
> | FICO | 0.704 (0.008) | 12,6,8,10  |
> | Churn |0.676 (0.031) | 11,6,7,10 |
> | Letter |0.745 (0.008)| 6,6,9,10  |
> | Credit | 0.8899 (0.018)| 9,2,2,10 |
> | Drybean | 0.9748 (0.017) |10,4,3,10  |
>
>
>
> **W6.**
>
> As suggested by the reviewer, we conducted runtime experiments not only on the Abalone dataset, but also on the Calhousing and Online datasets, which can be found in Table D.10.
> We observed that the runtime of NBM-NODE, which improves the scalability of ANOVA-NODE, is the shortest.
> The experimental results for runtime have been added in Appendix J of the revision.
>
> **Table D.10. Results of runtime in each model on Abalone, Calhousing, and Online dataset.**
> | Dataset | Size of dataset | \# of features  | NA$^{1}$M | NB$^{1}$M | ANOVA-N$^{1}$ODE | NBM-N$^{1}$ODE |
> |--------------------------------------------|-------|--------|---|----|---|---|
> | Abalone | 4K | 10| 6.6 sec | 3.0 sec | 4.2 sec | 1.5 sec |
> | Calhousing | 21K | 8 | 14.1 sec | 4.1 sec | 9.7 sec | 3.5 sec |
> |Online | 40K | 58 | 68 sec | 15.6 sec | 70 sec | 9.8 sec |
>
>
> **W7.**
>
> Unlike other models (such as Deep Neural Networks), in explainable models based on the Functional ANOVA model, which estimate components and provide interpretations using the estimated components, stability is crucially important.
> [1] discussed the stability of component estimation in Section 4.6 and compared the stability of NA$^{1}$M and NB$^{1}$M on Calhousing dataset.
>
>
> **References**
>
> [1] Radenovic, Filip, Abhimanyu Dubey, and Dhruv Mahajan. "Neural basis models for interpretability." Advances in Neural Information Processing Systems 35 (2022): 8414-8426.

---

> > ### Comment · Reviewer_yGHL · 2024-11-26
> > **Thank you.**
> >
> > Thank you for the additional response. It has clarified some of my concerns. However, I remain critical, primarily regarding the following points:
> >
> > **W2**: The hyperparameter grids have improved significantly. However, I have concerns about the results. The authors state that there were only minor changes (which can certainly occur) and that “setting lambda and alpha to their default values of 0 and fine-tuning the number of trees, learning rate, and max depth through grid search yielded better results.” These findings appear to contradict much of the existing literature, which raises questions about the search procedure. Could the authors elaborate on how the performance evaluation and hyperparameter optimization were conducted? Specifically, was cross-validation or repeated train-test splits used? This information seems to be missing from the paper (as far as I can tell) and should be included. Additionally, I could not find any details on how (categorical) features were encoded.
> >
> > Minor Comments:
> > * **W1**: I appreciate the addition of the figure to the introduction, as it effectively supports the motivation and claims. Including a brief description of the figure would further enhance its utility.
> > * **W4**: I understand the constraints imposed by space requirements. However, adding ANOVA-NODE to the table with the decision tree results in the appendix would provide a helpful direct comparison.

---

> > > ### Author Response · Authors · 2024-11-26
> > >
> > > We appreciate your additional comments and the responses to the additional concerns are as follows.
> > >
> > >
> > > **W2.**
> > >
> > > We mentioned that we observed better performance **for some datasets** (such as Online and Wine dataset), but we did not state that it showed better performance in most cases.
> > > In all our experiments, dataset was split into Train/Validation/Test datasets with a ratio of 0.7/0.1/0.2.
> > > We selected the hyper-parameters based on the prediction performance on the validation dataset for the model trained on the train data, and then evaluated the model's performance on the test data using the chosen hyper-parameters.
> > > Additionally, all categorical features were encoded using one-hot encoding.
> > >
> > >
> > > **W1.**
> > >
> > > As suggested by the reviewer, we add the description for Figure 1 in the revision.
> > >
> > > **W4.**
> > >
> > > As suggested by the reviewer, we add the results of prediction performance of ANOVA-NODE to Appendix I of the revision.

---

> > > > ### Comment · Reviewer_yGHL · 2024-11-26
> > > >
> > > > Do I understand correctly that no repeated evaluation or cross-validation was performed to measure the performance of the methods during your evaluation?

---

> > > > > ### Author Response · Authors · 2024-11-26
> > > > >
> > > > > We missed one point in our response for **W2**.
> > > > > Apologies for that.
> > > > > All experiments conducted in the paper are based on repeated evaluation.
> > > > >
> > > > > To clarify, we randomly split the train, validation and test data into the ratio 70/10/20 and evaluated its performance on the validation dataset using the model trained on the train dataset.
> > > > > We repeated this process 10 times with randomly split data, resulting in 10 prediction performance values for the validation datasets.
> > > > > Then, we selected the optimal hyper-parameters based on the average of the prediction performance values for the validation datasets.
> > > > >
> > > > > Finally, with the optimal hyper-parameters selected earlier, we fixed the model's hyper-parameters and used the 10 train-test dataset pairs obtained from the previous data splitting to train the model on the train datasets and evaluate its performance on the test datasets which corresponding to the train dataset.
> > > > >
> > > > > In our paper, all experiments results of prediction performance including Table D.2, D.3, D.4, D7, D8, and D7 presents the average of prediction performance values for 10 trials.

---

> > > > > > ### Comment · Reviewer_yGHL · 2024-11-26
> > > > > >
> > > > > > Thank you for the clarification - this addresses my concerns to a significant extent. I would recommend including this information (regarding the evaluation protocol and feature encoding) in the appendix for completeness. While I remain somewhat skeptical and unconvinced about the overall impact of the paper, I appreciate the effort the authors have put into addressing the feedback and acknowledge that the paper has improved. Accordingly, I am raising my score to 5.

---

> ### Author Response · Authors · 2024-11-27
>
> We are grateful for your feedbacks, which have greatly enriched our work.
> As suggested by the reviewer, we  have been provided more detailed information on the evaluation protocol and data preprocessing in Appendix C.2.
>
> **Overall impact of paper**
>
> Existing explainable models based on the Functional ANOVA model (e.g., NAM(Agarwal et al., 2021), NBM(Radenovic et al., 2022), etc.) provide interpretations using the estimated components.
> However, if the components are not identifiable, there are multiple interpretation for a given function.
> For example,
>
> $f(x_{1},x_{2}) = f_{1}(x_{1}) + f_{2}(x_{2}) + f_{1,2}(x_{1},x_{2})$   $\quad \cdots (1)$
>
> where $f_{1}(x_{1}) = x_{1}, f_{2}(x_{2}) = x_{2} , f_{1,2}(x_{1},x_{2}) = x_{1}x_{2}$  can be expressed as
>
> $f(x_{1},x_{2}) = f_{1}^{\*}(x_{1}) + f_{2}^{\*}(x_{2}) + f_{1,2}^{\*}(x_{1},x_{2})$  $\quad \cdots (2)$
>
> where  $f_{1}^{\*}(x_{1}) = -x_{1}, f_{2}^{\*}(x_{2}) = x_{2} , f_{1,2}^{\*}(x_{1},x_{2}) = x_{1}(x_{2}+2)$.
>
> In this case, the interpretations of main effect provided by case (1) and case (2) differs.
> Specifically, in case (1), it is interpreted that the main effect of $x_{1}$ provides a positive contribution to the output as $x_{1}$
> increases, whereas in case (2), it is interpreted as providing a negative contribution.
> In other words, if the components are not identifiable, the interpretations provided by XAI models based on the Functional ANOVA model cannot be trusted.
>
> In this paper, the main contributions of our work are as follows.
>
> **1.** We proposed ANOVA-NODE which used a specially modified version of NODT.
> ANOVA-NODE satisfies **specific conditions to ensure that the Functional ANOVA model we estimate is identifiable**.
> As a result, the interpretations provided by our model are more reliable compared to those of NAM(Agarwal et al., 2021) and NBM(Radenovic et al., 2022).
>
> **2.** Second, we proved a universal approximation theorem for our proposed model (ANOVA-NODE).
>
>
> **References**
>
> [1] Agarwal, Rishabh, et al. "Neural additive models: Interpretable machine learning with neural nets." Advances in neural information processing systems 34 (2021): 4699-4711.
>
> [2] Radenovic, Filip, Abhimanyu Dubey, and Dhruv Mahajan. "Neural basis models for interpretability." Advances in Neural Information Processing Systems 35 (2022): 8414-8426.

---

### Official Review · Reviewer_z8qZ · 2024-10-27

**Soundness:** 3
**Presentation:** 2
**Contribution:** 2
**Rating:** 6
**Confidence:** 3

**Summary:**

The paper examines G${}^d$AMs with Neural Oblivious Decision Ensembles (NODEs). After presenting general proofs of functional ANOVA (fANOVA) and its connection to SHAP values, the authors propose a novel architecture to enable a fANOVA-style framework for NODEs. Experiments demonstrate improved stability of this method over alternative approaches, along with competitive predictive performance.

**Strengths:**

- **Clarity**: The mathematical and technical exposition is generally precise and informative.
- **Experiments**: The empirical experiments offer valuable insights into the approach's strengths and advantages.

**Weaknesses:**

## Major Comments

- **Novelty/Contribution [NC]**: It is unclear what aspects of the paper are novel. Specifically:
    - **[NC1]**: Theorems 3.1 and 3.2 address functional ANOVA itself without focusing on NODEs and, if I understand correctly, are established results. Theorem 3.1 is a direct consequence of the sum-to-zero condition (Hooker, 2007), and Theorem 3.2 can be found, for example, in Herren and Hand (2022).
    - **[NC2]**: How does ANOVA-NODE differ in terms of computation from the approach proposed by Lengerich et al. (2020) or from the NODE-GAM method (including purification) presented by Chang et al. (2022), which includes fANOVA in Appendix D?
    - **[NC3]**:
        1. Since Theorem 3.3 addresses expressivity, why is the sum-to-zero constraint necessary? As ANOVA-NODE serves as a functional reparametrization of NODE-GAM, Theorem 3.3 should apply equally to NODE-GAMs of order $d$.
        2. If the optimal function is a GA${}^d$M, which naturally allows for an ANOVA decomposition (Wood, 2006), and efficient approximation methods for order-$d$ GAMs exist (Rügamer, 2024), what practical benefits do ANOVA-NODEs offer?
        3. The authors write as a contribution: "We prove the universal approximation property in that ANOVA-NODE can approximate any continuous function up to arbitrary precision" but not all continuous functions are Lipschitz, which is required.

- **Related Literature [R]**: An extended discussion of related literature would enhance readability and clarify the novelty of this work, particularly concerning the points above. See *References* for a starting point.

- **Experiments/Presentation [E]**:
    - **[E1]**: The finding that enforcing functional orthogonality consistently improves prediction performance is unexpected. Literature suggests that enforcing this constraint during training can sometimes reduce predictive accuracy. In contrast, post-hoc identifiability approaches (as e.g. discussed by the authors in the Appendix) do not influence the optimization while constituting a simpler optimization problem.
    - **[E2]**: Including GAMs as a benchmark could make the results more informative.
    - **[E3]**: A plot comparing stability to prediction performance could help explore any potential relationship between these metrics.
    - **[E4]**: If the ANOVA is functioning as expected, an analysis of variance (across all orders $d$ to show how much of the variation in the outcome can be explained by every complexity level) would add value to the results.

- **Presentation [P]**:
    - **[P1]**: (Related to NC) If the **NC** points are valid, restructuring the paper to focus primarily on ANOVA-NODE and its properties may enhance clarity.
    - **[P2]**: The motivation in Sec. 3 reads awkwardly. It states, "An unsolved but important problem in neural functional ANOVA algorithms is the identifiability of each component" followed by "A simple remedy... is to impose constraints" and then the sum-to-zero constraint, which "is neither a unique nor optimal identifiability condition." So eventually, the initially stated problem remains, doesn't it?
    - **[P3]**: Adding axis labels and facet titles (especially in Figures 2 and 3) would significantly improve readability. Additionally, Figure 2’s caption mentions **stability scores**, while the text refers to **similarity scores**.

## Minor Comments

- l.109: While the GA${}^d$M notation is useful, the original GAM literature also uses the term "GAM" to refer to models incorporating higher-order (tensor-product) splines, beyond main effects (see Wood, 2006).
- l.161: The example in Section 3.1 on functional identifiability could be more insightful; it currently does not address functional or parameter identifiability. Consider explicitly expressing $f$ as a sum of functions $f_1$, $f_2$, and $f_{12}$.
- l.285: The training discussion in Section 3.2 could be more detailed, especially regarding challenges with optimization and solutions.
- l.302: Consider using a different symbol than $\mathcal{S}$ for the similarity measure, as $S$ is already used for other notations.
- Table 1: Both Table 1 and Figure 2 discuss stability but show different value ranges. It may help to clarify that Table 1 shows overall stability, while Figure 2 shows individual scores. Also maybe more clearly highlight that the scores in Table 1 are normalized.
- Figures 3, 4, and those in the Appendix: The text in these figures is too small. For Figure 3, consider using individual y-axis scales or possibly a log scale, as the differences between models are currently difficult to discern.
- Appendix: It states, "NODE-GAM is not strictly a GAM, but a model that approximates a GAM." This might also apply to ANOVA-NODEs. In general --- as also stated in **NC** above --- more discussion about related methods and their differences to the authors' approach is required (and ideally in the main text)
- Manuscript: Consider proofreading the main text and Appendix to catch typos and grammar issues (e.g., l.2064 "AVNOA-NODE").

## References

- Blondel et al., 2016: https://papers.nips.cc/paper_files/paper/2016/file/158fc2ddd52ec2cf54d3c161f2dd6517-Paper.pdf
- Chang et al., 2022: https://arxiv.org/pdf/2106.01613
- Herren and Hand, 2022: https://arxiv.org/pdf/2208.09970
- Hiabu et al., 2023: https://proceedings.mlr.press/v206/hiabu23a/hiabu23a.pdf
- Hu et al., 2022: https://arxiv.org/pdf/2207.06950
- Hu et al., 2023: https://arxiv.org/pdf/2305.15670
- Köhler et al., 2024: https://arxiv.org/abs/2407.18650
- Kneib et al., 2019: https://link.springer.com/article/10.1007/s11749-019-00631-z
- Lengerich et al., 2020: https://proceedings.mlr.press/v108/lengerich20a/lengerich20a.pdf
- Limmer et al., 2024: https://arxiv.org/abs/2408.12319v1
- Rugamer, 2023: https://proceedings.mlr.press/v202/rugamer23a/rugamer23a.pdf
- Rügamer, 2024: https://proceedings.mlr.press/v238/ruegamer24a/ruegamer24a.pdf
- Wood, 2006: https://onlinelibrary.wiley.com/doi/abs/10.1111/j.1541-0420.2006.00574.x
- Wood, 2017: https://www.taylorfrancis.com/books/mono/10.1201/9781315370279/generalized-additive-models-simon-wood

**Questions:**

- It would be great if authors could comment on the mentioned weaknesses and whether I missed or misunderstood anything.
- Would it make sense to use an ANOVA kernel as in Blondel et al., 2016 to improve the scaling of ANOVA-N${}^d$ODES?
- Could a factorization approach as in Rügamer, 2024 incorporated to improve scalability?

---

> ### Author Response · Authors · 2024-11-25
>
> We appreciate your feedback on our paper and have made every effort to address your insightful questions. We have uploaded the revised version of the paper, reflecting the reviewer's feedback.
>
> **Weakness N1.**
>
> We agree with the reviewer’s comment, and therefore, Theorem 3.1 and Theorem 3.2 have been changed to proposition 3.1 and 3.2.
> Additionally, references have been added.
> Note that the main contribution of our paper is not Theorem 3.1 and Theorem 3.2, but rather the proposal of interpretation XAI model that can be trained using gradient-based optimization methods while satisfying the sum-to-zero condition for the functioanl ANOVA model, along with the associated universal approximation theorem.
>
> **Weakness N2.**
>
> [1] proposed a post-processing method to satisfy the sum-to-zero condition for piecewise-constant function, whereas ANOVA-NODE is a model that is learned while inherently satisfying the sum-to-zero condition.
>
> When applied to a given dataset of size n, the computation order of performing post-processing for a given point **x** is $\mathcal{O}(dn^{d-1})$, which is very demanding.
> Furthermore, performing post-processing requires storing a dataset, which causes memory efficiency issues.
> Therefore, interpretation, either locally or globally, is practically infeasible.
>
> To be specific, for a given point **x**, the computation order of calculating ANOVA-SHAP $\phi_{j}(f,$**x**$)$ is $\mathcal{O}(p^{d-1}dn^{d-1})$.
> Therefore, not only is it impossible to draw dependence plots, but performing local interpretation using ANOVA-SHAP is also infeasible.
> The method for post-processing in GA$^{d}$M and the process for calculating its computational order have been added in the Appendix L.
>
> **Weakness NC3.1.**
>
> A universal approximation theorem also may exists for NODE-GAM, but the most important aspect of Theorem 3.3 is that the universal approximation theorem holds for ANOVA-NODE using ANOVA-NODTs that satisfy the sum-to-zero condition.
> Additionally, since the function space of ANOVA-NODE is smaller than that of NODE-GAM, proving an approximation theorem for ANOVA-NODE is a much more difficult problem than for NODE-GAM.
>
> **Weakness NC3.2.**
>
> The $entmax({\textbf{x} -  b \over \gamma})$ in ANOVA-NODE corresponds to the basis function $B(\textbf{x})$ in spline basis representation of [2].
> While AHOFM ([2]) does not train the basis function, ANOVA-NODE can be considered to learn the basis by training $b , \gamma$.
>
> Also, we acknowledge that if all higher-order interactions are considered, the method proposed by [2] is more efficient in terms of scalability compared to ANOVA-NODE.
> However, the experiments in the paper show that considering interactions up to the second order is sufficient for real data.
> Moreover, when all higher-order interactions are considered in the functional ANOVA model, it is likely to be overfitted.
>
> To examine the relationship between considering all higher-order interactions and model overfitting, we trained ANOVA-N$^{3}$ODE and ANOVA-N$^{4}$ODE on the Calhousing dataset, and the results are shown in Table 3.
> We observed that considering all interactions of third-order or higher led to overfitting, resulting in a decrease in prediction performance.
>
> **Table C.1. Results of prediction performance in ANOVA-N$^{2}$ODE, ANOVA-N$^{3}$ODE, and ANOVA-N$^{4}$ODE.**
> | Model  | ANOVA-N$^{2}$ODE | ANOVA-N$^{3}$ODE | ANOVA-N$^{4}$ODE |
> |--------------------------------------------|-------|--------|--------|
> | RMSE (std)  | 0.512 (0.01) | 0.542 (0.02)  | 0.572 (0.02)  |
>
> Therefore, rather than considering all higher-order interactions, we should focus on the important ones.
> As demonstrated in the high-dimensional data experiments in our paper, this can be done by pre-selecting interactions using an interaction selection algorithm like Neural Interaction Detection ([3]).
> When these pre-selected interactions are applied to ANOVA-NODE, there is little difference in scalability compared to method in [2].
>
> **Weakness NC3.3.**
>
> We thank the reviewer for pointing out the errors. The errors have been corrected.
>
> **Weakness E.1.**
>
> In Appendix I, we confirmed that the prediction performance is better when sum-to-zero conditions are not imposed, and this was discussed.
>
> **Weakness E.2.**
>
> As suggested by the reviewer, we conducted additional experiments GAMs by using pygam python package, denote it as py-GAM.
> py-GA$^{d}M estimates each component of GA$^{d}$ by a cubic spline model.
> Table C.2 shows the average and standard deviation of the RMSE from 10 trials on the Abalone dataset.
>
> **Table C.2. Results of prediction performance in ANOVA-N$^{1}$ODE, ANOVA-N$^{2}$ODE, and ANOVA-N$^{3}$ODE.**
> | Model  | py-GA$^{1}$M | py-GA$^{2}$M | ANOVA-N$^{1}$ODE |  ANOVA-N$^{2}$ODE |
> |-----------------|-------|--------|--------|--|
> | RMSE (std)  | 2.189 (0.07) | 2.096 (0.09)  | 2.135 (0.09)  | 2.087 (0.08)  |

---

> ### Author Response · Authors · 2024-11-25
>
> **Weakness E.4.**
>
> The sum-to-zero condition is based on the product measure, so $Var(f)= \sum_{S}Var(f_{S})$ does not hold in ANOVA-NODE.
> Therefore, in the paper, instead of conducting experiments of the analysis of variance,
> we conducted experiments to analysis the importance components using $l1$ norm, and the experiments results on Calhousing dataset are presented in Section E.1.1.
>
> **Weakness P.1.**
>
> As suggested by the reviewer, we restructured the paper and uploaded it.
>
> **Weakness P.2.**
>
> ``is neither a unique nor optimal identifiability condition" means that there can be various types of constraints, other than the sum-to-zero condition, that ensure the identifiability of the functional ANOVA decomposition.
> Because the sum-to-zero condition also ensures the identifiability of components, the initially stated problem is solved.
>
> **Weakness P.3.**
>
> As suggested by the reviewer, we added axis labels in Figure 2 and changed the term 'similarity measure' to 'stability score'.
>
> **Minor l.161.**
>
> As suggested by the reviewer, we explicitly express the $f$ into $f_{1},f_{2},f_{1,2}$.
>
> **Minor l.285.**
>
> We have added a more detailed description of the challenges in optimization to the data preprocessing paragraph in the revision.
>
> **Minor l.302.**
>
> As suggested by the reviewer, we have replaced all $\mathcal{S}$ symbol with the $\mathcal{SC}$ symbol.
>
> **Minor Table 1: Both Table 1 and Figure 2 discuss stability but show different value ranges.**
>
> We add the results of unnormalized stability scores in Appendix G.2 of the revision.
>
> **Minor Figures 3, 4, and those in the Appendix.**
>
> Following the reviewer's suggestion, we have increased the text size in all the figures.
>
> **Minor Figures 3, 4, and those in the Appendix.**
>
> As suggested by the reviewer, we have added a more detailed comparison with NODE-GAM and included it in Appendix K of the revision.
>
> **Minor Manuscript:**
>
> The typo have been corrected.
>
> **Question 1.**
>
> It has already been addressed in the response to the Weakness section.
>
> **Question 2. Would it make sense to use an ANOVA kernel as in Blondel et al., 2016 to improve the scaling of ANOVA-N
> ODES?**
> **Question 3. Could a factorization approach as in Rügamer, 2024 incorporated to improve scalability?**
>
> Since ANOVA-NODT is also a model represented by the spline basis representation of equation (2) in [2](entmax can be viewed as basis) factorization method can be used to improve the scalability of ANOVA-NODE.
> Exploring the use of the factorization approach to improve the scalability of ANOVA-NODE seems interesting.
>
> Note that, in our paper, we proposed NBM-NODE which is an extended model of ANOVA-NODE, proposed in Appendix G.2 of this paper, which improves the scalability of ANOVA-NODE by utilizing basis functions, similar to the approach used in NBM.
> Note that in NBM-NODE, the number of ANOVA-NODT for basis does not depend on the number of components
>
> We conducted additional experiments to assess the improvement in scalability.
> We consider NA$^{1}$M, which has 3 hidden layers with 16, 16, and 8 units; 10 basis DNNs for NB$^{1}$M, which has 3 hidden layers with 32, 16, and 16 units; 10 trees for each component in ANOVA-N$^{1}$ODE; and 10 basis functions in NBM-N$^{1}$ODE.
>
> **Table C.3 Results of computaitonal complexity**
> | Model  | NA$^{1}$M| NB$^{1}$M| ANOVA-N$^{1}$ODE | NBM-N$^{1}$ODE|
> |---------------|-------|--------|--------|-------|
> | Run time| 6.6 sec | 3.0 sec  | 4.2 sec  | 1.5 sec |
> | \# of weight parameters | 4,490 | 1,113  | 670  | 170 |
>
> Table C.3 presents the runtime for each model when trained on a Abalone dataset and 100 epochs.
> Our computational environment consists of RTX 3090 and RTX 4090.
> We observed an improvement in scalability when we extended ANOVA-NODE using basis functions.
>
>
> **References**
>
> [1] Lengerich, Benjamin, et al. "Purifying interaction effects with the functional anova: An efficient algorithm for recovering identifiable additive models." International Conference on Artificial Intelligence and Statistics. PMLR, 2020.
>
> [2] Rügamer, David. "Scalable Higher-Order Tensor Product Spline Models." International Conference on Artificial Intelligence and Statistics. PMLR, 2024.
>
> [3] Tsang, Michael, Dehua Cheng, and Yan Liu. "Detecting statistical interactions from neural network weights." arXiv preprint arXiv:1705.04977 (2017).

---

> ### Comment · Reviewer_z8qZ · 2024-11-25
> **Preliminary Response to the Rebuttal**
>
> Dear Authors,
>
> Thank you for providing such detailed feedback. Due to time constraints, I have only skimmed parts of your response. I will follow up on some of your answers hopefully by the end of the day (AoE). However, what I have read so far already addresses many of my concerns. I will raise my score at least to 5, potentially higher after reading your response more carefully.
>
> One small request: Could the authors maybe fix the formatting of their post (for example, the $\textbf$ is not rendering correctly in markdown, I suggest using `**[TEXT]**`) to improve its readability? **Edit**: Thanks, looks good now.

---

> > ### Author Response · Authors · 2024-11-26
> >
> > We appreciate your valuable comments.
> > We have modified \textbf{text} using ** as per the reviewer's request.
> > If there are any additional requests, please let me know, and we will do our best to incorporate them.

---

> > > ### Comment · Reviewer_z8qZ · 2024-11-26
> > >
> > > Dear Authors,
> > >
> > > Thank you for your detailed response and for providing new results. Many of your points were helpful in clarifying my understanding of the paper.
> > >
> > > However, I still do not feel I fully understand:
> > >
> > > - what the paper's novelty is, and
> > > - how it differs from existing literature.
> > >
> > > To address this, I would appreciate it if the authors could add a discussion of related literature to the Appendix, particularly elaborating on the differences from the other ANOVA approaches mentioned in my initial comment, as well as the related techniques cited in **[NC1-3]** and **[R]**.
> > >
> > > For now, I am raising my score to 6.

---

> ### Author Response · Authors · 2024-11-27
>
> Thank you for your valuable feedback. It has greatly enriched our work.
> As suggested by the reviewer, we will add the discussion of related literature to Appendix and upload it by tomorrow.
>
> **Question 4. what the paper's novelty**
>
> ANOVA-NODE uses ANOVA-NODT, a modified version of Neural Oblivious Decision Tree (NODT) that satisfies the sum-to-zero condition.
> Of course, there may be other architectures that can be modified to satisfy the sum-to-zero condition, but it cannot be guaranteed that these modified architectures will increase the stability of component estimation practically.
> In other words, while the modified model ensures the uniqueness of the functional ANOVA decomposition theoretically, the components estimated by the modified model that satisfies the sum-to-zero condition may vary for each randomly sampled training dataset.
>
> For example, we discussed the post-processing for the sum-to-zero condition in Appendix M of the paper (revision version).
> We observed that while the neural network-based models, NAM(Agarwal et al., 2021) and NBM(Radenovic et al., 2022), did not show improved stability in component estimation even after post-processing, **the stability was improved in the case of GAM-NODE, which estimates components using NODT**.
>
> Therefore, we believe that the stability of  component estimation in ANOVA-NODE is due to the sum-to-zero condition and the characteristics of the NODT model structure.
> The relationship between the sum-to-zero condition and NODT will be explored in future work.
>
>
> **Question 5. how it differs from existing literature**
>
> The model (Rügamer, 2024 ) mentioned by the reviewer is a highly efficient algorithm for estimating the functional ANOVA model, but it does not satisfy the sum-to-zero condition or other conditions for identifiability.
>
> Therefore, this model may provide multiple interpretations for a given data point **x**.
> For example,
>
> $f(x_{1},x_{2}) = f_{1}(x_{1}) + f_{2}(x_{2}) + f_{1,2}(x_{1},x_{2})$   $\quad \cdots (1)$
>
> where $f_{1}(x_{1}) = x_{1}, f_{2}(x_{2}) = x_{2} , f_{1,2}(x_{1},x_{2}) = x_{1}x_{2}$  can be expressed as
>
> $f(x_{1},x_{2}) = f_{1}^{\*}(x_{1}) + f_{2}^{\*}(x_{2}) + f_{1,2}^{\*}(x_{1},x_{2})$  $\quad \cdots (2)$
>
> where  $f_{1}^{\*}(x_{1}) = -x_{1}, f_{2}^{\*}(x_{2}) = x_{2} , f_{1,2}^{\*}(x_{1},x_{2}) = x_{1}(x_{2}+2)$.
>
> In this case, the interpretations of main effect provided by case (1) and case (2) differs.
> Specifically, in case (1), it is interpreted that the main effect of $x_{1}$ provides a positive contribution to the output as $x_{1}$
> increases, whereas in case (2), it is interpreted as providing a negative contribution.
> In other words, if the components are not identifiable, the interpretations provided by models based on the Functional ANOVA model cannot be trusted.
>
> To solve this issue, post-processing would need to be performed, but due to the high computational complexity, it is practically infeasible.
>
> Furthermore, as mentioned in the response to **Question 4**, even if these models (Blondel et al., 2016 and Rügamer, 2024) are modified to satisfy the sum-to-zero condition, there is no guarantee that component estimation will be stable for each randomly sampled training dataset.
>
>
>
> **References**
>
> [1] Agarwal, Rishabh, et al. "Neural additive models: Interpretable machine learning with neural nets." Advances in neural information processing systems 34 (2021): 4699-4711.
>
> [2] Radenovic, Filip, Abhimanyu Dubey, and Dhruv Mahajan. "Neural basis models for interpretability." Advances in Neural Information Processing Systems 35 (2022): 8414-8426.
>
> [3] Rügamer, David. "Scalable Higher-Order Tensor Product Spline Models." International Conference on Artificial Intelligence and Statistics. PMLR, 2024.
>
> [4] Blondel, Mathieu, et al. "Higher-order factorization machines." Advances in neural information processing systems 29 (2016).

---

> > ### Comment · Reviewer_z8qZ · 2024-11-27
> >
> > Dear Authors,
> >
> > Thank you for the detailed response. Just to clarify, when you say:
> >
> > > there is no guarantee that component estimation will be stable for each randomly sampled training dataset.
> >
> > Do you mean that the model estimation exhibits high variance?
> >
> > Additionally, a quick follow-up question: Wood (2006, 2017 section 5.6.3) describes the construction of multivariate tensor product splines from univariate bases by marginalizing out the respective lower-dimensional bases and, IIRC, allow the construction of a fANOVA model (as, e.g., in Kneib et al., 2019). Is this the approach you’re using in the pyGAM examples you provided? And would employing such a basis construction address the identifiability issue you mentioned in your response to Question 5?

---

> ### Author Response · Authors · 2024-11-28
>
> Thank you for your additional questions.
> We have done our best to respond to your thoughtful questions
>
> **Question 6. Do you mean that the model estimation exhibits high variance?**
>
> We explain the points we mentioned earlier in more detail.
> For a randomly sampled training dataset, we can obtain the components $f_{S}$ estimated by the model, and by repeating this process $k$ times, we obtain $(\hat{f}^{i}_{S} , i=1,...,k )$ for $S \subseteq [p]$.
>
> ​In this case, for any $S \subseteq [p]$, if the components  $(\hat{f}^{i}_{S} , i=1,...,k )$ are estimated similarly, we say that the model is estimating the component stably.
>
> **Question 7. quick follow-up question.**
>
> Section 5.6.2 of Wood (2017) describes the SSANOVA (Gu, Chong, 2014).
> SSANOVA estimates the components in functional ANOVA model by using spline models which satisfy the sum-to-zero condition which we used.
> (They referred to the sum-to-zero condition as the averaging operator.)
>
> As far as I know, py-GAM estimates the components of a Functional ANOVA model using a spline model without identifiability condition (e.g., sum-to-zero condition).
>
> Using the averaging operator of SSANOVA, it seems possible to apply it to AHOFM( Rügamer, 2024), but further research is needed on this.
>
> **References**
>
> [1] Gu, Chong. "Smoothing spline ANOVA models: R package gss." Journal of Statistical Software 58 (2014): 1-25.

---

### Official Review · Reviewer_jRHP · 2024-10-29

**Soundness:** 3
**Presentation:** 3
**Contribution:** 2
**Rating:** 3
**Confidence:** 4

**Summary:**

The paper introduces a groundbreaking model that enhances the interpretability of machine learning models by addressing the instability in component estimation within the functional ANOVA framework. ANOVA-NODE ensures a unique decomposition of high-dimensional functions into lower-dimensional components, thereby providing a stable and interpretable model.

The authors demonstrate the model's ability to approximate smooth functions and establish a theoretical connection with SHAP values, facilitating the calculation of individual contributions to model predictions. Empirical experiments on various datasets verify the model's superior stability in component estimation and its competitive prediction performance compared to existing methods.

**Strengths:**

1. Writing: The structure of this work is well-crafted and stylistically sound, which could enhance the comprehension of the paper's modeling and computational aspects. However, there exist minor spelling and grammatical errors.
2. Theory: ANOVA-NODE can approximate smooth functions effectively and have established an interesting relationship between ANOVA-NODE and SHAP, a well-known interpretable method. This theoretical groundwork is crucial for the model's credibility, but with limited novelty compared to the existing representation theorem by Kolmogorov–Arnold. The proof seems solid but I have not checked carefully.
3. Experiments: Under several empirical validations, ANOVA-NODE offers more stable estimation of components compared to existing models, such as NAM and NBM, under variations in training data and initial model parameters. This is a valuable practical validation of the model's stability and interpretability. An interesting application to high-dimensional data uncovers its prediction performance and component stability, which is an important step towards real-world applicability.
4. Interesting extension: A new approach, ANOVA-SHAP, is developed to calculate SHAP values from the estimated ANOVA-NODE model, offering a computationally efficient alternative to existing SHAP calculation methods.

**Weaknesses:**

1. Wrong definitions for (generalized) additive models. As classical statistical models, additive models and their generalized version have been widely investigated from theoretical or empirical views. In line 111 (Page 3), the formulation indeed is the additive model instead of GAM, where GAM may involve a link mapping function g(*) like $f(x) = g(\sum_i f_i(x_i))$.

2.	Novelty in modeling. The contribution of integrating neural networks in approximating component functions for ANOVA seems limited, compared to some existing work like Kolmogorov–Arnold network. There may require more discussions and empirical comparisons with some state-of-the-art additive models, neural additive models and recent ANOVA series.

3.	Missing crucial mathematical background. The Kolmogorov–Arnold theorem is necessary for demonstrating the rationality of summing component functions with individual feature. Thus it is kindly suggested to add comments to help the readers to understand the background and contributions.

4.	Theory. The sum2zero condition is commonly used for ANOVA modeling. However, there may exist some limitations. First, as stated in Lime178 (Page 4), this constraint may lead to nonunique or nonoptimal results. Besides, it is necessary to state the key assumptions before presenting these theorems to avoid misunderstandings, as some assumptions or conditions are merely present in the appendix.

Furthermore, to better highlight the contributions, it is kindly suggested to summarize the challenges among the proof. Especially for Theorem 3.3, only $d=1$, sigmoid induced auxiliary function and bounded conditions are presented. It would be necessary to analyze more scenarios to support Theorem 3.3, like d=2 or more.

What is the meaning of $K_S S$ in Line 282?

5. Optimization & Experiments. The dimensions of these used synthetic or real datasets are not high enough, indeed. Although such additive modeling scheme is declared to relieve the curse of dimensionality, individual modeling w.r.t. each input feature inevitably reduces prediction accuracy and often leads to non-convex target loss. Thus I wonder if the ANOVA-NODE enjoys some properties like smoothness or convexity?

6. Moreover, how to determine the threshold $K_S$?
Is it possible to uncover the relationship between prediction accuracy and the interpretability based on the theoretical or empirical perspectives?

**Questions:**

Please see the comments in Weakness.

---

> ### Author Response · Authors · 2024-11-25
>
> We appreciate your feedback on our paper and have made every effort to address your insightful questions. We have uploaded the revised version of the paper, reflecting the reviewer's feedback.
>
> **Weakenss 1. Wrong definitions for (generalized) additive models.**
>
> In our paper $f$ can be a function of any quantity.
> For the regression model, we could have $f($**x**$)=\mathbb{E}(Y|$**X**$=$**x**$),$ and for the logistic regression, it could be $f($**x**$)= {\rm logit} \Pr(Y=1|$**X**$=$**x**$)).$
> More generally, in the generalized linear model, we could have $f($**x**$)=g(\mathbb{E}(Y|$**X**=**x**$))$ for a certain link function $g.$
> In the revision, we added this explanation right after introducing $f.$
>
>
> **Weakness 2. Novelty in modeling and 3. Missing crucial mathematical background.**
>
> As far as I know, KAN is an extension of projection pursuit regression, takes the sum of univariate functions as input to a activation function, and is an interpretable model.
>
> The differences between ANOVA-NODE and KAN are as follows.
>
> **2.1.** KAN captures interactions through the activation function, whereas the functional ANOVA model explicitly represents interactions with specific interaction terms.
>
> **2.2.** The $entmax({\textbf{x} - b \over \gamma })$ in ANOVA-NODE corresponds to the basis function $B(\textbf{x})$ in the activation function of KAN.
> While KAN does not train the basis function, ANOVA-NODE can be considered to learn the basis by learning $b , \gamma$.
>
> **2.3.** In KAN, the closed-form functional relationship between the output and components is not available, whereas in ANOVA-NODE, it can be determined.
>
> **2.4.** While the KAN model does not guarantee identifiability, ANOVA-NODE satisfies identifiability, ensuring the stability of the interpretations it provides.
>
> Therefore, these two models are different, and both are interpretable and useful.
> We left the comparison between these two models as a future work.
>
>
> **Weakness 3. Theory.**
>
> As mentioned in Line 178 (page 4), the sum-to-zero condition ensure the uniqueness of the functional ANOVA decomposition, thereby providing a stable interpretation.
> Note that, there are multiple conditions that ensure such uniqueness, and the sum-to-zero condition is just one of them.
> Also, theoretically, the global minimum does not change even with the sum-to-zero condition.
> However, since the model is trained by using a local minimum, the performance of the model can vary depending on the identifiability condition.
> Therefore, we do not know which condition provides the optimal results.
>
> As suggested by the reviewer, we summarize the challenges in the proof of Theorem 3.3 and add the specific assumptions to the theorem.
> The proof for the case when the interaction order $d \geq 2$ is very complex to express using mathematical formulas, so it was omitted.
> The most important and challenging part of the proof of Theorem 3.3 is decomposing the ensemble function $f_{\mathcal{E}}$ into ANOVA-NODTs.
> Therefore, we added how the decomposition is performed using a toy example for $d=2$ in the proof of Theorem 3.3.
>
> **Weakness 5. Optimization**  \&  **Experiments.**
>
> As pointed out by the reviewer, individual modeling with respect to each input feature may reduce prediction accuracy.
> However, we demonstrated through the experiments of prediction performance in the paper that considering up to second-order interactions is sufficient to achieve comparable prediction performance.
> Moreover, as far as I know, the target loss of many other models such as deep neural networks is also non-convex.
> The smoothness of ANOVA-NODE can be controlled by imposing constraints on the $\gamma$ parameter in Section 3.3 of the paper.
>
>
> **Weakness 6.**
>
> We did determine the threshold $K_{S}$ by empirical search.
> The experimental results of the prediction performance in ANOVA-NODE as $K_{S}$ increases are provided in Appendix D.1.
>
> Theoretically, as $K_{S}$ increases, ANOVA-NODE can approximate any smooth function well, which can be found in Theorem 3.3.
> Also, choosing the optimal $K_{S}$ value requires research on the convergence rate, which will be addressed as a future work.
>
> Finally, since we estimate each component of the function ANOVA model using an ensemble of $K_{S}$ NODTs, increasing $K_{S}$ does not affect the interpretability of ANOVA-NODE.

---

> > ### Author Response · Authors · 2024-12-01
> >
> > Dear reviewer jRHP,
> >
> > Once again, we sincerely appreciate your comments, which has been extremely helpful for our work.
> > We hope this message does not cause any inconvenience, but as we only have a few days left for discussions, we would like to ask whether our responses have sufficiently addressed your concerns.
> >
> > Thank you.

---

### Official Review · Reviewer_mCTR · 2024-11-04

**Soundness:** 3
**Presentation:** 2
**Contribution:** 3
**Rating:** 6
**Confidence:** 4

**Summary:**

This paper proposes a novel methodology within an interpretable framework. The proposed model belongs to the class of functional ANOVA models, structured as a parallel sum of independent functions for each main effect and higher-order interaction effects. Additionally, the model uses NODE based on Oblivious Decision Trees (ODT) for each effect-specific function. This approach addresses the common identifiability issue in functional ANOVA models. Through experimental design, the paper demonstrates the method's effectiveness in resolving identifiability issues using a similarity measure based on the variance of predictions across multiple functions. Furthermore, the authors reveal a connection between SHAP, a post-hoc explainable AI method, and ANOVA-SHAP, showing that ANOVA-SHAP can serve as an effective explainability technique grounded in the proposed method.

**Strengths:**

1. The proposed model $ANOVA-NODE$ is described clearly and is easy to understand, further it seems technically solid.
2. The contributions of this research to the field are well-defined, and the experimental results strongly support the claims made in the paper.
3. The experiments designed to demonstrate the advantages of the proposed method are well-constructed and effectively highlight the strengths of the approach.

**Weaknesses:**

Weakness 1. Reproducibility issue

In this paper, the description of the proposed method is clear, and the experimental results seem to strongly support the paper’s contributions. Nevertheless, one remaining concern in evaluating this paper is the lack of distributed code or guidelines, which would include research artifacts or data preprocessing steps.

- To address this concern, it would be advisable to release supplementary materials.
- If there are reasons preventing the release of materials including code and guidelines, those reasons should be explained.

Weakness 2.  Inconsistency between subsection title and Content

In subsection 2.3, "Neural Oblivious Decision Ensembles," the paper contains only a description of ODT without any mention of NODE. Did I observe this correctly? If so,

- The paper should provide an explanation of NODE. Alternatively, the subsection title should be changed, and an additional subsection should be dedicated to describing NODE.

Weakness 3. Need for Enhancements in Interpretability

The experimental results in the paper, along with those in the appendix, effectively demonstrate that the proposed method is more stable in terms of interpretability compared to other techniques such as $NAM$ and $NBM$. This stability is also observed in comparisons with second-order functional ANOVA models. However, given that the proposed method is an XAI model, would it be possible to provide interpretable plots (e.g., contour plots) for second-order interaction effects? (I understand that presenting interactions beyond second-order may be challenging.)

- Quantitatively representing the importance of second-order interaction effects compared to main effects could further enhance the interpretability aspect of the proposed method.

- Providing contour plots that illustrate changes in output with respect to variations in input for individual second-order interaction effects could improve explainability.

**Questions:**

Question 1. Is it possible to compare the computational complexity with $NAM$ and $NBM$ ? If so, I am curious about the differences in computational complexity.

Question 2. In terms of training strategy, many GAM-based additive models often first learn lower-order effects and then subsequently learn higher-order interaction effects. If my understanding is correct, it seems that $ANOVA-N^{d}ODE$ learns all orders of effects simultaneously. Would a sequential training strategy allow for more stable learning?

Question 3. Using NID, interactions targeted for learning were selected on a high-dimensional dataset. If we also select second-order or higher interactions on low-dimensional datasets (e.g., CALHOUSING, ABALONE, etc.), it seems that stability could be further improved, as with the previous question. Could we verify if my suggestion is correct?

**Details Of Ethics Concerns:**

I did not find any concerns related to the Flag For Ethics.

---

> ### Author Response · Authors · 2024-11-25
>
> We appreciate your feedback on our paper and have made every effort to address your insightful questions.
> We have uploaded the revised version of the paper, reflecting the reviewer's feedback
>
> **Weakness 1. Reproducibility issue.**
>
> As suggested by the reviewers, we have included the source code of ANOVA-NODE in the Supplementary Material and we add the description of the data preprocessing in Appendix C.2 of the revision.
>
>
> **Weakness 2. Inconsistency between subsection title and Content.**
>
> There was originally an explanation of NODE, but due to the page limit of the
> paper, it was removed. However, the title was not updated accordingly. In the
> revision, we add the description of NODE in subsection 2.3.
>
>
> **Weakness 3. Need for Enhancements in Interpretability.**
>
> The results of importance score for second-order interactions and main effects on Calhousing dataset are provided in Appendix E.1.1 of the revision. As suggested by the reviewer, contour plots for the second-order interactions have been added to Appendix E.1.1 of the revision.
>
>
> **Question 1.**
>
> As suggested by the reviewer, we conducted additional experiments to compare the computational complexity of NAM, NBM, ANOVA-NODE, and NBM-NODE based on the runtime of each model and the number of weight parameters.
> Note that NBM-NODE is an extended model of ANOVA-NODE, proposed in Appendix G.2 of this paper, which improves the scalability of ANOVA-NODE by utilizing basis functions, similar to the approach used in NBM.
>
> We consider NA$^{1}$M, which has 3 hidden layers with 16, 16, and 8 units; 10 basis DNNs for NB$^{1}$M, which has 3 hidden layers with 32, 16, and 16 units; 10 trees for each component in ANOVA-N$^{1}$ODE; and 10 basis functions in NBM-N$^{1}$ODE.
>
> **Table A.1. Results of computational complexity.**
> | Model  | NA$^{1}$M | NB$^{1}$M | ANOVA-N$^{1}$ODE | NBM-N$^{1}$ODE |
> |-----|-------|--------|--------|-------|
> | Run time | 6.6 sec | 3.0 sec  | 4.2 sec  | 1.5 sec |
> | \# of weight parameters | 4,490 | 1,113  | 640  | 174 |
>
> Table A.1 presents the number of weight parameters and runtime for each model when trained on a Abalone dataset with 100 epochs.
> Our computational environment consists of RTX 3090 and RTX 4090.
> We observed that our model has fewer weight parameters and takes less runtime compared to existing models (NAM and NBM).
>
>
> **Question 2.**
>
> As the reviewer understood, ANOVA-NODE learns the effects of all interaction orders simultaneously.
> When lower-order and higher-order interactions are trained sequentially, multicollinearity can obscure estimating the optimal model. That is, sequential learning would not be optimal which can be seen even in a linear model.
> For example, consider the model $Y=\beta_{0}+\beta_{1}X_{1}+\beta_{2}X_{2}+\epsilon$, where there is mulicollinearity between $X_{1}$ and $X_{2}$ (i.e. ${\rm corr}(X_1,X_2) \neq 0$).
>
> We estimate $\beta_{0},\beta_{1}$ and $\beta_{2}$ simultaneously by minimizing the loss, whose estimated coefficients are $\hat{\beta_{0}^{\*}}, \hat{\beta_{1}^{\*} }, \hat{\beta_{2}^{\*} }$.
> Altenatively, we estimate the $\beta_{0},\beta_{1},\beta_{2}$ sequentially by
> estimating $\beta_{0},\beta_{1}$ based on the model $Y=\beta_{0}+\beta_{1}X_{1}+\epsilon_{1}$ to have $\hat{\beta_{0}},\hat{\beta_{1}}$, and then
>  estimating $\beta_{2}$ based on the model $Y-\hat{\beta_{0}}-\hat{\beta_{1}}X_{1} = \beta_{2}X_{2} + \epsilon_{2}$ to have $\hat{\beta_{2}}$.
>
> In this case, $\{\hat{\beta_{0}^{\*}}, \hat{\beta_{1}^{\*}}, \hat{\beta_{2}^{\*}} \}$ and $\{\hat{\beta_{0}}, \hat{\beta_{1}}, \hat{\beta_{2}} \}$ are different and thus $\{\hat{\beta_{0}}, \hat{\beta_{1}}, \hat{\beta_{2}} \}$ is a non-optimal solution.
> As far as we understand, sequential learning is used in practice when simultaneous learning is computationally demanding.
>
>
> **Question 3.**
>
> We analyzed a wine dataset with 11 feature dimensions and a size of 4k, and applied NID to select the top 10 most important second-order interactions.
> Table A.2 shows the performance and stability scores of ANOVA-N$^{2}$ODE with and without NID.
> We observed an increase in stability when applying NID before ANOVA-NODE.
>
> **Table A.2. Results of prediction performance and stability score with and without NID.**
> | Model  | NID + ANOVA-N$^{2}$ODE| ANOVA-N$^{2}$ODE |
> |-----|-------|--------|
> | RMSE $\downarrow$ | 0.710 (0.019) | 0.704 (0.020)  |
> | Stability score $\downarrow$ | 0.027 | 0.049 |

---

> > ### Comment · Reviewer_mCTR · 2024-11-27
> >
> > Thank you for the detailed responses and the additional experiments conducted to address the concerns raised during the review process. I appreciate the effort and commitment the authors have demonstrated to improve the quality of the paper.
> >
> > However, regarding Question 2, I noticed that the response primarily focuses on sequential learning within the same order (e.g., first-order effects, as in the provided example). My original question, however, was aimed at understanding the strategy for transitioning from lower-order effects (e.g., main effects) to higher-order effects (e.g., second-order interactions). Specifically, I am interested in how the model approaches the sequential progression between these levels of effects in practice.
> >
> > I would appreciate it if the authors could provide further clarification or elaborate on this aspect, as it would help better understand the practical implementation and scalability of the proposed method when dealing with interactions across different orders.
> >
> > Thank you again for your efforts and detailed responses. I look forward to your insights on this matter.

---

> > > ### Author Response · Authors · 2024-11-28
> > >
> > > We appreciate your comments on our paper and have made every effort to address your insightful questions.
> > >
> > > **Question 2.**
> > >
> > > Currently, ANOVA-NODE learns all components simultaneously, and the relationship between sequential learning and stability, as suggested by the reviewer, is very interesting.
> > > Therefore, we conducted additional experiments to investigate whether sequential learning affects stability of component estimation in ANOVA-NODE.
> > >
> > > We considered the Wine dataset and split it into train, validation, and test datasets with a ratio of 0.7/0.1/0.2.
> > > We trained ANOVA-N$^{1}$ODE on the train dataset and determined the optimal epoch using the validation dataset, resulting in the trained ANOVA-N$^{1}$ODE model.
> > > We set the residuals of ANOVA-N$^{1}$ODE with respect to the target variable in the train dataset as the new target variable and trained ANOVA-N$^{2\*}$ODE which only considers second interactions.
> > >
> > > We determined the optimal ANOVA-N$^{2*}$ODE using the validation dataset and calculated the RMSE between the sum of the predictions from ANOVA-N$^{1}$ODE and ANOVA-N$^{2*}$ODE for the input **x** in the test dataset and the target variable of the test dataset.
> > >
> > > We repeated 10 times and obtaind estimated components $\{ \hat{f_{S}^{1}},..., \hat{f_{S}^{10}} \}$ for $S \subseteq [p] , |S| \leq 2$ by either ANOVA-N$^{1}$ODE or ANOVA-N$^{2*}$ODE.
> > > Then, we computed the stability of the components estimated in the same manner as described in Section 4.1 of the paper.
> > > The results are presented in Table A.3
> > >
> > > **Table A.3. Results of Stability score and RMSE.**
> > > | Model  | ANOVA-N$^{2}$ODE| ANOVA-N$^{1}$ODE +  ANOVA-N$^{2\*}$ODE(sequential learning) |
> > > |--------------------------------------------|-------|--------|
> > > | Average of RMSEs (std) $\downarrow$ | 0.704 (0.020) | 0.705 (0.022)  |
> > > | Stability score $\downarrow$ | 0.049 | 0.047  |
> > >
> > > We observed that there was little performance difference between the model that learns all components simultaneously and the model that first estimates the main effects and then sequentially estimates the second-order interactions.
> > > Additionally, the stability scores were nearly identical.
> > >
> > > We believe that under the sum-to-zero condition, each component becomes orthogonal in a certain Hilbert space, which is why this result occurs.

---

> > > > ### Comment · Reviewer_mCTR · 2024-12-02
> > > >
> > > > Thank you to the authors for their diligent experiments.
> > > >
> > > > However, I still believe that drawing conclusions based solely on experiments conducted on a single small dataset is somewhat risky.
> > > >
> > > > Regardless of the paper's acceptance outcome, I believe that if the authors further explore the related content, they could obtain more comprehensive and reliable experimental results and conclusions by extending the proposed method to second-order interactions or beyond.
> > > >
> > > > To my knowledge, this topic is frequently addressed in GAM or related communities as a property of heredity.
> > > >
> > > > I appreciate the efforts the authors have shown during the rebuttal period. However, considering the high standards of the conference, I have ultimately decided to maintain my initial score. Thank you.

---

> > > > > ### Author Response · Authors · 2024-12-02
> > > > >
> > > > > Thank you for your meaningful feedback. It has significantly enriched our work.
> > > > >
> > > > > Due to the short discussion period, we conducted experiments on a small dataset.
> > > > > As suggested by the reviewer, we will perform additional experiments for a sequential learning on various datasets (including high-dimensional datasets) and include the results in the paper.
> > > > >
> > > > > Thank you.

---

### Meta-Review · Area_Chair_fysU · 2024-12-17

**Metareview:**

The paper proposes ANOVA-NODE (Neural Oblivious Decision Ensemble), a novel interpretable framework that enhances functional ANOVA models by addressing their identifiability issues.
Through a series of experiments, the paper demonstrates the effectiveness of the proposed approach, particularly in resolving these identifiability challenges.

While the reviewers acknowledge that the paper is generally clearly written and the experimental results are of interest, they raise the following concerns:
- The paper’s organization could be improved to enhance clarity, with a particular focus on better highlighting the motivation, novelty, and contributions of the work (suggested by Reviewers z8qZ and yGHL).
- The experimental evaluation is limited, and the reviewers request more extensive experiments on larger and/or high-dimensional datasets (raised by Reviewers mCTR and jRHP), which they feel limits the overall significance of the work.

This is a borderline paper.
During the rebuttal phase, the authors actively engaged with the reviewers, resulting in improvements to the paper.
However, despite these productive interactions, the reviewers remain unconvinced about the significance of the contribution. As a result, I recommend rejecting the paper.

**Additional Comments On Reviewer Discussion:**

The reviewers raised the following concerns:
- Reproducibility issues (raised by Reviewers mCTR and yGHL): This concern was successfully addressed by the authors during the rebuttal.
- Presentation issues and/or limited novelty (raised by Reviewers jRHP, z8qZ, and yGHL): While the authors engaged actively with Reviewers z8qZ and yGHL, this concern was **only partially resolved**, and the reviewers did not reach a clear consensus on the significance of the work.
- Implementation details and hyperparameter tuning: This issue was successfully addressed by the authors during the rebuttal.
- Additional experiments on larger and/or high-dimensional datasets (requested by Reviewers mCTR, jRHP, and yGHL): This concern was also **only partially resolved** by the authors, leaving room for further improvement.

I have carefully taken all of the above points into account in reaching my final decision.

---

### Decision · Program_Chairs · 2025-01-22

Reject